# A Global Space-based Stratospheric Aerosol Climatology (Version 2.0): 1979-2018

Mahesh Kovilakam[1,2], Larry W. Thomason[2], Nicholas Ernest[1], Landon Rieger[3], Adam Bourassa[3], and Luis Millán[4]

[1]SSAI, Hampton, Virginia, USA
[2]NASA Langley Research Center, Hampton, Virginia, USA
[3]University of Saskatchewan, Saskatoon, Saskatchewan, Canada
[4]NASA Jet Propulsion Laboratory, California Institute of Technology, Pasadena, California, USA

**Correspondence:** Mahesh Kovilakam (mahesh.kovilakam@nasa.gov)

**Abstract.** A robust stratospheric aerosol climate data record enables the depiction of the radiative forcing of this highly variable component of climate. In addition to the radiative forcing, stratospheric aerosol also plays a key role in the chemical processes leading to ozone depletion. Therfore, stratospheric aerosol is one of the crucial parameters in understanding climate change in the past and potential changes in the future. As a part of Stratospheric-tropospheric Processes and their Role in Climate (SPARC) Stratospheric Sulfur and its Role in Climate (SSiRC) activity, the Global Space-based Stratospheric Aerosol Climatology (GloSSAC) was created (Thomason et al., 2018) to support the World Climate Research Programme (WCRP)'s Coupled Model Intercomparison Project Phase 6 (CMIP6) (Eyring et al., 2016). This data set is a follow-on to one created as a part of Stratosphere-Troposphere Process and their Role in Climate Project's Assessment of Stratospheric Aerosol Properties (ASAP) activity(SPARC, 2006) and a data created for Chemistry-Climate Model Initiative (CCMI) in 2012 (Eyring and Lamarque, 2012). Herein, we discuss changes to the original release version including those as a part of v1.1 that was released in September 2018 that primarily corrects an error in the conversion of Cryogenic Limb Array Etalon Spectrometer (CLAES) data to Stratospheric Aerosol and Gas Experiment (SAGE) II wavelengths, and the new release, v2.0. Version 2.0 is focused on improving the post-SAGE II era (after 2005) with the goal to mitigate elevated aerosol extinction in the lower stratosphere at mid and high latitudes noted in v1.0 as noted in Thomason et al. (2018). Changes include the use of version 7.0 of Optical Spectrograph and InfraRed Imaging System(OSIRIS), the recently released Cloud-Aerosol Lidar and Infrared Pathfinder Satellite Observations (CALIPSO) Lidar Level 3 Stratospheric Aerosol profile monthly product, and the new addition of SAGE III/ISS. Here, we use an observed relationship between OSIRIS extinction at 750 nm and SAGEII/ SAGE III/ISS extinction at 525 nm to derive Altitude-Latitude based monthly climatology of Angstrom exponent to compute OSIRIS extinction at 525 nm, resulting in a better agreement between OSIRIS and SAGE measurements. We employ a similar approach to convert OSIRIS 750 nm extinction to 1020 nm extinction for the post-SAGEII period. Additionally, we incorporate the recently released standard CALIPSO stratospheric aerosol profile monthly product into GloSSAC with an improved conversion technique of 532 nm backscatter coefficient to extinction using an observed relationship between OSIRIS 525 nm extinction and CALIPSO 532 nm backscatter. SAGE III/ISS data is also incorporated in GloSSAC to extend the climatology

to the present and to test the approach used to correct OSIRIS/CALIPSO data. The GloSSAC v2.0 netcdf file is accessible at https://doi.org/10.5067/glossac-l3-v2.0 (Thomason, 2020).

## 1 Introduction

Stratospheric aerosols play a key role in determining the chemical (e.g. Hofmann and Solomon, 1989; Fahey et al., 1993;
Solomon et al., 1996) and radiative (e.g. Minnis et al., 1993; Ridley et al., 2014) balance of the atmosphere. Variations in the stratospheric aerosol levels due to volcanic activity can impact climate on the magnitudes from subtle surface temperatures changes (e.g. Solomon et al., 2011) to the more profound surface cooling (e.g. Hansen et al., 1992; Robock and Mao, 1995; Robock, 2000) and associated precipitation changes (e.g. Haywood et al., 2013; Iles et al., 2013).

The impact of volcanic eruptions is often studied using Global Climate Models (GCM). Some of these modeling studies
(e.g. Stenchikov et al., 2006; Berdahl and Robock, 2013; Fyfe et al., 2013) rely on observational data to represent stratospheric aerosols while others (e.g. Aquila et al., 2013; Mills et al., 2016; Timmreck et al., 2018) interactively model stratospheric aerosol variations. A robust stratospheric aerosol climatology can play a key role to the success of both approaches to modeling the effects of stratospheric aerosol either as a direct input to GCMs or as an observational data set with which to verify the performance of an interactive aerosol scheme (e.g. Aquila et al., 2013; Mills et al., 2016). GloSSAC dataset has also been used
in an observational study (Stocker et al., 2019) to quantify the temperature footprint of 21st century eruptions that enables to better quantify temperature trends related to anthropogenic forcing. As a part of SPARC's SSiRC activity, GloSSAC was created (Thomason et al., 2018) to support, among other endeavors, the WCRP's CMIP6 project (Eyring et al., 2016; Zanchettin et al., 2016). This data set spans from 1979 to 2016 and is a follow-on to one created as a part of SPARC's Assessement of Stratospheric Aerosol Properties (ASAP) activity (SPARC, 2006). Herein, we report on the development of GloSSAC version
2.0 data that extends the data set through 2018. While data sources are mostly unchanged from earlier versions, there are significant improvements in the use of OSIRIS and CALIPSO data with inclusion of SAGE III/ISS data for the first time. As a result, a major version change is warranted and users should use this version even for periods prior to the end of v1.1 (December 2016).

## 2 The construction of version 2.0

Figure 1 depicts the measurements that are currently used for constructing GloSSAC data. While Thomason et al. (2018) discusses about the measurements that have been used in GloSSAC v 1.0 dataset in detail, some of the main features of entire GloSSAC v2.0 dataset including various space based measurements, their limitations and some challenges are worth mentioning here. We divide the entire dataset into three periods based on the measurements used. The first period being the pre-SAGE II period (January 1979- September 1984), followed by SAGE II period (October 1984 - August 2005) , post-SAGE
II period (September 2005- May 2017), and SAGE III/ISS period (June 2017-present). Pre-SAGE II period data mostly consists of data from solar occultation measurements such as SAM II , SAGE and some surface based Lidar measurements (Thomason

et al., 2018). For SAGE II period, the measurements are dominated by solar occultation measurements that provide multi-wavelength measurements for size information. For the post-SAGE II era, we are limited to single wavelength measurements from OSIRIS and/or CALIPSO. While OSIRIS and CALIPSO continue to make daily global measurements with a less direct measurement of aerosol extinction coefficient that requires further assumption of particle size, additional direct measurements

of aerosol extinction coefficient from SAGE III/ISS are now available that provides a roughly monthly coverage of multi-wavelength measurements since June 2017.

We now use OSIRIS version 7.0 rather than the previous release version 5.07. OSIRIS (2001 to present) remains a key component of the GloSSAC data as it is the only data set that spans both the SAGE II period (1984-2005) and the start of the SAGE III/ISS mission in 2017. In version 5.07, aerosol extinction at 750 nm was retrieved at approximately 2 km

resolution using a multiplicative relaxation technique (Bourassa et al., 2012). OSIRIS version 7.0 data (Rieger et al., 2019) uses a multiwavelength retrieval that improves the accuracy of the extinction product by reducing sensitivity to the unknown particle size distribution in the inversion. With these changes, the retrieved extinction coefficient at 750 nm is in better agreement with observations by SAGE II and SAGE III/ISS than the version used in GloSSAC v1.0 (v5.07). A detailed description of the retrieval process and comparisons to SAGE II and SAGE III/ISS are available in Rieger et al. (2019). It should be noted that

while the OSIRIS mission continues, data for 2018 is not available at the time the data record was developed and thus not included in the GloSSAC data set.

We continue to make use of CALIPSO's Cloud-Aerosol Lidar with orthogonal Polarization (CALIOP) aerosol backscatter coefficient measurements. The CALIPSO mission (Winker et al., 2009) has been providing cloud and aerosol profiles since 2006. CALIOP data was used in GloSSAC v1.0 during the post-SAGE II era, critically in regions and time periods where

OSIRIS data are not available. For GloSSAC v1.0, we used CALIOP version 4.0 level 1 aerosol data product at 532 nm and identified and removed observations that suggested the presence of cloud using the depolarization measurement. The remaining data was compiled into the spatial temporal resolution used in GloSSAC (monthly, 5 deg latitude bins, and 0.5 km altitude bins). Since the release of GloSSAC v1.0, the CALIPSO team has released a standard stratospheric aerosol extinction and backscatter products (Kar et al., 2019) in a spatial/temporal resolution compatible with GloSSAC. This data product is now used rather

than the level 1 data. While we use the standard CALIPSO stratospheric aerosol product, enhanced levels of aerosol extinction in the lower stratosphere are consistently noted in the entire dataset after comparing against OSIRIS and SAGE III/ISS. We, therefore decided to use a conformance process which is described below that helps reduce the bias in the lower stratosphere and also at higher latitudes.

New to GloSSAC, we now use version 5.1 data from the latest series of Stratospheric Aerosol and Gas Experiment on Inter-

national space station (SAGE III/ISS) satellite. Like SAGE II, SAGE III/ISS uses the solar occultation technique (McCormick et al., 1979) to make near global measurements of stratospheric aerosol, ozone, water vapor, and nitrogen dioxide with a vertical resolution of 1 km on a roughly monthly time scale. Aerosol extinction coefficient at 9 wavelengths are provided including at the GloSSAC standard wavelengths near 525 and 1020 nm. SAGE III/ISS data are available from June 2017 and we incorporate the data into GloSSAC from that time forward as the primary data set similar to the way in which SAGE and SAGE II

are used earlier in the data record.

For individual instruments, we have expanded the practice of including data in the original parameter and wavelength as possible (e.g. OSIRIS extinction at 750 nm, and CALIOP backscatter coefficient) in the GloSSAC spatial temporal resolution and now include most data sets as components of the GloSSAC data set. The point of origin of each individual dataset is clearly denoted for all data points in the now 40-year record at the primary 525 and 1020 nm wavelengths. As with v1.0, we recommend using original, unconformed, and uninterpolated data for comparison purposes. Conformance follows the basic paradigm of v1.0 but has changed in some instances to reflect better understanding of instrumental differences (Thomason et al., 2018). SAGE II version 7.0 continues to act as the standard throughout the data set. As a result, similar to GloSSAC v1.0, data from other platforms and sometimes made of different aerosol properties and/or at different wavelengths are made, on average, to match or 'conform' with SAGE II at 525 and 1020 nm. This is mostly done using simple linear scaling factors rather than retain offsets that may be created by switching from one data source to another. For instance in v1.0, the conversion factor used for 532 nm backscatter coefficient to 525 aerosol extinction coefficient (53 sr) was selected to minimize the overall difference between SAGE II and CALIOP backscatter data (Thomason et al., 2018). An exception to this process was higher than expected extinction in the lower stratosphere in mid and high latitudes during the CALIOP/OSIRIS era based on observations at similar overall aerosol levels during the SAGE II period. Both GloSSAC v1.0 and v1.1 retained these offsets in extinction as, at the time, it was not clear if the difference was due to real geophysical variability possibly driven by volcanic activity or due to deficiencies in conversion process or in the source data itself. A goal of this release was to understand and, as necessary, mitigate this discrepancy (Thomason et al., 2018). While the changes in the both CALIOP and OSIRIS source data sets have reduced the apparent anomaly in extinction inferred for this period, both data sets, adjusted using the conformance approaches used in v1.0, continue to suggest higher levels of aerosol extinction coefficient in the lower stratosphere than observations by SAGE II would suggest . At the same time, very similar discrepancies were noted between OSIRIS and CALIOP data and SAGE III/ISS observations some 12 years later. Given these facts, we are now confident that these differences are due to measurement/conformance deficiencies and we have developed new techniques to bring these data sets into greater conformity with SAGE II; they are discussed below.

Additional changes to this GloSSAC version relative to the v1.0 as described in Thomason et al (2018) include all changes made in the interim version 1.1. The changes made to interim version 1.1 is described below in Section 2.1. Within v2.0, a key data set (McCormick et al., 1979) used in the SAGE/SAGE II gap period (1982-1984) has been updated. We have expanded the GloSSAC v2.0 netCDF to include stratospheric aerosol optical depth at the primary reporting wavelengths of 525 and 1020 nm and we now retain the native measurements for all instrument data sets (e.g., CALIOP backscatter coefficient at 532 nm) as well as values after conversion to 525 and/or 1020 nm. We also now include reported measurement uncertainty and zonal variability where available at both the native measurement form and at the GloSSAC extinction wavelengths. The updated file resides at NASA's Atmospheric Sciences Data Center and has a unique DOI:10.5067/GLOSSAC-L3-V2.0 .

## 2.1 Changes prior to the end SAGE II period

### 2.1.1 Updates to the SAGE II Era (October 1984-August 2005)

Since the release of GloSSAC v1.0, an interim version was released that corrected a coding error in the initial implementation of the conversion of CLAES infrared aerosol extinction coefficient data to the SAGE II and primary GloSSAC reporting wavelength at 1020 nm. The primary impact was a substantial exaggeration, sometimes exceeding a factor of 2, in the peak aerosol extinction associated with the Pinatubo eruption in the tropics in the several months following the eruption in June 1991. CLAES data becomes available in October 1991 but is used in combination with a ground-based lidar product to estimate the aerosol levels from July to September 1991 and is used standalone for a decreasing span of altitude and latitude until the end of its mission in April 1993 (Thomason et al., 2018; SPARC, 2006). The use of CLAES (Massie et al., 1996) and Halogen Occultation Experiment (HALOE) (Thomason, 2012) data in GloSSAC version 1.1 was also decreased by only employing it to fill latitude/temporal bins in which SAGE II data existed but did not extend as low as the tropopause. This significantly reduced the impact of both CLAES and HALOE in the data set but made spatial and temporal sampling in this period more consistent the sampling throughout SAGE II mission. These changes are continued into v2.0. Some additional quality control (QC) was performed that caught a few defects missed in v1.0.

Compared to the important change requiring the release of version 1.1, only relatively minor additional changes to the SAGE II and pre-SAGE II portions of the data set are included in v2.0. The changes to the SAGE II era within GloSSAC include improved outlier filtering for the SAGE II aerosol extinction coefficient data. In this case, we eliminate as outliers, any data in a GloSSAC bin that exceeds 3.5 median absolute deviations from the median value of all observations within a GloSSAC bin. This filtering has a minimal impact on the results for 452, 525 and 1020 nm but substantially reduces noise in the data at 386 nm, and reduced the need for manual QC repairs of that data set. Users should continue to use caution using the SAGE II 386 nm aerosol extinction coefficient data as a low bias is evident in this data in the lower and upper stratosphere and at all altitudes as aerosol extinction coefficient magnitudes approach background levels (Thomason et al., 2008, 2018). Where aerosol extinction is only available at 1020 nm, the extinction coefficient data sets at 452 and 386 nm are now filled using the same mechanism to fill missing data at 525 nm so that all four wavelength arrays are complete above the tropopause throughout the SAGE II era, as is described in Thomason et al. (2018).

In addition, with the apparent success of filling the high latitudes using the equivalent latitude/latitude mechanism developed based on Manney et al. (2007) for v1.0 (Thomason et al., 2008), we have reduced the role of simple linear interpolation at high latitudes and allow the new equivalent latitude/latitude mechanism fill more of the missing data at high latitudes. While the overall changes with this approach are small, it did reduce the need for manual QC of the data set as the interpolation process was apparently responsible for creating some of that faulty data. The CLAES (Massie et al., 1996) and HALOE (Thomason, 2012) data sets now include zonal standard deviation and median reported measurement uncertainty following the approach used for SAGE II data. The conversion of CLAES and HALOE data follow the methodology described in Thomason et al. (2018). In addition, we now compute the standard deviation of the extinction conversion as a function of the CLAES/HALOE

extinction and use this as uncertainty in the conformed 525 and 1020 nm data products as this source of uncertainty is significantly larger than the reported measurement uncertainties.

### 2.1.2 Updates to the pre-SAGE II Era (January 1979- September 1984)

A component contributing to pre-SAGE II era in previous GloSSAC versions has been completely revised. In v1.0, SAM
II 1000 nm aerosol extinction profile data on potential temperature surfaces was used for input in middle and high latitudes between the end of the SAGE I period in November 1981 and the start of the SAGE II data set in October 1984. It was originally created for use in the ASAP data set (SPARC, 2006) and the original code has been lost. While the revised file was nominally created in same way as the existing data file, the values extended along the isentropic surfaces can be significantly smaller at times than those used in v1.0 particularly in the Southern Hemisphere during the Spring. The cause for this difference is
not clear but may be partly due to changes in the potential temperature fields used in the original construction (from NCEP) compared to the new version (from MERRA 2). This data is used at high latitudes in the northern hemisphere (above 60N) but is a key component in the southern hemisphere south of 30S. Generally, the lack of mid-latitude data in the Southern Hemisphere is a significant limiting factor to the quality of the GloSSAC depiction of stratospheric aerosol during the gap between the end of the SAGE I mission and the start of the SAGE II mission.

## 2.2 Updates to the Post-SAGE II space-based Era for v2.0

Data from a large number of space-based instruments are available for stratospheric aerosol from 1979 through 2018 (see Figure 1). As with v1.0, GloSSAC uses as few instruments as necessary to complete the climatology. We make this decision to limit the impact of differences between instruments due to measurement techniques and wavelength range as well as an assessment of the general quality of the instrument's data set. Before September 2005, the data set is dominated by solar occultation
instruments with the end of SAGE II mission effectively marking the end of the solar occultation era (Thomason et al., 2018). After the end of the SAGE II mission in August 2005, GloSSAC data are exclusively dependent on observations by OSIRIS and CALIOP until mid-2017 and this dependence is not simply a change in instrument but also represents a fundamental change in the measurements provided by these instruments. A positive factor is that these instruments provide near global daily coverage compared to the roughly monthly rate from a solar occultation instrument such as SAGE II. These instruments measure aerosol
using techniques that are less direct than those by solar occultation and whose accuracy is dependent to some extent on aerosol properties that are not directly observed. OSIRIS observations at 750 nm, for instance, are dependent on estimates of the aerosol scattering phase function which relates to the aerosol size distribution and composition. This relationship plays a role in converting the measured radiance to aerosol extinction coefficient and can cause a bias in the product, even though the v7 algorithm attempts to reduce this dependence. OSIRIS stratospheric aerosol measurements can also be sensitive to the presence
of cloud and thus requires caution in using OSIRIS observations near the tropopause, particularly in the tropics. CALIOP uses lidar to measure stratospheric aerosol backscatter coefficient at 532 nm. While this instrument provides by far the greatest density of measurements the precision of individual measurements in the stratosphere is poor and substantial averaging is required to provide GloSSAC compatible profiles with a precision comparable to those provided by either OSIRIS or a SAGE

instrument. The accuracy of CALIOP data is strongly dependent on a normalization process that assumes that a region of the stratosphere (>37 km) is aerosol free; an assumption that is never correct and subjects CALIOP to potential bias in it's backscatter coefficient measurements. Finally, the conversion from backscatter coefficient to extinction coefficient presents a source of bias; as this process depends on details of an unknown aerosol composition and size distribution (Kar et al., 2019)

that is a another potential source of bias. For the CALIOP stratospheric aerosol product, an aerosol extinction-to-backscatter ratio of 50 sr (Kar et al., 2019) has been used — a value that is typically used for background stratospheric aerosol (Jäger and Deshler, 2002, 2003; Illingworth et al., 2015; Kremser et al., 2016).

A new data set included in GloSSAC v2.0 is SAGE III/ISS. This instrument, whose mission began in June 2017, is an updated version of the SAGE III/Meteor 3M instrument and works in a manner substantially the same as SAGE II. Within

10 GloSSAC, SAGE III/ISS plays a crucial role in understanding the apparent enhanced lower stratospheric aerosol throughout the OSIRIS/CALIOP period by comparing results from the on-going overlap period. Given the potential issues in the usage of OSIRIS and CALIOP in GloSSAC and the new information provided by the new SAGE III mission, we have completely revised the analysis process for these instruments for v2.0. This process is described below.

### 2.2.1 Cloud Clearing Method

Stratospheric aerosol measurements by CALIOP, OSIRIS, SAGE II, and SAGE III/ISS can be affected by the presence of cloud in the lower stratosphere/near tropopause. However, for the purposes of GloSSAC, clouds are considered an interfering species and measurements that are inferred to be influenced by the presence of clouds are identified and eliminated from further use in the data set. Generally, clouds are found in the lower stratosphere (as inferred from the MERRA tropopause) and downward and in the wintertime polar vortex as polar stratospheric clouds (PSCs). The efficacy of PSC identification is variable from

instrument to instrument (including SAGE II): ice PSCs are identified effectively in all source data sets but the efficacy of saturated ternary solution (STS) and nitric acid trihydrate (NAT) PSC identification is a mixed with at least some likely to be identified as aerosol and retained in the analysis. CALIOP stratospheric aerosol product is cloud-cleared and no additional cloud processing is necessary. While the OSIRIS version 7.0 aerosol data product is similarly cloud screened (Rieger et al., 2019),we, however found some additional clearing was beneficial to the analysis. Cloud identification process for OSIRIS

is a single wavelength process and based on the straightforward observation that the presence of some cloud among mostly aerosol observations skews the distribution of aerosol extinction coefficient toward larger values (Thomason and Vernier, 2013). Assuming that such positive outliers contain cloud, we have devised a simple statistical approach to cloud identification based on Interquartile Range (IQR). This technique is based on median statistics rather than mean as the extinction coefficient distribution at altitudes from the lower stratosphere into the troposphere may be skewed by the presence of cloud. Measurements

that include the presence of cloud may have extinction up to several orders of magnitude larger than local aerosol extinction coefficient that, in turn, have large impacts on standard statistical quantities like standard deviation. In the presence of unknown but potentially large outliers, IQR is a more conservative measure of the spread of a distribution than standard deviation (Iglewicz and Hoaglin, 1993).

In our implementation, we use yearly data at each altitude (0.5 km) and latitude (5 degree) bin to determine an extinction coefficient probability density function. We used lower quartile (Q1) and upper quartile (Q3) of the underlying distribution to find IQR, which is defined as Q3-Q1, a good measure of the spread in the data relative to median. Here, an extreme outlier is defined as Q3+(3.5*IQR) and a more conservative outlier (Q3+(1.5*IQR)) are used for comparison (Iglewicz and Hoaglin, 1993). A complicating factor is the presence of several small volcanic events during the OSIRIS lifetime and a major pyrocumulus event (August 2017). It is possible that an aggressive outlier process could misidentify measurements of fresh material from either phenomena as cloud events leading to their erroneous exclusion from the data set. Of course, the very nature of GloSSAC is not conducive to a meaningful depiction of the early evolution of recent volcanic events (Thomason et al., 2018). Figure 2 shows the probability density function of aerosol extinction at 750 nm for two cases in 2011 at 18 km with the threshold used (dashed vertical line) for the outlier detection, whereas Figure 3 demonstrates vertical profile of threshold values with mean and median of the distribution. Figure 3 clearly shows the difference between mean and the median profile as the mean profile has higher extinction values in the lower stratosphere due to skewness in the mean statistics. We found that the conservative outlier appeared to remove many enhanced aerosol measurements particularly when stratosphere is perturbed due to volcanic/pyrocumulus events, whereas the extreme outlier was effective at identifying outliers in the density distribution. Therefore we use the extreme outlier to clear cloud-affected observations from the data set.

For SAGE II cloud clearing, we use the ratio of aerosol extinction at two wavelengths (525 and 1020 nm) as a stand in for aerosol size as well as the magnitude aerosol extinction coefficient. In this space, away from major volcanic events like Pinatubo, clouds manifest a large extinction with 525 to 1020 nm aerosol extinction ratio near 1. Typical stratospheric aerosol manifests smaller extinction magnitudes and with extinction ratios between 2 and 4. Cloud identification is complicated by mixed fields of view where observations transition between mostly cloudy extinctions and extinction ratios and those more typical of purely aerosol. As a result, the distribution of extinction and extinction ratio shows a continuum between clearly cloud and clearly aerosol observations that makes distinguishing purely aerosol measurements from those affected by cloud ambiguous. Various techniques to parse these mixed measurements have been developed and GloSSAC makes use of the technique developed by Thomason and Vernier (2013). A similar technique for SAGE III/ISS is in development but not currently available. For this version of GloSSAC, we note 1) no large volcanic aerosol event have occurred during the SAGE III/ISS mission prior to the end of 2018 and 2) the overall aerosol loading of the stratosphere in this period is low. As a result we can eliminate most observations containing cloud by eliminating all observations where the extinction ratio is less than 2 below 24 km. Given the timing of observations, SAGE III/ISS through 2018 had not observed any polar stratospheric clouds. Not surprisingly, this approach misses some mixed cloud-aerosol observations but these are effectively identified using the IQR-based cloud detection algorithm used with OSIRIS.

## 2.3  Comparison of OSIRIS with SAGE II and SAGE III/ISS measurements

Within the GloSSAC paradigm, it is important to not simply observe the agreement between OSIRIS and SAGE II but also mitigate these differences as effectively as possible. We compare SAGE II and OSIRIS once the data have been incorporated into GloSSAC measurement grid: zonally averaged monthly level 3 product at 5 degree latitude and 0.5 km altitude resolution.

OSIRIS version 7.0 standard aerosol product is reported at a wavelength of 750 nm. During the overlap period (2002-2005), we can directly compare these measurements and develop a technique to close observed mutual biases. This technique can then be applied to the entire OSIRIS data record with the assumption that the basic nature of the bias is consistent throughout the period. This is an assumption that may be challenged by a number of small volcanic eruptions occurring during this period

which likely change the nature of the aerosol size distribution but for which little means to assess and mitigate are currently available. In addition to 750 nm, OSIRIS extinction can be calculated at 525 nm, a SAGE II measurement wavelength, using a constant Angstrom exponent of 2.33 to convert from 750 nm to 525 nm (Rieger et al., 2015). The one critical exception to this occurs after mid-2017 when the SAGE III/ISS mission begins. SAGE III/ISS makes extinction coefficient measurements at 521 and 756 nm that are near both wavelengths at which OSIRIS reports extinction coefficient. Since the SAGE III/ISS instruments

operates in a manner similar to SAGE II, the expectation is that there would be minimal bias between these instruments at least at the strongest aerosol measurement wavelengths of 525 and 1020 nm.

A minimal conformance test is that the comparison between SAGE II and OSIRIS in the first overlap period should be very similar to the comparison between SAGE III/ISS during the second overlap period. We expect that the observed correspondence between OSIRIS, SAGE II and SAGE III/ISS will show this consistency only if 1) the performance of OSIRIS does not change

with time [1], 2) SAGE II and SAGE III/ISS are relatively unbiased with each other [2], and 3) the state of the stratosphere has not changed to the point where it has had a deleterious impact on OSIRIS aerosol retrievals. In figures 4a and 4b, it is apparent that for much of the stratosphere the difference between SAGE II and OSIRIS is less than 10% particularly in Figure 4a for a rather benign (less affected by volcanic/fire events) October 2004. However, it is also clear that OSIRIS extinction is consistently higher than SAGE II in the lower stratosphere with percentage difference exceeding 50% near the tropopause.

Another departure is shown in Figure 4b for March 2005 that shows similar features as October 2004. However, in the tropical low and middle stratosphere there is a difference of about 50% in an enhanced aerosol layer associated with the eruption of Manan in January 2005. Manam is typical of small eruptions observed by SAGE II and later by SAGE III in which a substantial increase in aerosol extinction coefficient occurs with an increased in 525-to-1020 nm aerosol extinction ratio. This suggests that extinction is dominated by smaller particles in the volcanic layer than prior to the eruption and is much different than the

large extinction/large particle size associated with larger eruptions such as the June 1991 eruption of Pinatubo. Similar small volcano effects are noted in the SAGE II data associated with Ruang in 2002 and by SAGE III/ISS Ambae in July 2018. Figure 5 shows a measurement comparison between June 2017 OSIRIS at 525 nm and SAGE III/ISS at 521 nm (a) and between OSIRIS at 750 nm and SAGE III/ISS at 756 nm (b). Since OSIRIS data is only available through the end of 2017 at this time, relatively little comparative data is currently available. Comparisons between OSIRIS and SAGE III/ISS are complicated

in the overlap period by the pyrocumulus fire event in August 2017 which was inhomogeneously distributed in the northern hemisphere throughout the second half of 2017. OSIRIS extinction at 750 nm is in reasonable agreement with SAGE III/ISS

---

[1]While the OSIRIS instrument performance has relatively remained unchanged over time, the scattering angle has slowly drifted, and the fraction of ascending/descending node measurements has changed. These factors may affect overall data quality.

[2]While the differences between SAGE II and SAGE III meteor aerosol extinction coefficient are relatively smaller, some previous studies (Thomason et al., 2010; Damadeo et al., 2013) reported a small bias between SAGE II (v 6.2) and SAGE III (v 4.0) meteor that are within $\pm$ 10% for measurement wavelengths of 525 and 1020 nm for the altitudes between 7 and 25 km.

(b) except in the lower stratosphere and in tropical latitudes where the difference can exceed 20% in a pattern similar to that seen between SAGE II and OSIRIS (Figure 5a). However, the differences between OSIRIS at 525 nm and SAGE III/ISS at 521 nm, shown in Figure 5a, are significantly larger than those seen in Figure 4a that suggests either deficiencies in the conversion process of OSIRIS measurements from 750 nm to 525 nm or that SAGE III/ISS 521 nm data is low-biased in the lower and mid stratosphere at 521 nm though probably not at 756 nm. In general, comparisons of OSIRIS suggest general agreement with both SAGE datasets except in the lower stratosphere where it appears that OSIRIS is biased high relative to the SAGE II and III/ISS measurements. The need for long-term consistency among data sets effectively requires that OSIRIS be brought into conformance (bias reduced as much as possible) with SAGE II measurements.

## 2.4 OSIRIS extinction coefficient conformance process

Ultimately, the need to reduce the observed differences between SAGE II and OSIRIS are most relevant during the period between the end of the SAGE II mission and the start of the SAGE III/ISS mission. We have developed a mechanism for this in which we have derived a monthly mean Angstrom exponent (hereafter "pseudo Angstrom exponent") for the overlap period using SAGE II and SAGE III/ISS 525 nm aerosol extinction with OSIRIS data at 750 nm effectively creating a 4-year climatology of pseudo Angstrom exponent based on measured values. Within GloSSAC, the Angstrom model for aerosol extinction wavelength dependence for 525 and 750 nm is given by

$$k_{525[t,m,i,j]} = k_{750[t,m,i,j]} \left(\frac{\lambda_{525}}{\lambda_{750}}\right)^{\eta_{[m,i,j]}}$$

where, $k_{525[t,m,i,j]}$, and $k_{750[t,m,i,j]}$ are extinctions at 525 nm and 750 nm respectively, $\eta_{[m,i,j]}$ is the pseudo Angstrom exponent while the indices $[t,m,i,j]$ represent year, month, latitude, and altitude respectively. $\left(\frac{\lambda_{525}}{\lambda_{750}}\right)$ represents ratio of wavelengths at 525 and 750 nm. All data are gridded to 5 degree latitude and 0.5 km altitude resolution. A monthly median climatology of pseudo Angstrom exponent is then computed and smoothed using a 3X3 median boxcar filter in altitude and latitude. Linear interpolation is used to fill in missing values in the monthly grid. Figure 6 shows altitude versus latitude plots of monthly climatology of pseudo Angstrom exponent and shows a modest annual cycle particularly in the tropical lower stratosphere. While the standard OSIRIS 525 nm aerosol extinction product uses a fixed Angstrom exponent of 2.33, we compute values between 1 and 3 for much of the stratosphere and occasionally are less than 0 in the lower tropical stratosphere. We do not assume that the derived value for pseudo Angstrom exponent has any physical meaning as it accounts for not just the actual behavior of aerosol but also for potential deficiencies in both data sets and it is simply a means to push OSIRIS extinction measurements toward those produced by SAGE II. Using this climatology of pseudo Angstrom exponent values, we can convert any month of OSIRIS data to 525 nm. For example, using the monthly-based pseudo Angstrom exponent for October 2004, the agreement between modified OSIRIS and SAGE II are almost entirely below 10% (Figure 4c). To some extent this is the expected result though it generally suggests that the pseudo Angstrom exponent is reasonably stable throughout the overlap period. However for the March 2005 analysis, OSIRIS predictions for 525 nm remain substantially greater than those measured by SAGE II in the region containing material from the recent Manam eruption (Figure 4d). This suggests that the changes in the observed Angstrom exponent and climatological pseudo Angstrom value remain significantly different during this small

volcanic event. This is not an issue for GloSSAC for this eruption as SAGE II data is used in this period. However during the period between SAGE II and SAGE III/ISS there are a number of similar small volcanic events that could easily have a similar behavior. Figure 7 demonstrates how a variety of small to very large eruptions manifest themselves in the SAGE II/III-ISS data record. It shows the 1020-nm extinction coefficient and 525 to 1020-nm aerosol coefficient ratio at the peak extinction following 5 eruptions (Pinatubo, Nyamuragira/Nevado del Ruiz, Ruang, Manam, Ambae) connected to values that occurred at the same altitude/latitude just prior to the eruption. The Pinatubo eruption, that dominates much of the SAGE II record is by far the largest event and is the only one where the apparent aerosol size increases with the event (decreased aerosol extinction ratio). Most events in this plot show that aerosol 'size' is apparently smaller after a volcanic event than before the event occurred. Sometimes this is fairly subtle (Nyamuragira/Nevado del Ruiz) but sometimes can be very pronounced (Ruang). There is a tendency for smaller eruptions to produce larger extinction ratio but this is not, for this limited sample, sufficiently well behaved to be considered predictive. The conformance process, as it is currently implemented would produce horizontal lines in Figure 7 showing no change in ratio from before an event to afterwards. While this is clearly a shortcoming, we are not able at this time to account for changes in extinction ratio for volcanic events in GloSSAC v2.0 where the data set is based on only one wavelength (either OSIRIS or CALIOP). As a result, it is likely that GloSSAC extinction for small volcanic events during the OSIRIS/CALIOP period will be biased high to an unknown extent. Further study into this period may result in changes in a future version of GloSSAC. The overall conformability of OSIRIS, SAGE II and SAGE III/ISS is ultimately tested by the comparison in Figure 5c which shows the comparison of revised OSIRIS 525 nm aerosol extinction coefficient data, as computed using the SAGE II-based pseudo Angstrom exponent, and SAGE III/ISS measured 521 nm aerosol extinction coefficient. In this comparison, we see that differences that were generally larger than 20% and often in excess of 50% are now reduced to mostly less than 10% except at high northern latitudes at higher altitudes. While these differences are larger than those found with SAGE II and OSIRIS, the new conformance process clearly is a better step forward in combining these data sets into a uniform data set.

## 3   Comparison of CALIOP with OSIRIS and SAGE III/ISS measurements

In GloSSAC v1.0, CALIOP and OSIRIS were used as equal partners in which extinction values from each instrument were used where the other was not available and averaged where both existed. Unfortunately, there is no direct overlap between CALIOP and SAGE II mission lifetimes (missing by about 8 months) so that direct comparisons of these data sets is not possible. In v1.0, CALIOP 532 nm backscatter coefficient was converted to 525 nm extinction using the mean ratio of OSIRIS 525 nm extinction to CALIOP backscatter coefficient in the GloSSAC data set. This value, 53 sr, is roughly consistent with the extinction-to-backscatter ratio used within CALIOP data processing (50 sr). In v1.0, CALIOP converted 525 nm aerosol extinction is roughly consistent with OSIRIS; particularly in producing more aerosol extinction in the lower stratosphere than would be expected based on similar, but in this case not contemporaneous, SAGE II values. As a result, we compare the recently released standard CALIOP extinction coefficient product (Kar et al., 2019) with conformed OSIRIS and SAGE III/ISS data. All three data sets are used at the GloSSAC spatial/temporal resolution. Since CALIOP stratospheric aerosol extinction is reported

at 532 nm, a constant Angstrom exponent of 2.33 is used here for the conversion of particulate extinction from 532 to 525 nm (an adjustment of 2%). Figure 8a and 8b show relative difference plots between OSIRIS/ SAGE III/ISS and CALIOP for November 2017. Equatorward of 30 degrees and above roughly 18 km, the differences between the CALIOP extinction product and OSIRIS and SAGE III/ISS are generally between $\pm$ 20%. However, the CALIOP standard extinction coefficient product is much larger (>50%) in the lower stratosphere globally as well as in the entire stratosphere poleward of 40S and 40 N. While some of these differences may be attributable to the wildfire-driven pyrocumulonimbus (PyroCb) events (e.g. Peterson et al., 2018), similar discrepancies persist even when the stratosphere is unperturbed by any volcanic/PyroCb events.

## 3.1 Conforming CALIOP backscatter coefficient to GloSSAC extinction coefficient

Due to the discrepancy between OSIRIS/SAGE III and CALIOP, we conform CALIOP data using an empirical scaling factor (SF) which is defined as the ratio of bias corrected OSIRIS extinction at 525 nm and CALIOP backscatter coefficient at 532 nm. It is analogous to an extinction-to-backscatter coefficient except that it also attempts to account for bias between OSIRIS and CALIOP and thus the SF should not be viewed as reflecting only underlying aerosol properties. We compute the SF using monthly OSIRIS 525 nm extinction and CALIOP 532 nm particulate backscatter coefficients. The particulate backscatter and extinction products in the standard CALIOP level 3 stratospheric data are retrieved using a lidar ratio of 50 str (Kar et al., 2019). As we are effectively revising this factor, we do not use products derived using it but rather rederive the backscatter coefficient using the attenuated scattering ratio that is also reported as a part of this product. In this approach, we also assume that the transmission of the atmosphere is close to 1 throughout the stratosphere which allows us to neglect the attenuation term in the equation of total attenuated backscatter. With this approximation, the particulate backscatter coefficient is computed using simplified formula (SR*MBKS)-MBKS, where SR is the scattering ratio and MBKS is the molecular backscatter and is also provided in the CALIOP stratospheric aerosol product. Since the assumption that the transmission of stratosphere is close to 1 is clearly not correct, it is not surprising that the recomputed backscatter coefficient is somewhat less than with the level 3 retrieved particulate backscatter. We find in general that the difference increases with decreasing altitude and is about 10% difference at 18 km and close to 30% at 10 km and that the relative differences exhibit low variance at any given altitude/latitude/time bin. Since, in order to conform the CALIOP data to the conformed OSIRIS data at 525 nm, we are forced to use SFs that are empirical and account for several effects including both aerosol-related effects and bias between the two data sets, it ultimately does not matter a great deal whether we use the standard CALIOP stratospheric backscatter product or the alternative product described above. For GloSSAC v2.0, we choose to use the alternative backscatter product but users should keep in mind that this choice merely adjusts the size and perhaps the physical meaning of the SF and not the outcome of the 525-nm extinction coefficient estimation. As the CALIOP stratospheric product evolves, we will reconsider the approach in future releases of GloSSAC. Figure 9a shows annual median of OSIRIS 525-nm extinction to CALIOP 532-nm backscatter ratio as a function of altitude and latitude. Here, the magnitude of the SF varies from between 25 and 65 sr except at higher altitude at polar latitudes where SFs are less than 10 sr. Figure 9b shows the standard deviation for this depiction and we find that the overall behavior of the SF is reasonably consistent throughout the entire period. To implement this conversion process, we use monthly median values of the SF for the entire overlap period for OSIRIS and CALIOP (06/06 to 12/17) and apply

the conversion factors to the entire CALIOP data set at a simple multiplication factor dependent on only latitude and altitude. While it is tempting to infer that CALIOP data processing for this product is using the wrong extinction-to-backscatter ratio, it is important to recall that it seems apparent that CALIOP stratospheric extinction values are biased high compared to OSIRIS conformed data and SAGE III/ISS comparisons by consistent margins in the lower stratosphere and high latitudes where the

largest departures of the SF from a value of 50 str occur. It is worthwhile to note here that Kar et al. (2019) also computed stratospheric aerosol 532 nm lidar ratio using extinction coefficients from SAGE III/ISS and backscatter measurements from CALIOP for the period June 2017 through August 2018. We note the pattern of their lidar ratio (Figure 13 of Kar et al. (2019)) is more or less consistent with our SF in Figure 9a.

   Figures 8 includes the percent difference plots between the empirically scaled CALIOP 525 nm extinction and OSIRIS (c)
and SAGE III/ISS (d) for November 2017. It is clear that while the differences are not eliminated, the difference between CALIOP and OSIRIS (Figure 8c) is now mostly within ± 20%, compared to the standard CALIOP extinction shown in Figure 8a where the difference between standard CALIOP stratospheric extinction coefficient and OSIRIS was often more than 50%. Similarly, the difference between the scaled CALIOP stratospheric extinction and SAGE III/ISS is now mostly below 30% except near the tropical lower stratosphere and polar high altitudes, whereas the the difference between standard CALIOP
stratospheric extinction coefficient and SAGE III/ISS is also often more than 50%. Figure 10 shows monthly time series of SAGE II, OSIRIS, bias corrected OSIRIS, CALIOP, bias corrected CALIOP, and SAGE III/ISS extinction for altitudes and latitudes where conformance has been an issue between the data sets. The time series plots show: 1) the OSIRIS and CALIOP extinction coefficient at 525 nm before conformance are substantially higher than those provided by SAGE II/SAGE III/ISS and 2) these differences are substantially reduced after the conformance process. There are still some clear observations where
significant differences among the products exist. Most can be associated with episodic enhancements by a number of small volcanic eruptions (see Table 1) and a major pyrocumulous event in August 2017. Part of these differences may be due to spatial inhomogeneity of the distribution of aerosol in the first several months following eruptions. However, as previously discussed, the conformance processes applied to OSIRIS and CALIOP cannot adequately account for fresh aerosol from these events. Nonetheless, given the status of OSIRIS/CALIOP/SAGE conformance in GloSSAC v1.0, the process used in the new
version represents an imperfect step forward.

## 4   Constructing 1020 nm Extinction Record in the post-SAGE II period

For GloSSAC v1.0 (Thomason et al., 2018), extinction measurements at 525 and 1020 nm were included in the stratospheric aerosol record. The post-SAGE II record was focused on producing a uniform extinction coefficient record at 525 nm and aerosol extinction coefficient at 1020 nm was inferred using an empirical relationship between 525 to 1020 nm extinction
ratio and 525 nm extinction derived from SAGE II data. However, this process was not fully successful and a noteworthy bias between the SAGE II and post-SAGE II period at 1020 nm was observed in v1.0 (Thomason et al., 2018) leading to recommendations to avoid using 1020 data in the v1.0 data set in the post SAGE II part of the record. For GloSSAC v2.0. We use a different strategy focused on using monthly pseudo Angstrom exponents computed using measurements between OSIRIS

and SAGE II/SAGE III-ISS that is functionally identical to the method used in section 4.2 to infer extinction at 525 nm from OSIRIS. We produce a monthly average of pseudo Angstrom exponent using SAGE II/SAGE III/ISS 1020 nm and OSIRIS 750 nm extinction coefficient (Figure 11). Generally, we'd expect to derive a similar pseudo Angstrom exponents for the 750 to 1020 nm conversion as used in the case of inferring OSIRIS 525 nm extinction from SAGE II 525 nm extinction coefficient. However

we find, particularly in the lower stratosphere, significantly larger exponents for the 750-to-525 nm extinction conversion than for the 750-to-1020 nm conversion. For instance, in the lower tropical stratosphere, we find values for the former as large as 0 whereas as comparable values for the latter are less than -3.

We use these monthly pseudo Angstrom exponents to infer an OSIRIS extinction coefficient product at 1020 nm for the post-SAGE II period. Again, we do not propose these exponents as having only information regarding aerosol optical properties as

they also account for systematic differences between aerosol data products. These empirical parameters are simply what are required to bring OSIRIS aerosol extinction data into conformance with SAGE II/III data and fulfill the goals of GloSSAC v2.0. In order to infer 1020-nm extinction coefficient from CALIOP 532 nm backscatter data, we follow the same procedure as we employed to infer CALIOP 525 nm extinction in section 5.1. Figure 9c shows annual median of OSIRIS extinction at 1020 nm to CALIOP 532 nm backscatter ratio. Despite whatever shortcomings these values may have relative to reflecting aerosol

properties, we use these ratios to compute CALIOP extinction at 1020 nm.

Figure 12 shows the monthly time series for SAGE II, SAGE III/ISS and conformed OSIRIS and CALIOP data at some lower stratospheric levels. We find that OSIRIS 1020 nm extinction is in reasonable agreement with SAGE II and SAGE III/ISS 1020 nm extinction coefficient with an overall agreement between OSIRIS and SAGE II/SAGE III-ISS of $\pm 20\%$. It should be noted that there are still some significant outliers between SAGE products and the converted OSIRIS values almost always

associated with significant increases in the stratospheric aerosol burden such as those following Manam volcanic eruption in January 2005 where apparent changes in aerosol size may be playing an out sized role. Figure 12 further shows that CALIOP 1020 nm extinction fits in reasonably well with OSIRIS, and SAGE III/ISS, suggesting that the inferred 1020 nm extinction is fairly robust. It should also be noted that, in some cases (particularly for February 2016 and October 2018) when no CALIOP data is available, we linearly interpolate CALIOP data in time between January (September) and March (November) of 2016

(2018) to fill in the missing monthly data following methods used in interpolating SAGE II data. In GloSSAC version 2.0, the data between August 2005 (after SAGE II mission ends) and June 2006 (CALIOP mission starts), the only data set available is OSIRIS and there are cases when missing data in OSIRIS needs to be filled. In such cases, we either use the closest month data or a linear interpolation when there is more than two consecutive months of data available at each grid point, while assuring the consistency and continuity in the merged data. SAGE III/ISS data is incorporated into the extinction record from June 2017

and we prioritize SAGE III/ISS data over OSIRIS and CALIOP whenever SAGE III/ISS data is available. Figure 13 and 14 show extinction from bias corrected 525 and 1020 nm respectively for OSIRIS, CALIOP, SAGE III/ISS and merged extinction at 47.5 N latitude. Overall, the continuity in the data is maintained after merging different datasets. Uncertainties related to each variable are also included in the final data product (see supplementary information on how uncertainties are estimated).

## 5 Stratospheric aerosol optical depth

A formal stratospheric aerosol optical depth (SAOD) at 525 and 1020 nm is included in GloSSAC v2.0. We compute SAOD by integrating aerosol extinction at respective wavelengths from the monthly-average tropopause in the data set to 40 km following the method described in Thomason et al. (2018). Figure 15 shows latitude versus monthly time series of SAOD from GloSSAC v1.1 , this version (v2.0) and the difference between v2.0 and v1.1 for the entire 40 year record that depicts major and minor volcanic eruptions. Figure 15a shows a consistent 525 nm optical depth enhancement at mid and high latitudes across the transition between SAGE II and OSIRIS in August 2005 which is a reflection of the lower stratospheric discrepancy observed in previous versions. Thomason et al. (2018) suggested that this could be related to the January 2005 eruption of Manam but also recognized that it may be an instrumental artifact. We show SAOD from the merged GloSSAC v2.0 data in Figure 15b where updated versions of OSIRIS and CALIOP data have been conformed to SAGE II (CALIOP indirectly so) and we now include data from SAGE III/ISS. The relative difference between GloSSAC 2.0 and GloSSAC 1.1 is shown in Figure 15c. It is evident from Figure 15c that smallest difference between version 2.0 and v1.1 occurs in the Southern Hemisphere, while in Northern Hemispheric higher latitudes the difference is between 10 and 40%. Compared to the optical depths shown in Figures 15a and 15b, it is clear that while some enhancement after 2005 remains it is substantially smaller than in v1.1. For the period between 1979 and 2005, the differences between version 1.1 and 2.0 are seen during and following El Chichón (1982) and Pinatubo (1991) eruptions. The increase in percent difference in 1991 is mostly driven by a change in version 2.0 to not interpolate from May to July in SAGE II data.

At 1020 nm, the SAOD was noted as a deficiency in v1.0 as the conformance process did not produce satisfactory results for this period. In Figure 15d, it is clear that a much larger discontinuity across the instrument transition in 2005 at 1020 nm than at 525 nm. In Figure 15e, we observe that evidence for a discontinuity across the instrument change in 2005 is significantly reduced and, in fact, it is not clear that one exists at all. The differences in the two versions is shown in Figure 15f where we find that the optical depth for this period is reduced by as much as 50% at high latitudes.

Figure 16 shows a similar plot but for version 1.0 and 2.0. The significant differences between version 1.0 and 2.0 remain somewhat similar to differences between v 1.1 and 2.0 shown in Figure 15. Figure 17 shows monthly time series of GloSSAC v2.0 SAOD at 525 nm for three different latitude bands for the entire 40 year record. Signatures of major volcanic eruptions (e.g. El Chichón in 1982 and Mount Pinatubo in 1991) and several other minor volcanic eruptions as listed in Table 1 in the post-SAGEII era are clearly evident in Figure 17. Figure 17 also shows how individual measurements of OSIRIS, CALIOP and SAGE III/ISS contribute toward the post-SAGE II aerosol record of GloSSAC v2.0. Additionally, a global SAOD time series for the entire record is shown in Figure 18. The percent difference of global SAOD of earlier versions with respect to v 2.0 is shown in Figure 18 (b) and (d). The largest percent difference in SAOD occurs in June 1991 in both 525 and 1020 nm percent difference plot (b,d), which is due to a change in version 2.0 of GloSSAC to not to interpolate from May to July in SAGE II extinction coefficient data that are related to the usage of CLAES data. The difference in SAOD in the post-SAGE II period (after 2005) is due to the changes in version 2.0 that has improved over the older versions. This difference is mostly due to a reduction of SAOD at higher latitudes as they are evident from Figure 15 (c,f). For 1020, the difference between older versions

and version 2.0 is much larger because of the reduction in discontinuity across instruments due to the conformance process used in version 2.0. We also note that in Figure18d, the SAOD has significantly reduced (as much as 50% or more) for 1020 nm as they are evident from Figure 15f as well. These differences may have implications on climate modeling as one of the recent modeling studies (Rieger et al., 2020) shows that the difference between v3.0 and v4.0 of CMIP6 stratospheric aerosol data

that are derived from v 1.0 and v1.1 of GloSSAC causes a reduction in instantaneous top-of-the atmosphere radiative forcing following Pinatubo eruption.

As with v1.0, we cannot exclude the possibility that on-going volcanic activity plays a dominate role in the apparent enhancement after 2005. This possibility is bolstered by noting that optical depths shown in Figures 15b and 15e approach those observed in 2004, 2013, early 2014 during a lull in a decade of repeated minor volcanic stratospheric enhancements. We

also note that several recent modeling studies (e.g. Schmidt et al., 2018; Aubry et al., 2020) using sulfur dioxide emissions in aerosol-climate models, have reported an enhancement in SAOD for the post-2005 time period. Nevertheless, we cannot exclude that some bias across the transition between SAGE II and the OSIRIS/CALIOP period continues to exist. It is possible that with further overlap between SAGE III/ISS and OSIRIS and CALIOP that a better understanding of instrumental differences will permit a more robust conformance process for future versions of GloSSAC.

**6   Conclusions and Future Plans**

Here we present v2.0 of GloSSAC that extends from 1979 through 2018 with the addition of new SAGE III/ISS data toward the end of the record and with some changes to the data used in the post-SAGE II era. We now use OSIRIS version 7.0 data (Rieger et al., 2019) instead of version 5.07, which was used in GloSSAC v1.0. The OSIRIS version 7.0 data is improved in terms of data quality in the lower stratosphere. However, the bias in the lower stratosphere is not entirely resolved as higher

20   percent differences still exist between OSIRIS and SAGE II/SAGE III/ISS. Here, we use a conformance process to reduce the observed differences between OSIRIS and SAGE II/SAGE III/ISS, which is based on monthly pseudo Angstrom exponent computed using 750 nm OSIRIS extinction and 525 nm SAGE II/SAGE III/ISS extinction coefficients. A similar approach is implemented for inferring 1020 nm extinction coefficient. We continue to make use of CALIOP data in this version as well with some changes. At the time of development of GloSSAC v1.0, there was no standard stratospheric aerosol product available from

CALIOP and therefore used its version 4.0 level 1 aerosol backscatter coefficient product at 532 nm and identified and removed observations that suggested the presence of cloud using the depolarization measurement (Vernier et al., 2009; Thomason et al., 2018). The cloud-cleared backscatter coefficients were then compiled into the spatial temporal resolution used in GloSSAC (monthly, 5 deg latitude bins, and 0.5 km altitude bins). Now that the CALIPSO standard stratospheric aerosol product (Kar et al., 2019) is available, we use the standard stratospheric aerosol product. However, we note that standard CALIOP aerosol

extinction coefficient computed at 525 nm tend to overestimate aerosol extinction in the lower stratosphere globally as well as in the entire stratosphere poleward of 40S and 40N. We, therefore use a SF based on conformed OSIRIS extinction coefficient and CALIOP estimated backscatter to conform the CALIOP data. The conformed CALIOP extinction coefficient is in reasonable agreement with OSIRIS and SAGE III/ISS with some exceptions. These changes resulted in an improved version 2.0 data.

Some important changes in version 2.0 include, change in the usage of CALES data during Pinatubo time period that resulted in a reduction of extinction coefficient following Pinatubo, and improved extinction coefficients in the post-SAGE II era data. These changes in version 2.0 reflect in 525 nm SAOD data as about 80% reduction of SAOD occurs in June 1991 when compared against previous versions due to changes in the usage of CLAES data, whereas in the post-SAGE II period we note

no significant change in SAOD at 525 nm. However, for 1020 nm extinction we note a significant improvement in the post-SAGEII era that results in significant reduction (as much as 50% or more) of SAOD in version 2.0 due to the new conformance process based on monthly angstrom exponent, where we observe that evidence for a discontinuity across the instrument change in 2005 is significantly reduced. Overall, the quality and robustness of the stratospheric aerosol product have improved for GloSSAC v2.0 with some issues that still persist in the data set which we mention below:

– Despite using a monthly based measured pseudo Angstrom exponent in converting OSIRIS extinction from its native wavelength of 750 nm to 525 nm, we note that this method has its own limitations during periods when the stratosphere is perturbed due to volcanoes/PyroCb events. OSIRIS 525 nm extinction coefficient is somewhat biased high in such events where we have SAGE II/SAGE III/ISS data to compare. Although, the monthly based pseudo Angstrom exponent correction significantly improves the comparison between OSIRIS and SAGE II extinction at 525 nm, we note some-

what large percent differences (>20%) between OSIRIS and SAGE II following Manam eruption (Figure 4d). Similar instances are also noted when both data are available and more so in the tropical latitudes between 20S and 20N. We, unfortunately do not have SAGE II measurements after August 2005 to validate OSIRIS data. However, toward the end of the record we use SAGE III/ISS to validate OSIRIS and CALIOP. For this time period, the agreement between SAGEIII-ISS and OSIRIS/CALIOP extinction coefficient is mostly within ±20%, but degrades somewhat during and

following Canadian PyroCb events in 2017. Coincident OSIRIS and SAGE III/ISS measurements during this time period are studied in Bourassa et al. (2019).

– Since we use CALIOP stratospheric aerosol data to fill in missing OSIRIS data for the time period from June 2006 onward, issues pertaining to CALIOP data should also be mentioned. The standard CALIOP stratospheric aerosol product provides backscatter coefficient as it needs to be converted to extinction coefficient at the wavelength of interest

for the GloSSAC usage. The standard CALIOP product also provides extinction at 532 nm. However, we note that CALIOP standard extinction product is biased high when compared against SAGE III/ISS and OSIRIS (Figure 8a,b). We, therefore compute particulate backscatter from scattering ratio and molecular backscatter coefficient and further infer extinction coefficient using Altitude-Latitude based scale factor (SF) which is derived from conformed OSIRIS extinction coefficient and estimated CALIOP backscatter coefficient. With this bias correction, bias between OSIRIS

and corrected CALIOP extinction has now been reduced. However, it should also be noted that any bias in OSIRIS data will be reflected in CALIOP inferred extinction as we use OSIRIS extinction in the conversion process. While it gives us confidence in both bias corrected OSIRIS and CALIOP extinction in comparison with SAGE III/ISS that the differences between these data sets are mostly within ±20%, the representation of aerosol extinction coefficient particularly during small volcanic eruptions that occurred during OSIRIS/CALIOP period (2006-2017) is still a challenge with single wave-

length measurements (either OSIRIS or CALIOP). We also note that the bias in the lower stratosphere is not entirely resolved as differences still exist between OSIRIS/CALIOP and SAGE measurements.

- While the inferred 1020 nm extinction in GloSSAC v2.0 for the post-SAGEII era (2005-2018) is improved compared to v1.0, there are limitations with the 1020 nm extinction. We note that deducing size information using 525 and 1020 nm extinction ratio for the period between August 2005 and June 2017 may still be an issue with single wavelength measurement from either OSIRIS or CALIOP, particularly during and following a volcanic event, despite some improvement in the inferred 1020 nm extinction. While this is clearly a limitation, we are not able to address changes in extinction ratio for volcanic events in v2.0 where the data set is based on only one wavelength. As a result, it is likely that GloSSAC extinction for small volcanic events during the OSIRIS/CALIOP period will be biased high to an unknown extent. Further study into this period may result in changes in a future version of GloSSAC. Since June 2017, multi-wavelength extinction coefficient data became available from SAGE III/ISS, giving us an opportunity to compare/validate OSIRIS/CALIOP data particularly during and following such events.

There are additional datasets that are available for stratospheric aerosol extinction coefficients. We plan to evaluate and use these datasets, including SCanning Imaging Absorption spectrometer for Atmospheric CartograpHY (SCIAMACHY) and Ozone Mapping Profiler Suite (OMPS).

## 7 Data File Contents and Accessibility

The contents of the GloSSAC v2.0 netCDF file is similar to that for v1.0. Some additional data records include gridded SAGE III/ISS data at 9 wavelengths and the original unconformed data for CLAES, HALOE, OSIRIS, CALIOP. All space-based data sets include the median reported uncertainty and the zonal standard deviation for each instrument product and for its conformed version in each altitude/latitude/time bin in which data is available. Version 2.0 also includes SAOD at 525 and 1020 nm. The contents of the GloSSAC v2.0 netCDF file are listed in Table A2 with variable name and description. Table A3 lists the data flag values associated with the dataset.

The GloSSAC v2.0 netcdf file is available from NASA Atmospheric Data Center (https://asdc.larc.nasa.gov/data/GloSSAC/ GloSSAC_V2.0.nc) and referenced using its DOI: https://doi.org/10.5067/glossac-l3-v2.0 (Thomason, 2020).

## 8 Data availability

The GloSSAC v2.0 netcdf file is available from NASA Atmospheric Data Center (https://asdc.larc.nasa.gov/data/GloSSAC/ GloSSAC_V2.0.nc) and referenced using its DOI: https://doi.org/10.5067/glossac-l3-v2.0 (Thomason, 2020).

*Author contributions.* MK and LT conceived the idea and methodology used in this paper. MK carried out the analysis, while NE helped with version 1.1 of the data (data prior to 2001). LR and AB provided us with version 7.0 of OSIRIS data, participated in the scientific

discussion in regard to OSIRIS data in particular. LM provided help with the equivalent latitude that was used in filling data for SAGEII era. MK wrote the manuscript, while all authors reviewed the manuscript and provided with advice on the manuscript and figures.

*Competing interests.* The authors declare that they have no conflict of interest.

*Acknowledgements.* We acknowledge the support of NASA Science Mission Directorate, and the SAGE II and III/ISS mission teams. SAGE
5 mission is supported by NASA Science Mission Directorate. SSAI personnel are supported through STARSS III contract. Contributions by AB and LR were supported by the Canada Space Agency through the Earth System Science Data Analysis grant program.

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

**Table 1.** List of significant volcanic eruptions/wild fire events that occurred during the entire record of GloSSAC

| Volcano Name | Eruption Date | Latitude |
|---|---|---|
| St. Helens (He) | 27 Mar 1980 | 46N |
| El Chichón (El) | 4 Apr 1982 | 17N |
| Nevado del Ruiz (Ne) | 14 Nov 1985 | 5S |
| Kelut (Ke) | 10 Feb 1990 | 8S |
| Pinatubo (Pi) | 15 Jun 1991 | 15N |
| Cerro Hudson (Ce) | 12 Aug 1991 | 46S |
| Rabaul (Ra) | 19 Sept 1994 | 4S |
| Ulawun (Ul) | 29 Sept 2000 | 5S |
| Shiveluch (Sh) | 22 May 2001 | 56N |
| Ruang (Rn) | 25 Sept 2002 | 2N |
| Reventador (Rv) | 03 Nov 2002 | 0N |
| Manam (Mn) | 27 Jan 2005 | 4S |
| Soufriere Hills (Sh) | 20 May 2006 | 16N |
| Tavurvur (Tv) | 07 Oct 2006 | 4S |
| Chaiten (Ch) | 02 May 2008 | 42S |
| Okmok (Ok) | 12 Jul 2008 | 55N |
| Kasatochi (Ka) | 07 Aug 2008 | 55N |
| Fire/Victoria (Vi) | 07 Feb 2009 | 37S |
| Sarychev (Sv) | 12 Jun 2009 | 48N |
| Nabro (Nb) | 13 Jun 2011 | 13N |
| Kelut (Ke) | 13 Feb 2014 | 8S |
| Calbuco (Cb) | 22 April 2015 | 41S |
| Canadian Wildfires (Cw) | August 2017 | 51N |
| Ambae | 27 July 2018 | 15S |

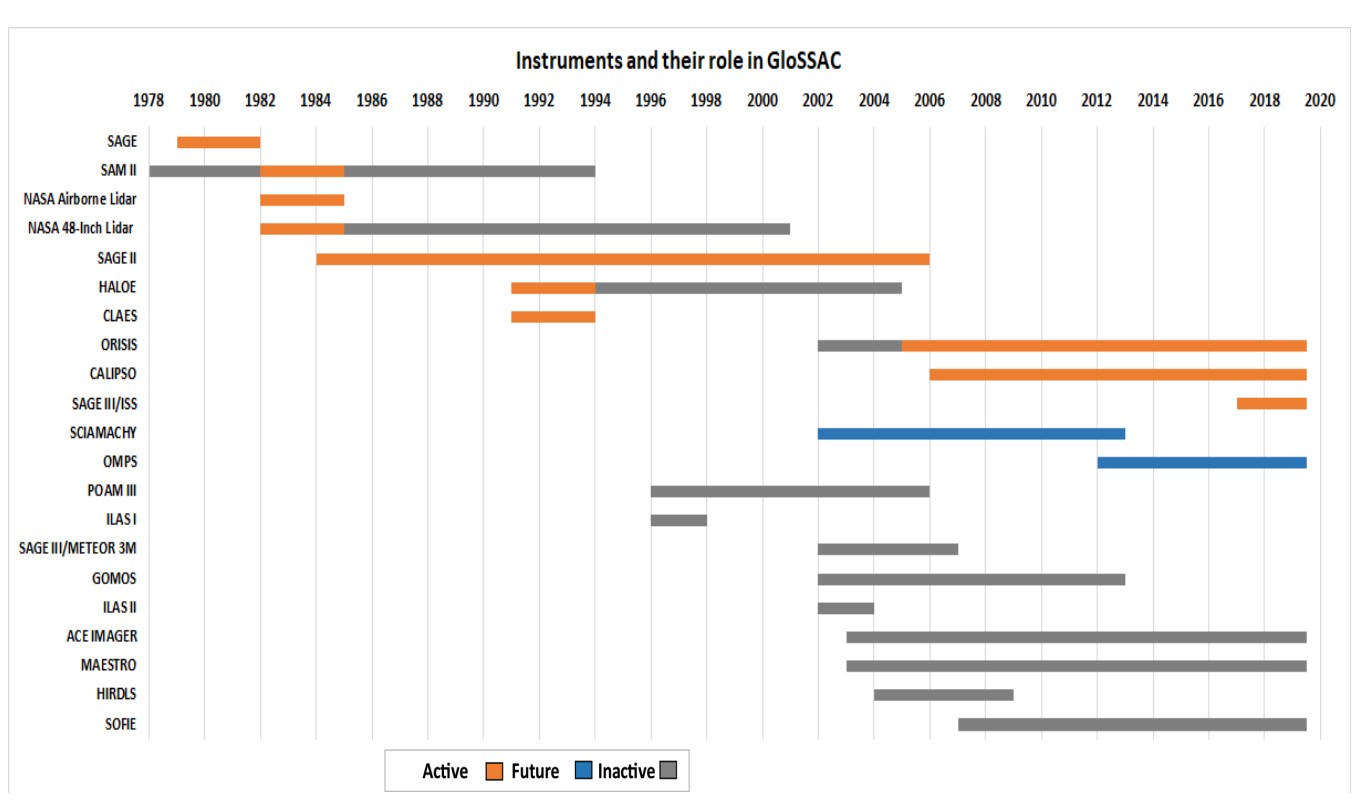

**Figure 1.** Space-based measurements of stratospheric aerosol extinction coefficient data used in GloSSAC.

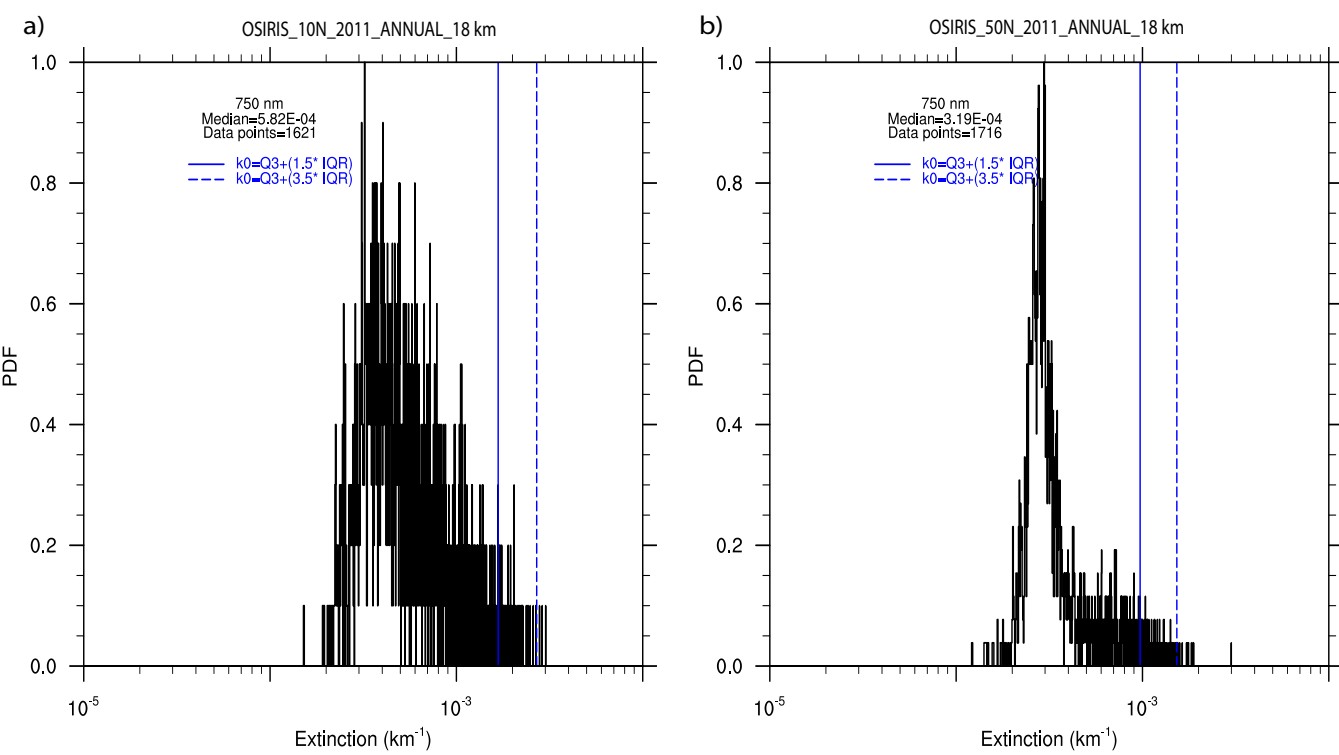

**Figure 2.** Probability density function (PDF) of aerosol extinction at 750 nm. PDF is shown as number of events normalized to the maximum value. The solid blue vertical line represents upper outlier in the data while dashed blue vertical line represents extreme outlier.

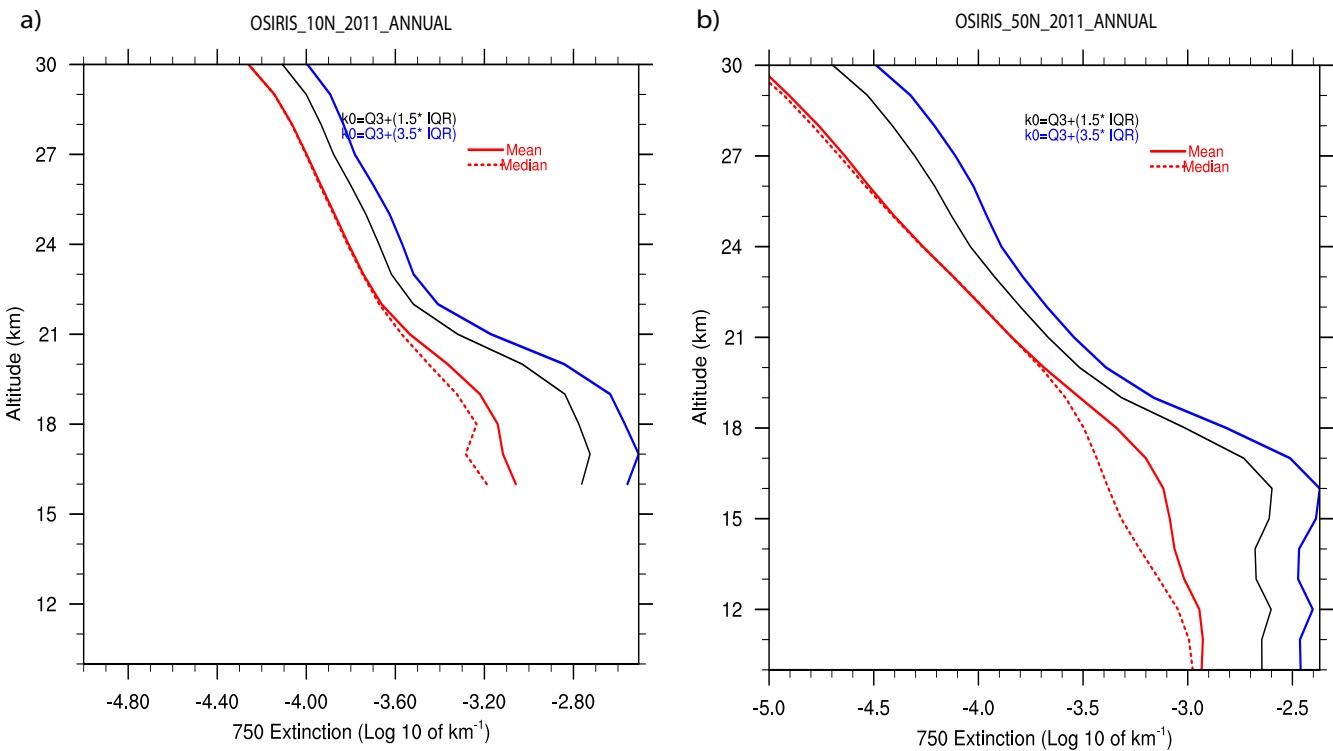

**Figure 3.** Vertical profiles of OSIRIS version 7.0 level 2 extinction coefficient at 750 nm for the separation of aerosol from enhanced aerosol/cloud values along with mean and median of extinction before cloud clearing for (a) 10N for 2011, and (b) same as in (a) but for 50 N.

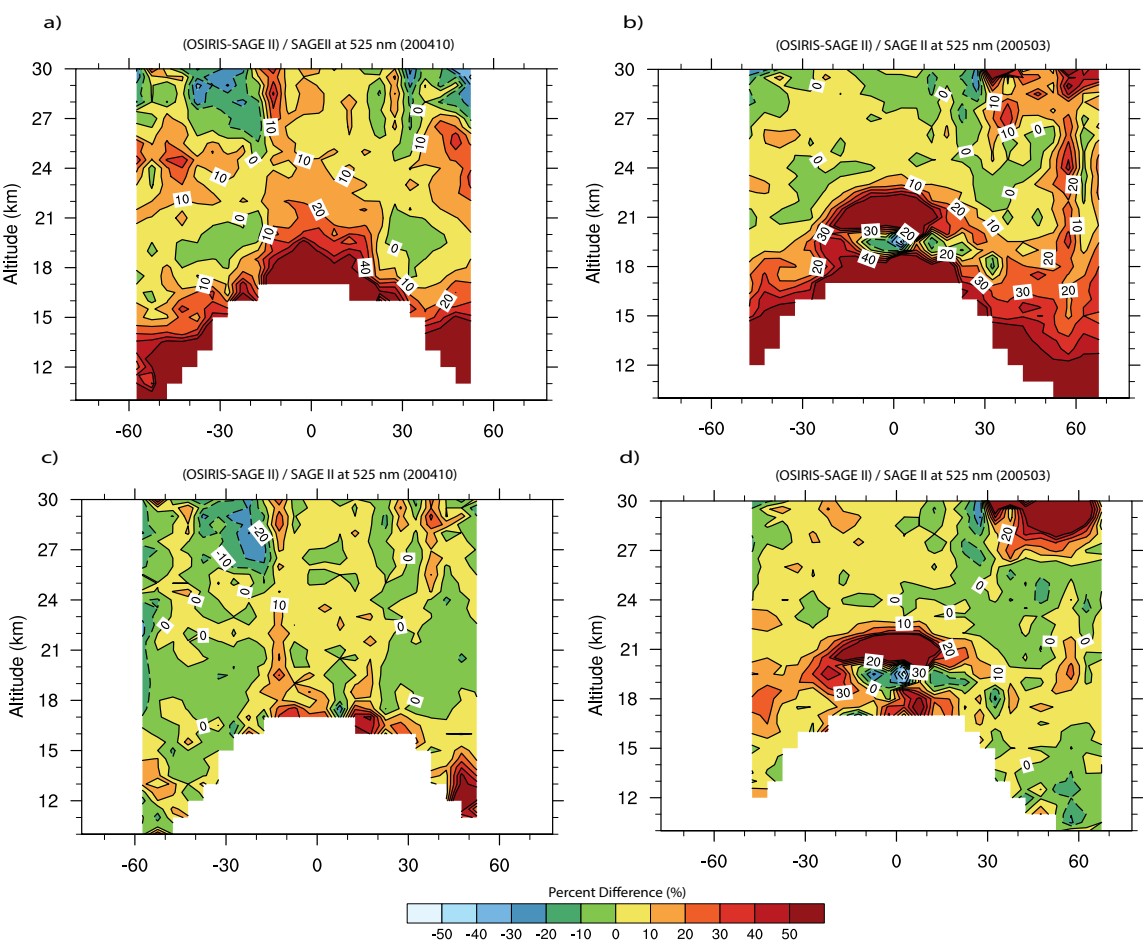

**Figure 4.** Altitude versus latitude of percent difference between OSIRIS and SAGEII extinction coefficient at 525 nm. (a) and (b) show the percent difference plots for 200410 and 200503 respectively for which OSIRIS extinction coefficient at 525 nm is computed using a constant Angstrom exponent of 2.33. (c) and (d) are same as in (a) and (b) but after implementing a monthly based Angstrom exponent to compute OSIRIS extinction at 525 nm.

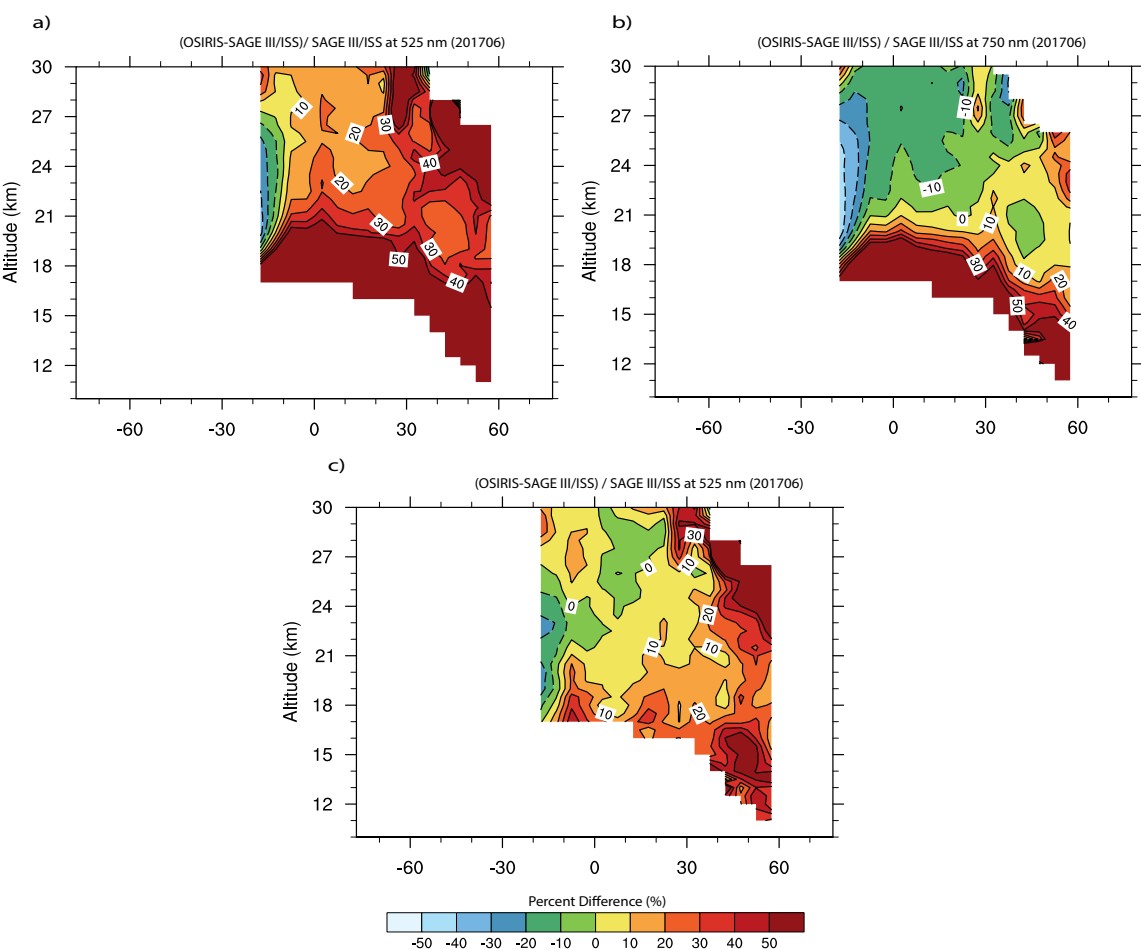

**Figure 5.** Altitude versus latitude of percent difference between OSIRIS and SAGE III/ISS extinction coefficient (a) at 525 nm for 201706, and (b) at 750 nm for 201706. OSIRIS extinction coefficient used in (a) is computed using a constant Angstrom exponent of 2.33, whereas in (c) a monthly based pseudo Angstrom exponent is used to compute OSIRIS extinction at 525 nm.

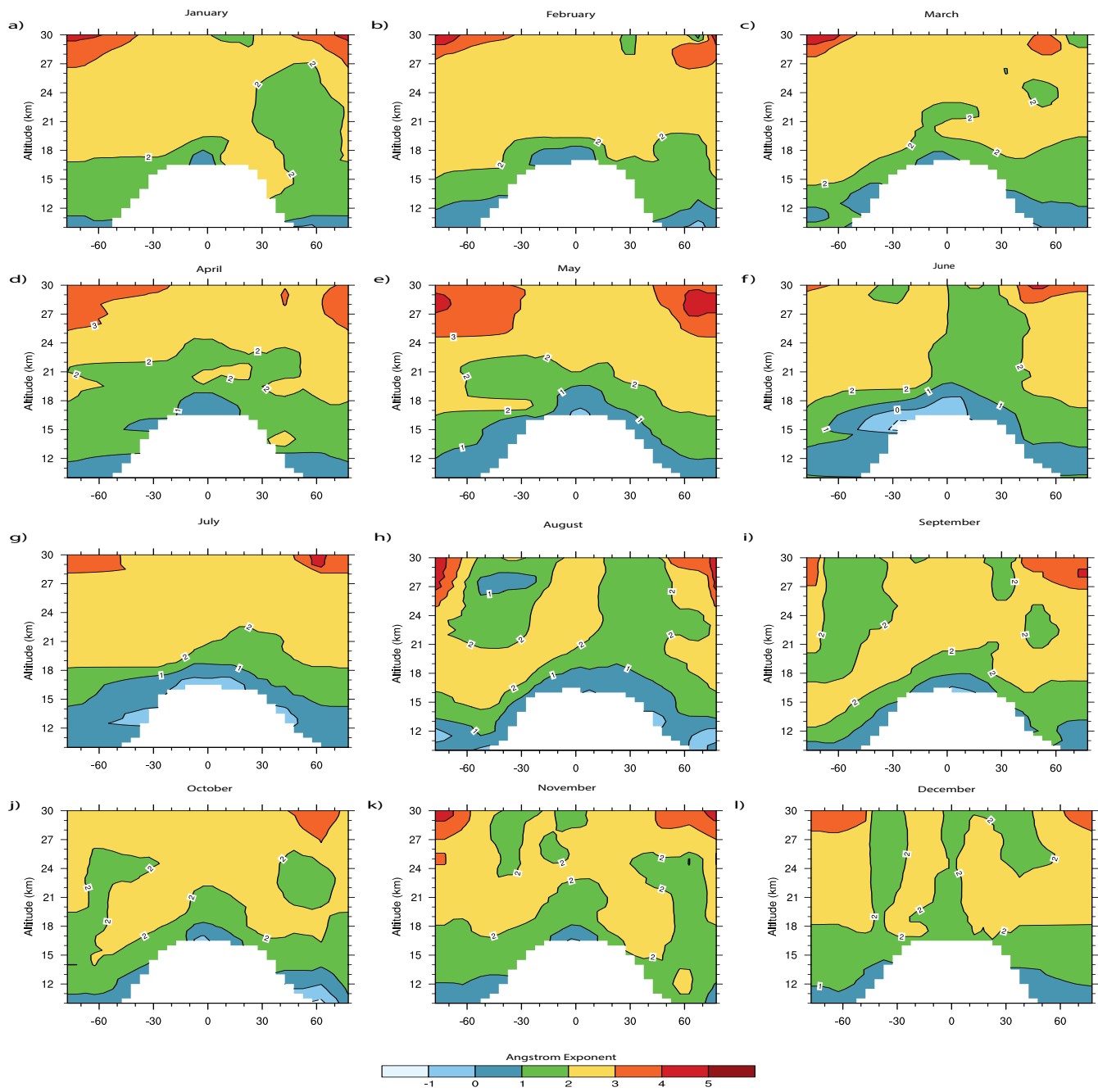

**Figure 6.** Altitude versus Latitude of pseudo Angstrom exponent monthly climatology derived using OSIRIS 750 nm extinction and SAGEII/ SAGE III/ISS 525 nm extinction. A 3X3 median smoothing is used to remove any outliers and then linearly interpolated to fill in any missing data.

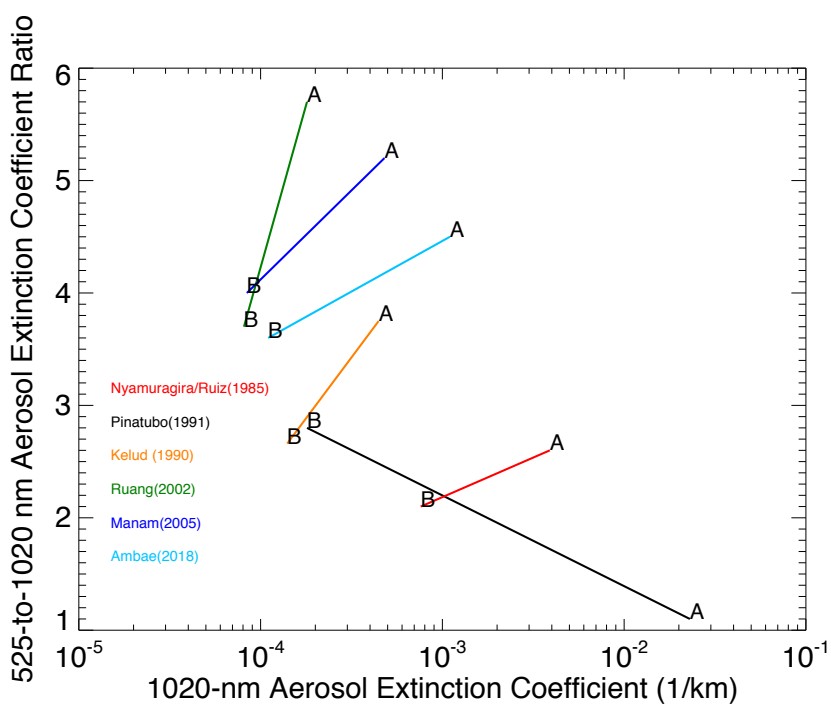

**Figure 7.** 525 to 1020 nm extinction ratio versus 1020 nm extinction ratio for several volcanic eruptions since 1990. The change in aerosol extinction coefficient and extinction ratio for each volcanic event is denoted denoted by two points "B" and "A" which represents change in extinction coefficient and extinction ratio "Before" and "After" eruption.

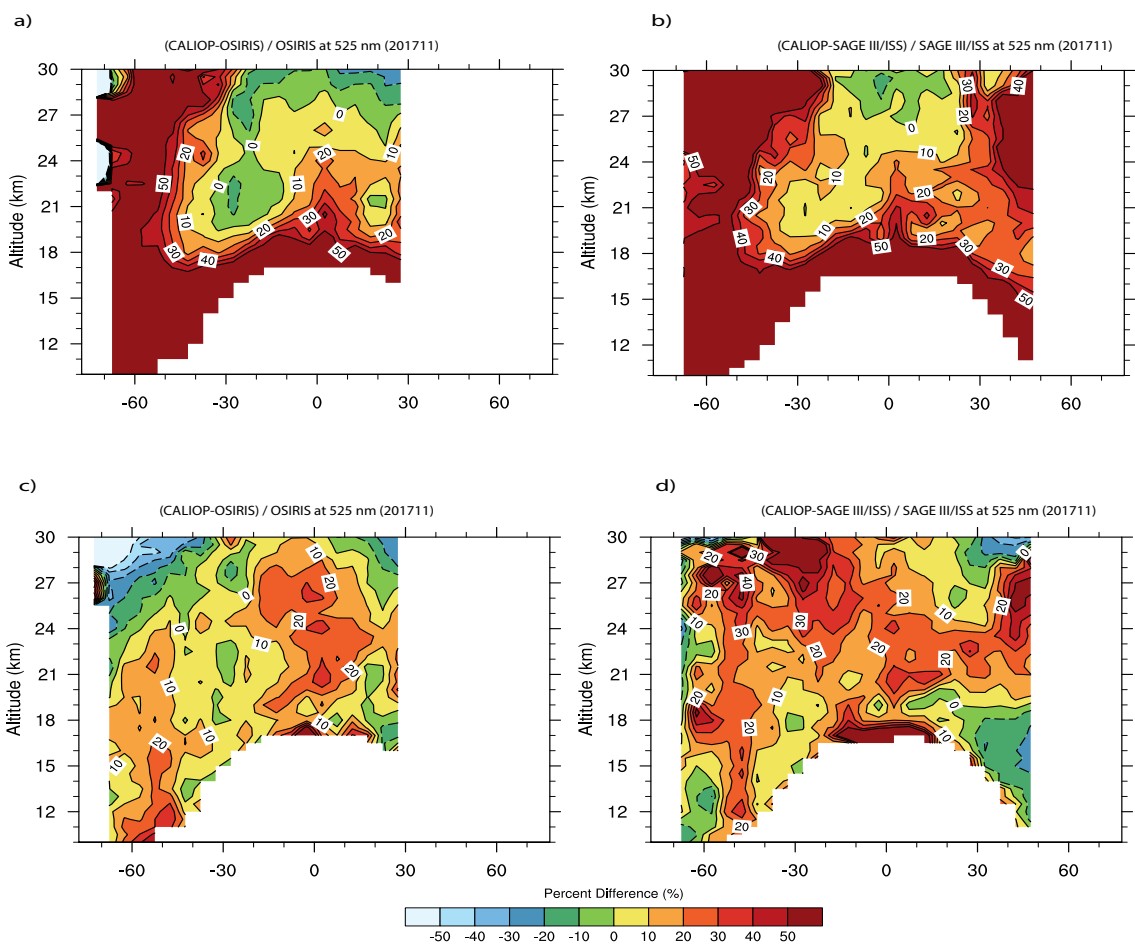

**Figure 8.** Altitude versus latitude of percent difference between CALIOP, OSIRIS and SAGE III/ISS extinction coefficients. Percent difference computed (a) between CALIOP and bias corrected OSIRIS extinction coefficient at 525 nm for 201711, and (b) between CALIOP and SAGE III/ISS extinction coefficient at 525 nm for 201711. (c) and (d) are same as in (a) and (b) but CALIOP extinction is bias corrected.

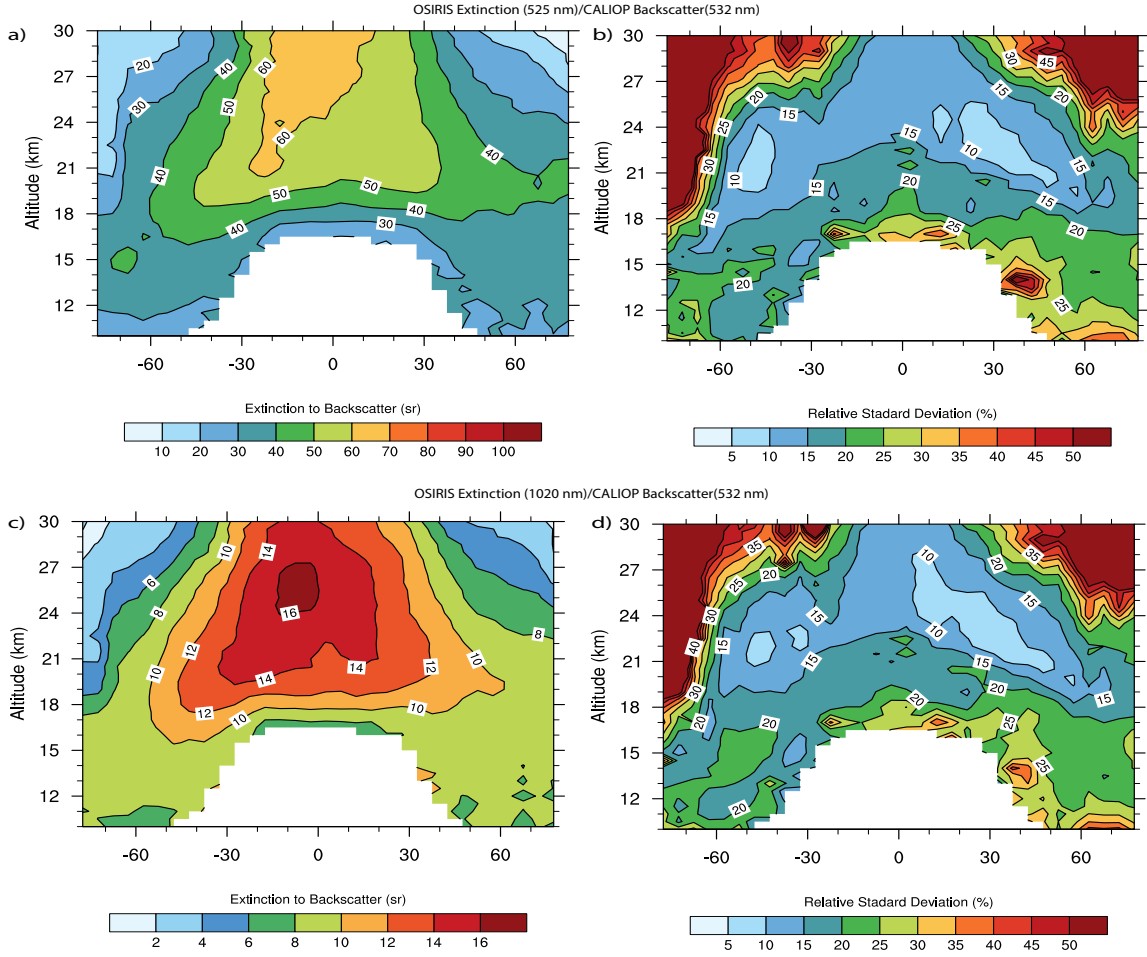

**Figure 9.** Altitude-Latitude dependence of median annual OSIRIS extinction to 532 nm backscatter ratio (OSIRIS/CALIOP) for the overlap period (2006-2017). (a) OSIRIS 525 nm extinction to CALIOP 532 backscatter ratio (SF), (b) Relative standard deviation of (a) is computed at each grid point with respect to the median value in percent. (c) and (d) are same as in (a) and (b) but using OSIRIS 1020 nm extinction.

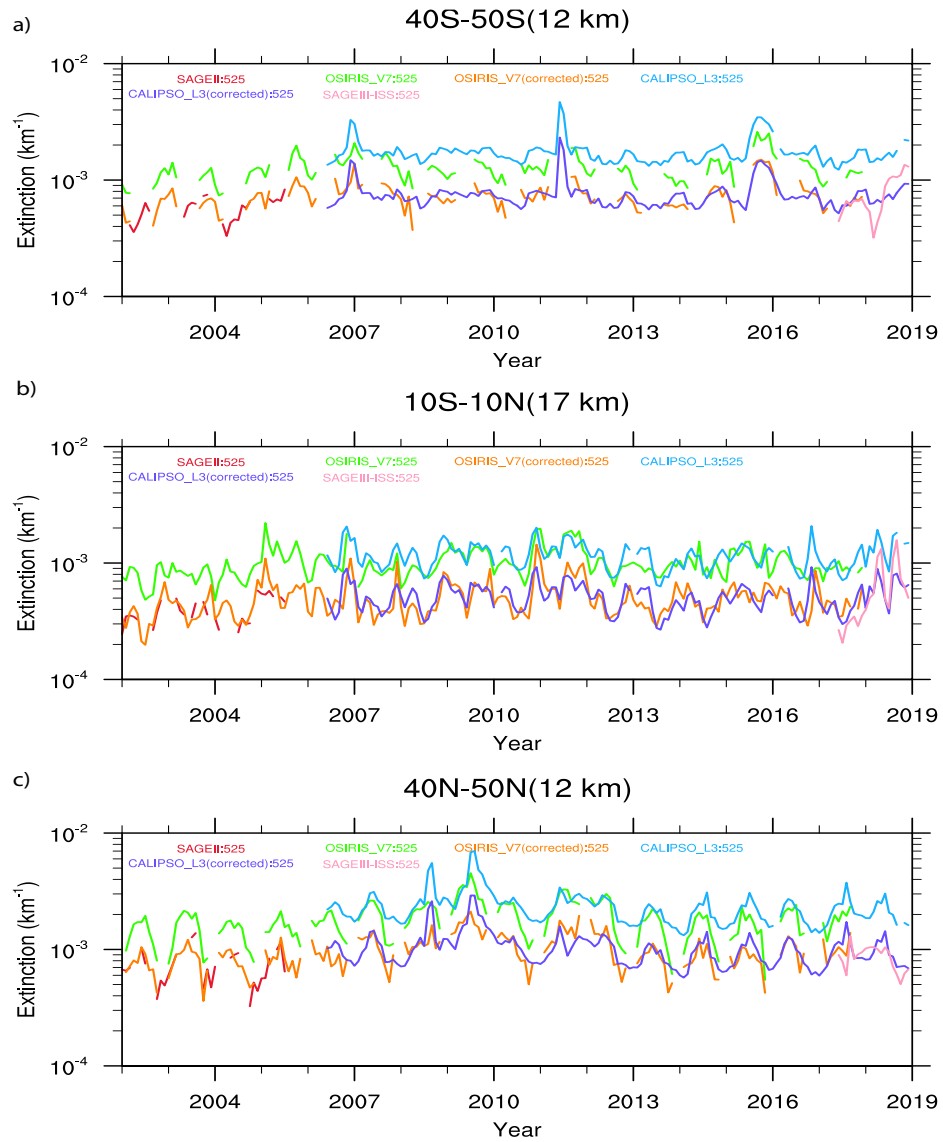

**Figure 10.** Zonally averaged monthly time series plots of extinction coefficient at 525 nm for different latitude bands and altitudes.

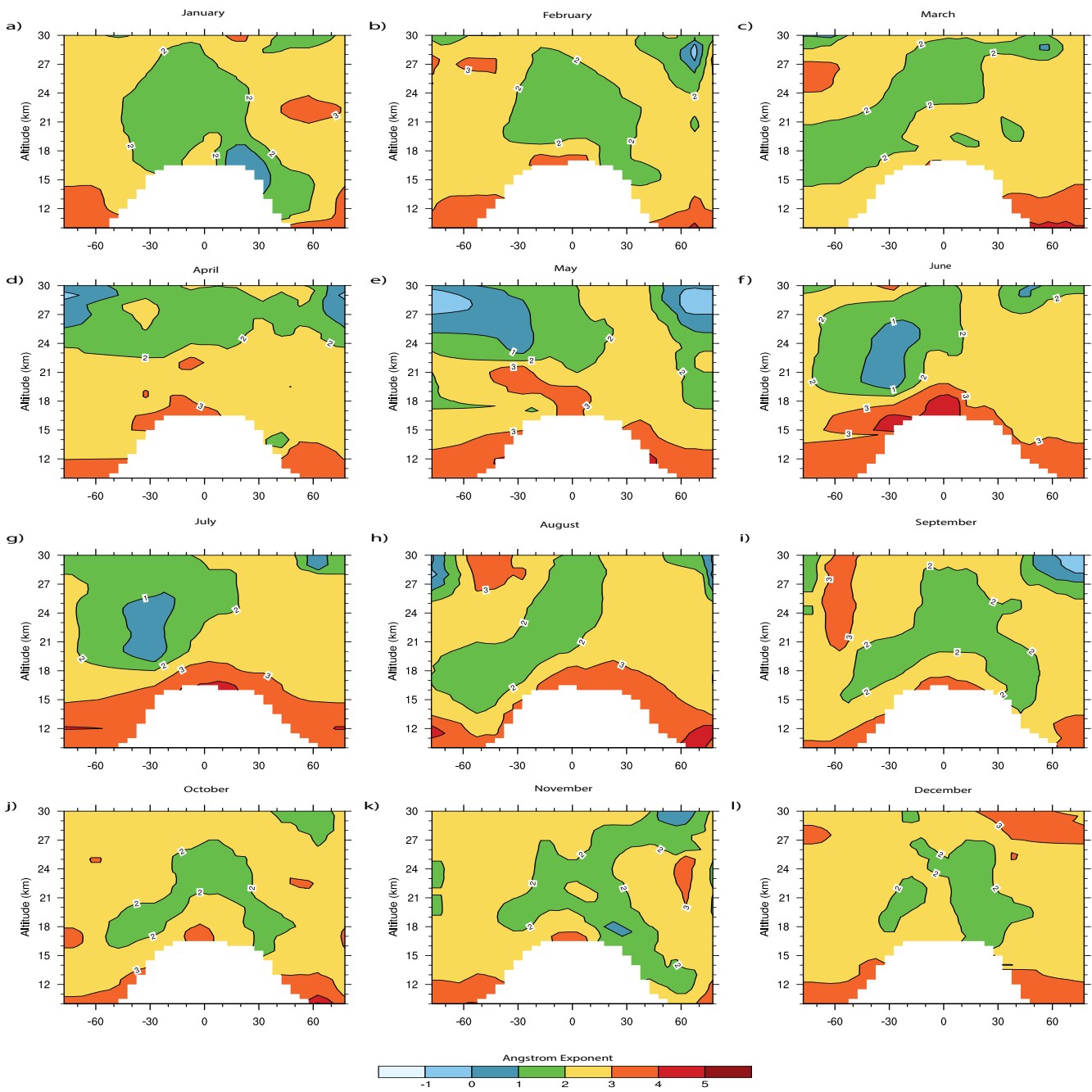

**Figure 11.** Altitude versus Latitude of pseudo Angstrom exponent monthly climatology derived using OSIRIS 750 nm extinction and SAGE II/SAGE III-ISS 1020 nm extinction. A 3X3 median smoothing is used to remove any outliers and then linearly interpolated to fill in any missing data.

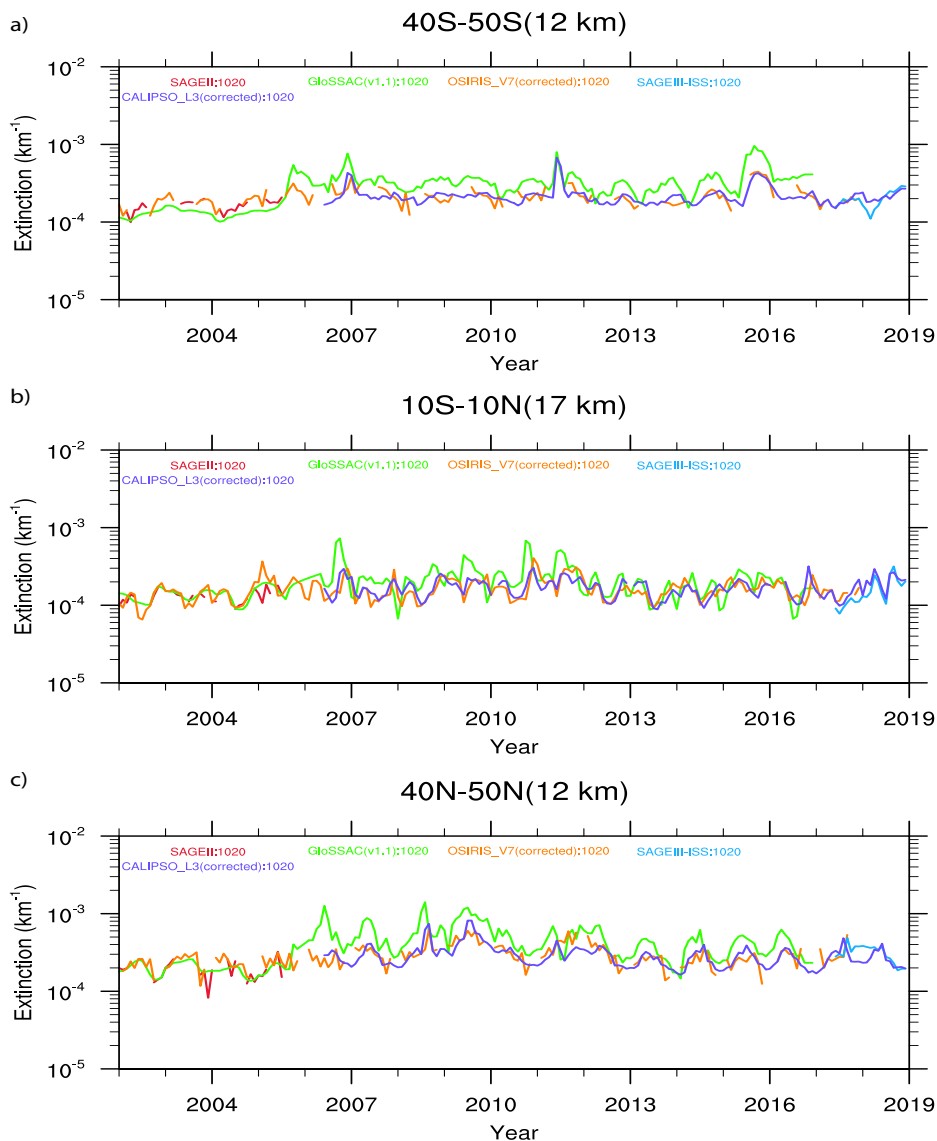

**Figure 12.** Zonally averaged monthly time series plots of extinction coefficient at 1020 nm for different latitude bands and altitudes.

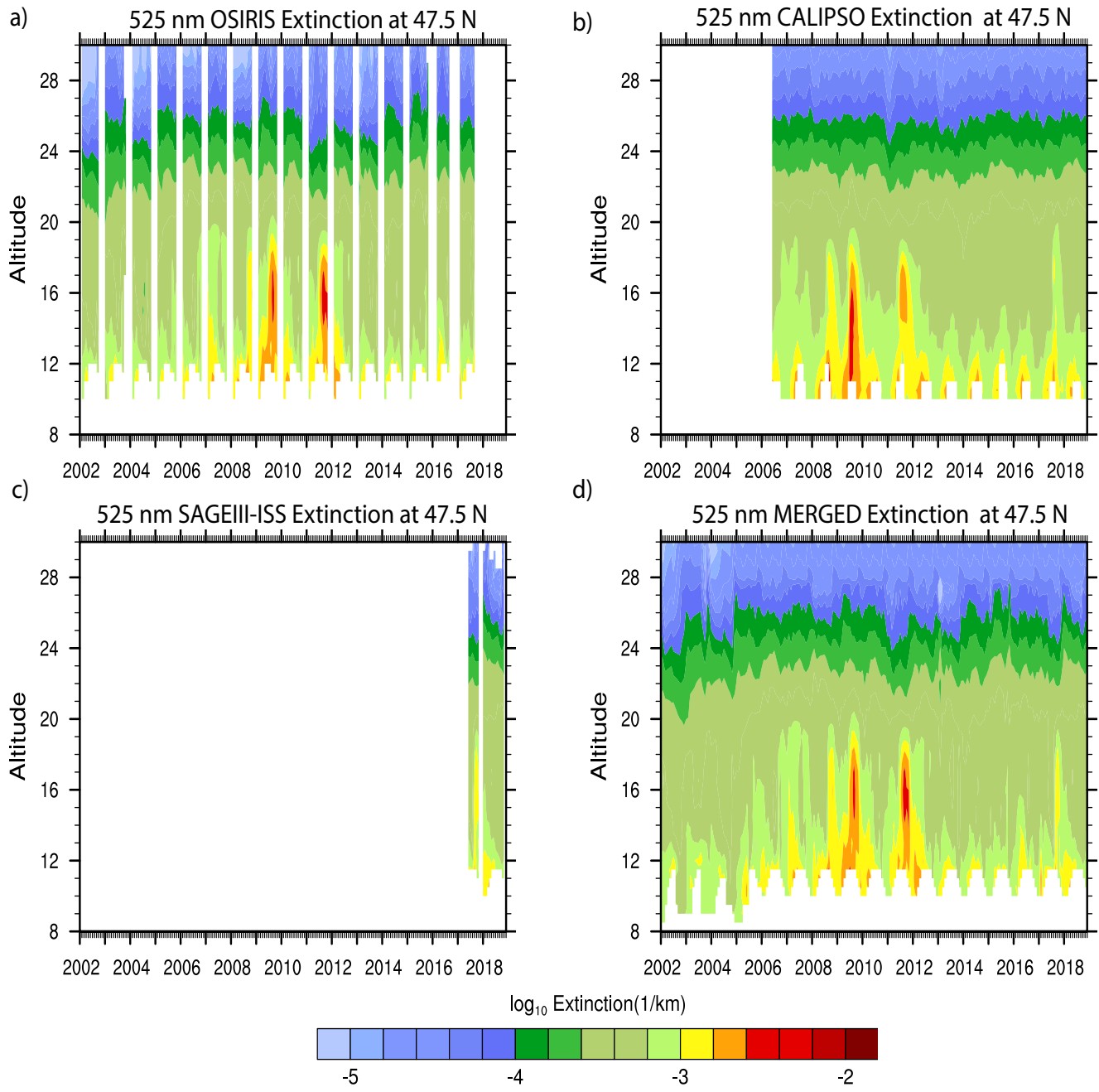

**Figure 13.** Altitude versus monthly time series of 525 nm extinction coefficient at 47.5 N latitude. (a) bias corrected OSIRIS extinction coefficient , (b) bias corrected CALIOP extinction coefficient (c) cloud cleared SAGE III/ISS extinction coefficient and (d) final merged 525 nm GloSSAC v2.0 extinction coefficient.

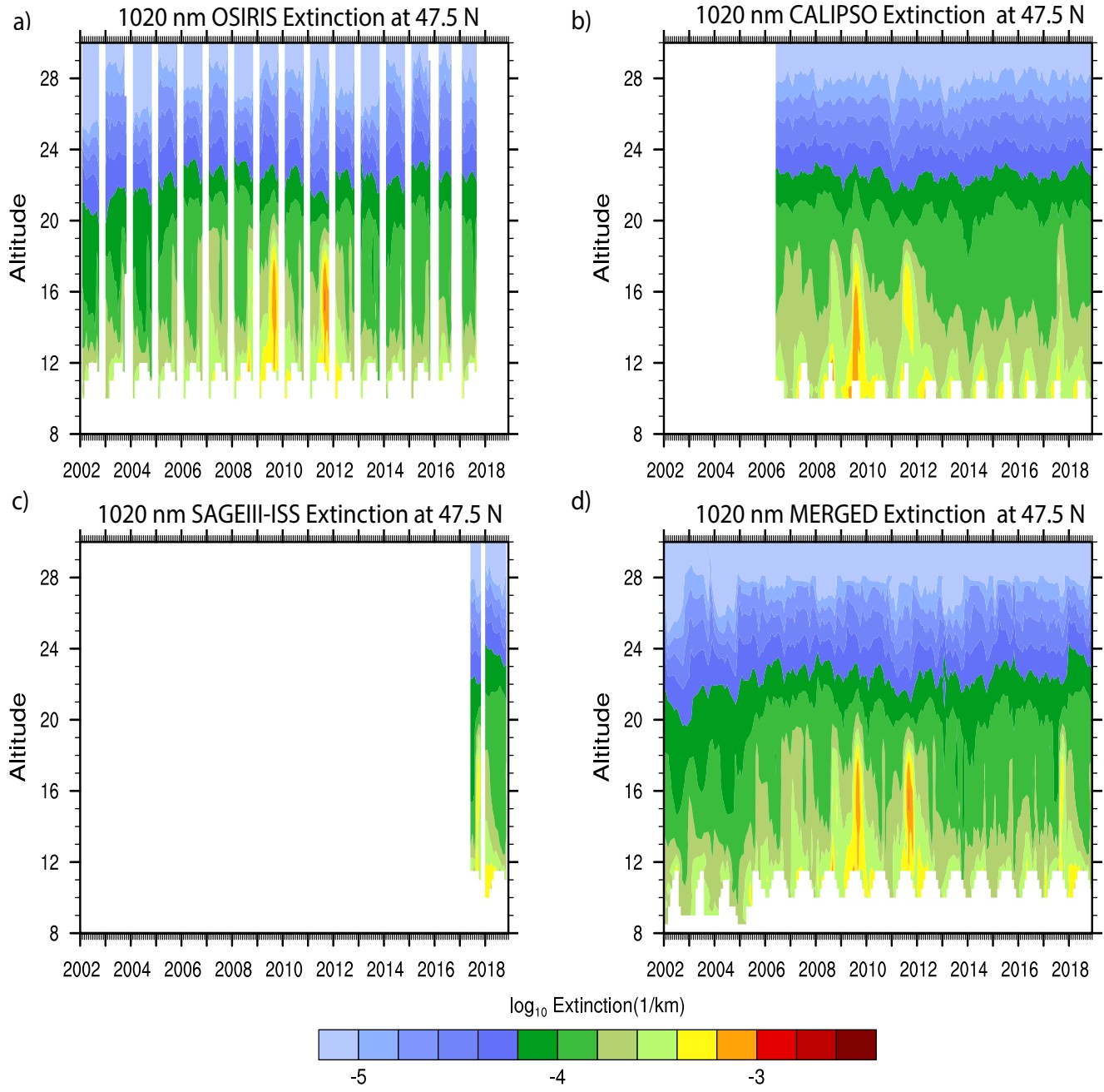

**Figure 14.** Altitude versus monthly time series of 1020 nm extinction coefficient at 47.5 N latitude. (a) bias corrected OSIRIS extinction coefficient , (b) bias corrected CALIOP extinction coefficient (c) cloud cleared SAGE III/ISS extinction coefficient and (d) final merged 1020 nm GloSSAC v2.0 extinction coefficient.

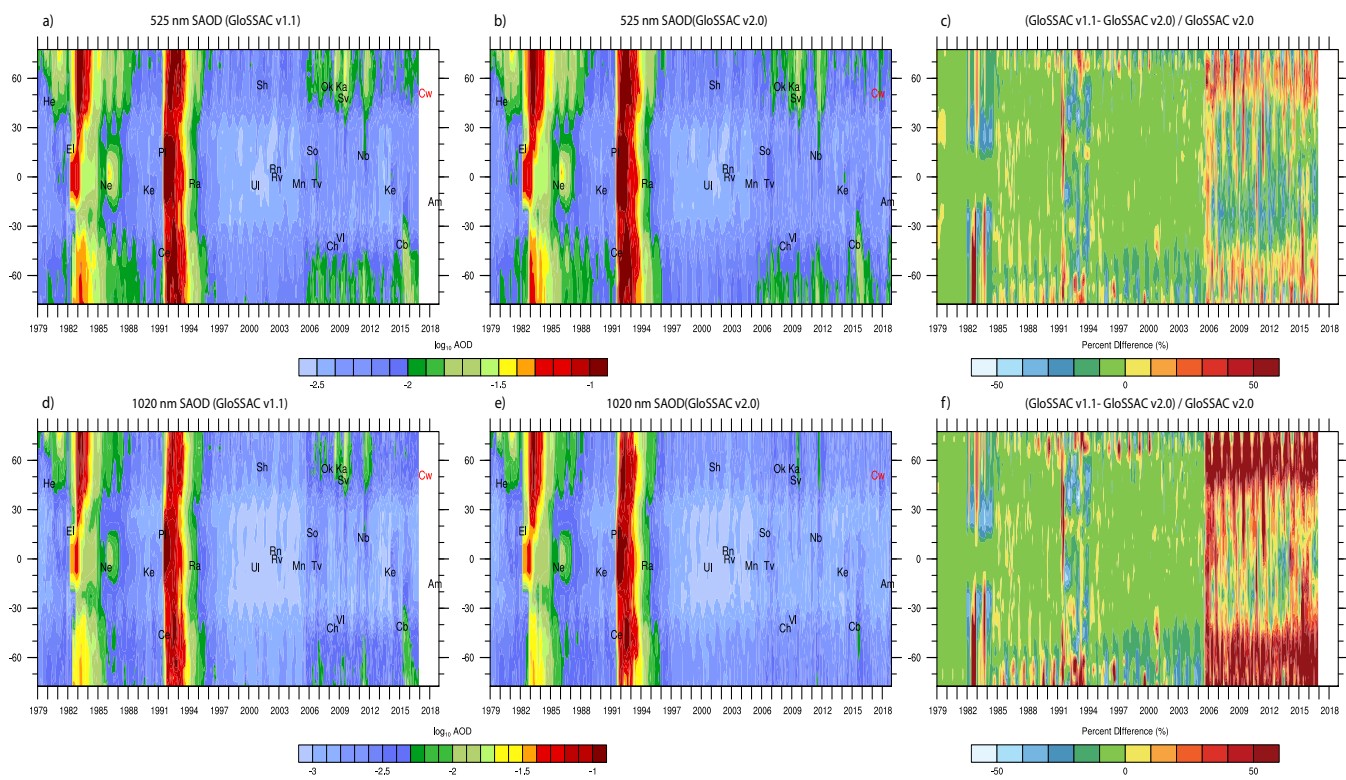

**Figure 15.** Latitude versus monthly time series of SAOD at 525 and 1020 nm: (a) using GloSSAC v1.0 , (b) using merged OSIRIS bias corrected version 7.0, bias corrected CALIOP, and SAGE III/ISS data (GloSSAC 2.0) and (c) percent difference between a and b. (d), (e), and (f) are same as (a), (b), and (c) but for 1020 nm extinction. For 525 nm extinction, OSIRIS data is bias corrected using monthly pseudo Angstrom exponents shown in Figure 5, while CALIOP 525 nm extinction is inferred using OSIRIS 525 nm extinction to CALIOP 532 nm backscatter ratio. For 1020 nm, OSIRIS data is bias corrected using monthly pseudo Angstrom exponents shown in Figure 11, while CALIOP 1020 nm extinction is inferred using OSIRIS 1020 nm extinction to CALIOP 532 nm backscatter ratio.

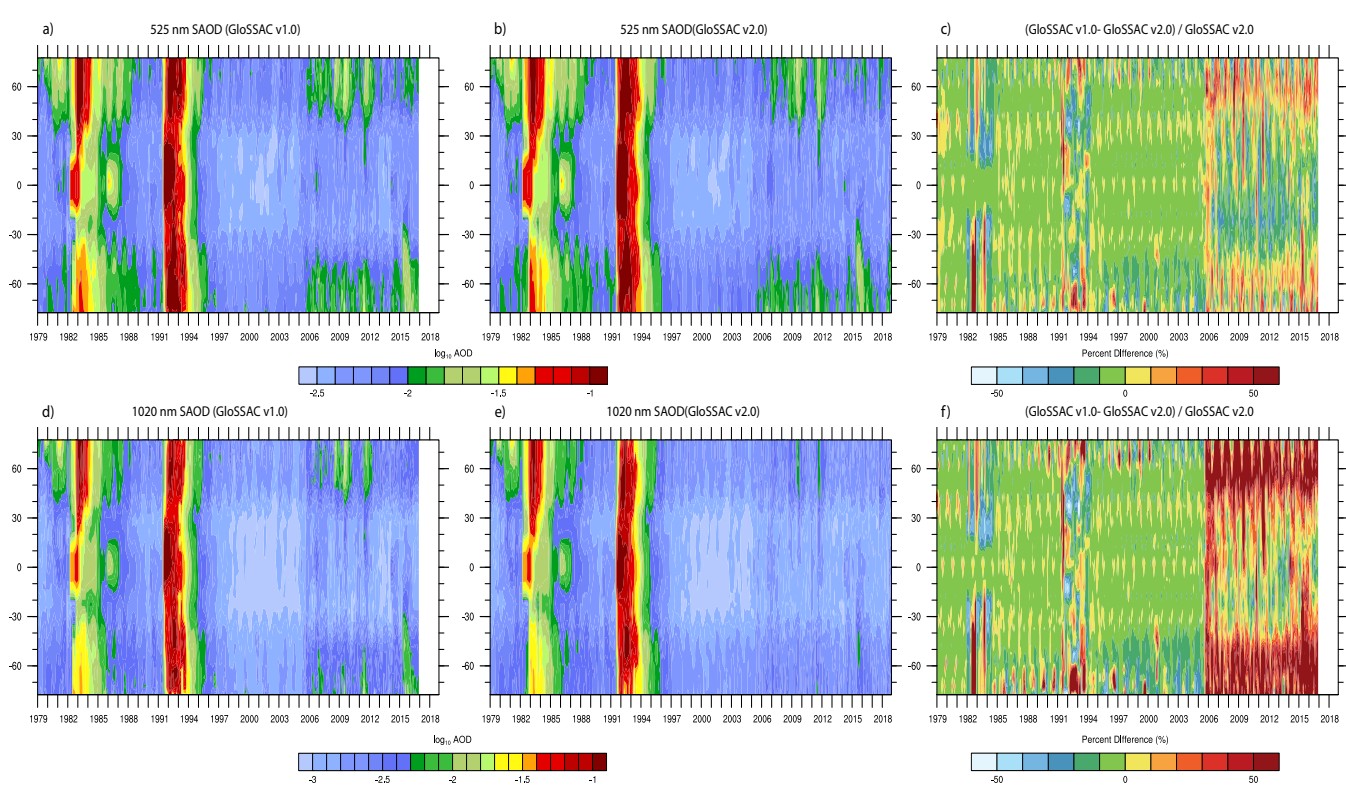

**Figure 16.** Same as in Figure 15 but for GloSSAC v 1.1 is replaced by GloSSAC v 1.0

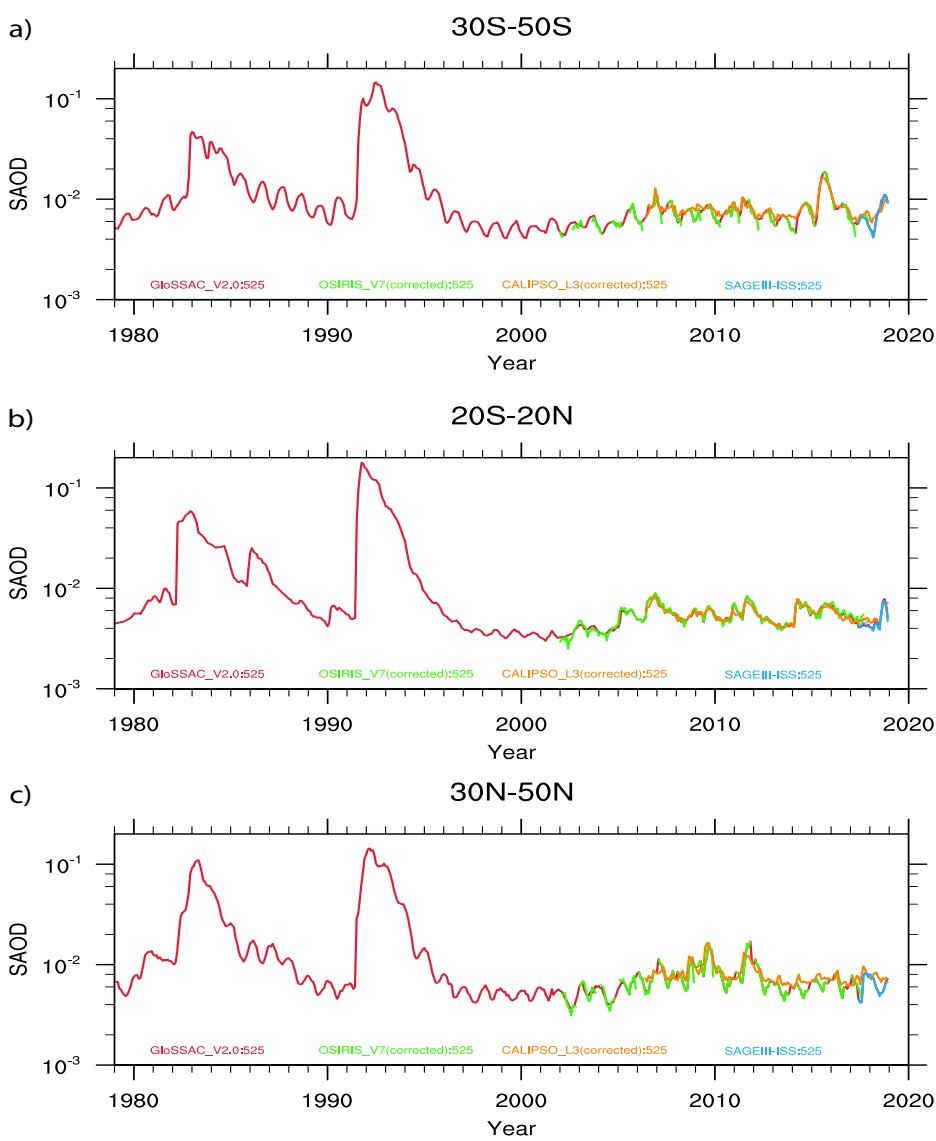

**Figure 17.** Zonally averaged monthly stratospheric aerosol optical depth at 525 nm for (a) 30S-50S , (b) 20S-20N , and (c) 30N-50N .

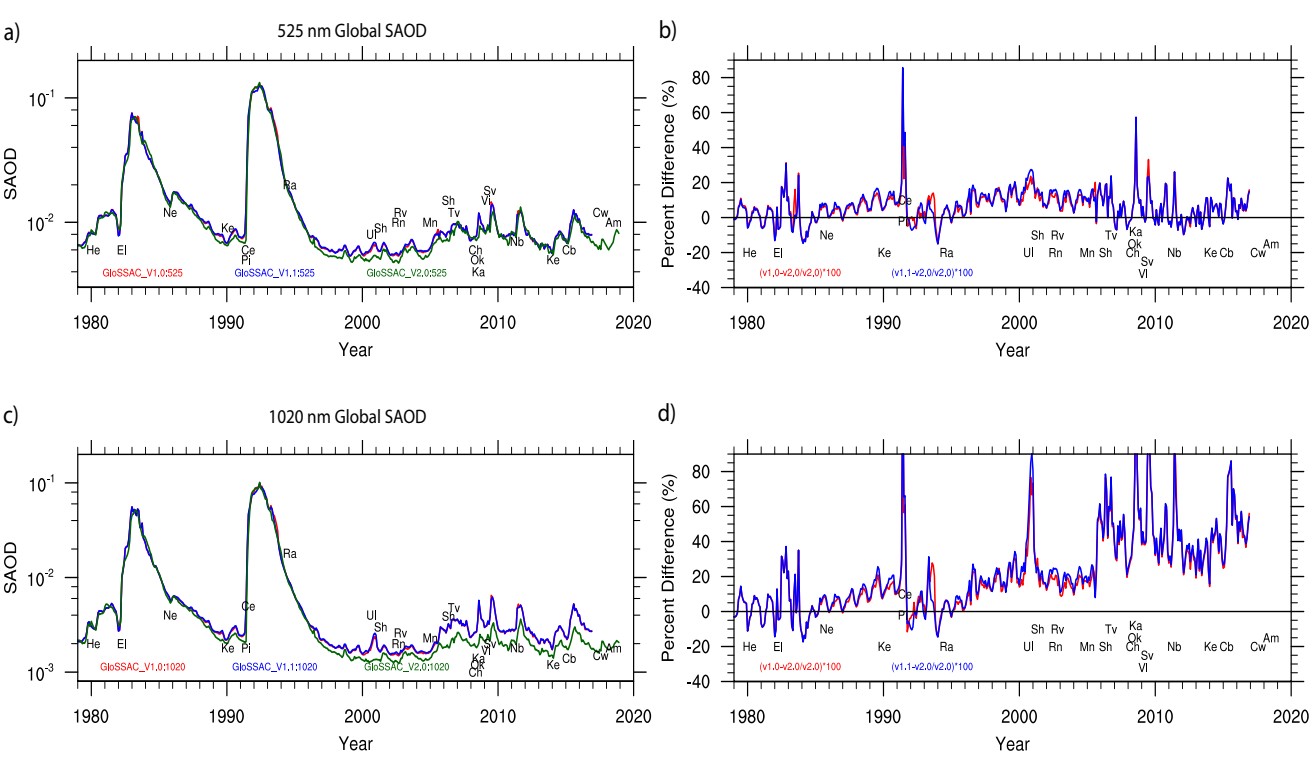

**Figure 18.** Global stratospheric aerosol optical depth at 525 (a) and 1020 nm (c). Percent difference between versions for 525 and 1020 nm extinction are shown in (b) and (c) respectively. Percent difference are computed with respect to v 2.0 as (v1.0 (v1.1)-v2.0/v2.0)*100.

**Table A1.** Acronyms

| | |
|---|---|
| ASAP | Assessment of Stratospheric Aerosol Properties |
| CALIOP | Cloud-Aerosol Lidar with orthogonal Polarization |
| CALIPSO | Cloud-Aerosol Lidar and Infrared Pathfinder Satellite Observation |
| CCM | Chemistry-Climate Model |
| CCMI | Chemistry-Climate Model Intercomparison |
| CLAES | Cryogenic Limb Array Etalon Spectrometer |
| CMIP | Climate Model Intercomparison Project |
| CMIP6 | Coupled Model Intercomparison Project version 6 |
| GCM | Global Climate Model |
| GloSSAC | Global Space-based Stratospheric Aerosol Climatology |
| HALOE | Halogen Occultation Experiment |
| ISS | International Space Station |
| NASA | National Aeronautics and Space Administration |
| NAT | Nitric Acid Trihydrate |
| OSIRIS | Optical Spectrograph and InfraRed Imager System |
| PSC | Polar Stratospheric Cloud |
| SAGE | Stratospheric Aerosol and Gas Experiment |
| SF | Scale Factor |
| SPARC | Stratospheric-tropospheric Processes and their Role in Climate |
| STS | Saturated Ternary Solution |
| SSiRC | Stratospheric Sulfur and its Role in Climate |
| VEI | Volcanic Eruption Index |
| WCRP | World Climate Research Programme |

**Table A2.** List of Variables in the netcdf file

| Variable Name | Description |
| --- | --- |
| Caliop_Backscatter_Coefficient_532 | Estimated CALIOP backscatter coefficient at 532 nm regridded to GloSSAC grid |
| Caliop_Extinction_Coefficient_532 | Standard CALIOP extinction at 532 nm from CALIOP Level 3 file regridded to GloSSAC grid |
| Caliop_Aerosol_Extinction_Coefficient | Conformed CALIOP aerosol extinction coefficient at 525 and 1020 estimated using Scale Factor |
| Caliop_Aerosol_Extinction_Coefficient_Flag | CALIOP Aerosol Extinction Coefficient Flag |
| Caliop_Scale_Factor_Median | Annual median Scale Factor of CALIOP data estimated at 525 and 1020 nm |
| Caliop_Scale_Factor_Standard_Deviation | Standard Deviation of Scale Factor of CALIOP data estimated at 525 and 1020 nm |
| CLAES_Extinction_Coefficient | CLAES extinction coefficient interpolated to SAGE altitude resolution at 7.80 $\mu$ m |
| CLAES_Extinction_Coefficient_Standard_Deviation | Standard Deviation of CLAES extinction coefficient at 7.80 $\mu$ m |
| Glossac_Aerosol_Extinction_Coefficient | GloSSAC extinction coefficient at GloSSAC wavelengths |
| Glossac_Aerosol_Extinction_Coefficient_Flag | Merged Aerosol Extinction Coefficient Flag |
| Glossac_Aerosol_Optical_Depth | Aerosol Optical Depth at GloSSAC wavelengths |
| Glossac_Aerosol_Extinction_Coefficient_Median | Zonal Median of Merged Aerosol Extinction Coefficient at GloSSAC wavelengths |
| Glossac_Aerosol_Extinction_Coefficient_Std | Standard Deviation of Merged Aerosol Extinction Coefficient at GloSSAC wavelengths |
| HALOE_Extinction_Coefficient | HALOE extinction coefficient interpolated to SAGE altitude resolution at 3.40 $\mu$ m |
| HALOE_Extinction_Coefficient_Standard_Deviation | Standard Deviation of HALOE extinction coefficient at 3.40 $\mu$ m |
| High_Altitude_Climatology | Monthly Climatology for the high altitude stratosphere |
| Osiris_Extinction_Coefficient_750 | Zonally averaged 750 nm extinction coefficient interpolated to SAGE altitude resolution |
| Osiris_Median_Extinction_Coefficient_750 | Zonal median 750 nm extinction coefficient interpolated to SAGE altitude resolution |
| Osiris_Standard_Deviation_Extinction_Coefficient_750 | Zonal standard deviation of 750 nm extinction coefficient interpolated to SAGE altitude resolution |
| Osiris_Aerosol_Extinction_Coefficient | Conformed Osiris Aerosol Extinction coefficient at 525 and 1020 nm |
| Osiris_Aerosol_Extinction_Coefficient_Flag | OSIRIS Aerosol Extinction Coefficient Flag |
| Osiris_Aerosol_Extinction_Coefficient_Standard_Deviation | Estimated standard deviation of conformed aerosol extinction coefficient at 525 and 1020 nm |
| Sageiii_ISS_Aerosol_Extinction_Coefficient | Zonal mean SAGE III/ISS Aerosol Extinction coefficient at nine wavelengths |
| Sageiii_ISS_Aerosol_Extinction_Coefficient_Median | Zonal median of SAGE III/ISS Aerosol Extinction Coefficient at nine wavelengths |
| Sageiii_ISS_Aerosol_Extinction_Coefficient_Standard_Deviation | Zonal Standard Deviation of SAGE III/ISS Aerosol Extinction Coefficient at nine wavelengths |
| Sageiii_ISS_Cloud_Cleared_Aerosol_Extinction_Coefficient | SAGE III/ISS Cloud Cleared Aerosol Extinction coefficient at nine wavelengths |
| Sageiii_ISS_Cloud_Cleared_Aerosol_Extinction_Coefficient_Median | Zonal median of SAGE III/ISS Cloud Cleared Aerosol Extinction Coefficient at nine wavelengths |
| Sageiii_ISS_Cloud_Cleared_Aerosol_Extinction_Coefficient_Standard_Deviation | Zonal Standard Deviation of SAGE III/ISS Cloud Cleared Extinction Coefficient at nine wavelengths |
| Sageiii_ISS_Cloud_Cleared_Aerosol_Extinction_Coefficient_Flag | SAGE III/ISS cloud cleared aerosol extinction coefficient flags |
| Stratospheric_Background | Clean period monthly climatology |

**Table A3.** GloSSAC data flag values and description

| Flag Value | Source |
| --- | --- |
| 1 | SAGE II |
| 2 | CLAES empirically scaled to 1020 nm |
| 3 | HALOE empirically scaled to 1020 nm |
| 4 | Equivalent latitude reconstruction |
| 5 | ASAP-based tropical lidar fill data for the Pinatubo period, it is used in part in the June 1991 to September 1991 period |
| 6 | Pinatubo June fix where data from May 1991 is used where no SAGE II observations occur rather than interpolating between very clean May 1991 and very volcanic July 1991 |
| 7 | 525 estimates from valid 1020 nm data |
| 8 | CALIOP backscatter coefficient converted to 525 nm extinction coefficient based on Scale Factor (SF) |
| 9 | OSIRIS bias corrected extinction coefficient based on monthly Angstrom exponent |
| 10 | CALIOP backscatter coefficient converted to 1020 nm extinction based on Scale Factor (SF) |
| 11/12 | Linearly interpolated from points within 2 months. No additional interpolation involving altitude or latitude is included |
| 14 | SAM II/SAGE data from January 1979 through December 1981 |
| 15 | Replicated (same value) downward in Lidar period (1982–1984); mostly only below 10 km and at higher latitudes |
| 16 | 1000 nm SAM II extinction and extinction inferred from airborne and ground-based lidar (January 1982 and October 1984) |
| 20 | High-altitude climatology; average of data between 1984 and 1990 and between 1995 and 2005 |
| 21 | Quality controlled data, values removed and interpolated across. |
| 22 | Some individual holes in otherwise continuous data patched using adjacent grid spots |
| 24 | Estimated 525 nm data where 1020 nm data exists during the Pinatubo period |
| 27 | Linearly interpolated bias corrected OSIRIS extinction coefficient at 525 and 1020 nm |
| 28 | Linearly interpolated bias corrected CALIOP extinction coefficient at 525 and 1020 nm |
| 29 | SAGE III/ISS extinction coefficient |
| 30 | Linearly interpolated SAGE III/ISS extinction coefficient |