# Peer review of "A Global Space-based Stratospheric Aerosol Climatology (Version 2.0): 1979-2018"

_Earth System Science Data, 2020_

## Referee Comment (RC1) · Thomas Aubry (Referee) · 5 Jul 2020

Radiative forcing from stratospheric aerosols is a major driver of climate variability. Building robust and consensual observational records of stratospheric aerosol observations is thus critical to understand past climate changes and to predicting future climate. For the satellite era, this task is made particularly challenging by the fact that different periods are covered by different instruments with different measurement technique, so that producing an homogeneous and consistent continuous record is difficult. To address this challenges, the Global Space-based Stratospheric Aerosol Climatology (GloSSAC) dataset v1.0 was produced (Thomason et al., ESSD 2018) and has since become the reference dataset of stratospheric aerosol observations. In particular, it has been used as input to historical experiments of Phase 6 of the Climate Model

[Figure]

Intercomparison Project.

This manuscript presents the latest version v2.0 of the GloSSAC dataset which follows the original version v1.0 and its update v1.1. Many improvements have been done since version v1.0 and in particular: i) the dataset is now extended to year 2018; ii) an error in the processing of data from the CLAES instrument has been corrected; iii) the processing of data from the OSIRIS and CALIPSO instrument has been improved, in particular thanks to data from SAGE III having become recently available and overlapping with the OSIRIS/CALIPSO observations. Given these major improvements and the importance of the GloSSAC dataset, this manuscript will be an extremely valuable contribution and I recommend its publication following minor revisions.

The manuscript describes and justifies in great details the updates to GloSSAC, and generally reads well. There are a few places where I think the figures or text could be improved (cf minor comments below). My main comment is that even though technical changes since v1.0 are well described, the paper lacks figure(s) and discussion extensively comparing v2.0 to v1.1 and v1.0 (cf specific comment below), which seems important given this paper provides an update to an existing dataset. I think this would represent an important improvement to the paper, and that it wouldn't require much work from the authors which is why I recommend minor revisions.

Specific comments

1) If I'm not mistaken Figure 15 is the only figure showing the final differences between two versions of the GloSSAC (v2.0 and v1.1) and only do so for the 2002-2016 period. I don't think GloSSAC updates have been documented in a peer-reviewed literature since GloSSAC 1.0 release (Thomason et al. 2018)? It thus seems very important to show the differences between all three versions (1.0, 1.1 and 2.0) and for the full period in common (1979-2016). From my own analyzes of GloSSAC version it seems that each version is different in the post-Pinatubo period, which is one of the period with the most research on stratospheric aerosol forcing. I thus really think that figure

15 should be extended to include all versions/the full common period. In addition to showing contour plot of SAOD as a function of latitude and time, I also think it would be very valuable to compare global mean SAOD time series between the three versions as this is the canonical metric for stratospheric aerosol impact on climate. Section 5 of the manuscript should then be extended to discuss these differences in greater details. Getting an idea of SAOD differences among GloSSAC versions will likely be a major expectation of GloSSAC users from this paper, so I strongly encourage the authors to address this comment.

2) In line with comment 1 above, I think that the abstract should end with a few sentences summarizing the main changes between versions in terms of SAOD. The abstract is very focused on the technical changes in GloSSAC 2.0 which is of course appropriate for an ESSD paper, but it is currently hard for a scientist with little expertise in remote sensing to get a sense of the impacts of these changes on the GloSSAC product from the current abstract.

3) This comment is very much a suggestion. Figure 1 is an excellent introduction figure to the manuscript. I was wondering if it could be complemented (or if you could add a new figure) showing a timeline of some of the main features/limitations/challenges in the GloSSAC record, such as what type of instrument is used (e.g. solar occultation or other), the resolution/frequency of measurements (e.g. global daily coverage with OSIRIS/CALIOP vs global monthly coverage with SAGE instruments), assumptions required (e.g. periods in which an assumption on size distribution is required), etc... Such a figure would enable people with limited expertise in remote sensing to understand in one glance some of the main features of the GloSSAC dataset before using it, which I believe would be very valuable.

Minor comments

page 1 line 1: I stumble a bit on the first sentence of the abstract. In addition to being a bit cumbersome, it introduces what a stratospheric aerosol dataset should do, but the

second sentence does not follow on the dataset so it's confusing.

page 1 line 7: I don't think Zanchettin et al. (2016) is the adequate reference for CMIP6 unless you are talking specifically about VolMIP.

page 1 line 14: reformulate "noted in v1.0 as noted in"

Page 2 line 4-5: "can impact climate on scales from the subtle [...] to the more profound [...]": I find this wording vague and confusing: are you talking about the timescales of the impacts? Their magnitude?

page 2 line 7-9: I would be more specific and clear about the difference in these modelling approaches, e.g. "Some of these modelling studies directly use observations of stratospheric aerosol optical properties as input, whereas other use observations of SO2 as input and interactively simulate stratospheric aerosol life cycle"

page 2, line 6-14: You focus on GCMs study as a motivation but I feel like you could include other examples that have used the GloSSAC dataset to make important contributions. In a purely observational study, Stocker et al (2019, https://agupubs.onlinelibrary.wiley.com/doi/full/10.1029/2019GL084396) quantify the temperature footprint of 21st century eruptions using GloSSAC which in turns enable to better quantify temperature trends related to anthropogenic forcing. Aubry et al. (2020, https://agupubs.onlinelibrary.wiley.com/doi/full/10.1029/2019JD031303) used GloSSAC to calibrate a box model of volcanic forcing. Such box model is the typical tool used to derive SAOD/forcing time series from ice-core records so that's an important application of dataset like GloSSAC. These are just suggestions and application to GCM study is of course a major motivation, but I think it's nice to highlight that applications of the GloSSAC dataset go beyond that.

page 2, line 14: I am not too sure where to put this comment, but it feels like the paper under review by Rieger and co-authors (https://gmd.copernicus.org/preprints/gmd-2019-381/) is very relevant to your paper and should be mentioned if its publication

status allows it. Maybe instead of in the introduction it would fit better in your discussion of an extended figure 15 (cf specific comments) showing the differences in SAOD between v1.0, v1.1 and v2.0 for 1979-2016.

page 2, line 15: it's a bit confusing as you say "mostly unchanged" followed by "significant improvements" and "major version change". These statements sound contradicting and you may want to reformulate.

page 3, line 13-33: this feels like a very detailed and technical discussion of the changes you made for an introduction. I feel like this content should be in section 2 instead?

page 4, line 7: there and hereafter, I suggest you provide date in parenthesis when you refer to XXX instrument period. It will be very helpful for readers not perfectly familiar with the period spanned by different instrument.

page 4, lines 10-18: if I'm not mistaken no figure illustrate these results? The Pinatubo period is of course of utmost interest to climate modelers so it feels like there should be a figure accompanying this paragraph? (although if you extend Figure 15 according to my suggestions that would illustrate this paragraph well)

page 5, line 14-16: this is an important comment. Again it would be nice to specify dates in parenthesis for user who are not familiar with SAGE missions dates. I know that the reader could just look at Figure 1 but it would facilitate the reading if you also provide such dates directly.

General comment on section 2: I enjoyed this section and although I don't have the expertise to understand all details, you clearly highlight the differences in methods/limitation/challenges of different periods of the GloSSAC record. A figure with a timeline showing this features would be a very neat addition (see specific comment # 3 for a more detailed suggestion)

page 6 line 5-6: "another potential source of bias is repeated twice in the same sentence; improve wording

page 6 line 7: maybe you could give an idea of the uncertainty on the 50 sr value?

page 6, lines 25-34: Is this cloud-clearing method more challenging to apply when there is a very large volcanic eruption (e.g. Pinatubo like or larger)? I'm just wondering whether the IQR would be larger following a large eruption.

Page 8 line 4-5: does the Raikoke 2019 eruption provide a good test for this hypothesis?

Page 9 line 2-3: why do you say "though probably not at 756nm"? Doesn't figure 5b show a strong high bias in the lower stratosphere and low bias in the tropical mid-stratosphere?

Page 9 line 14: I can't find the definition of lambda

section 3-4: these sections were generally clear and provide a good overview of differences between instruments and data processing/conforming procedures employed by the authors.

page 13 line 33: So I guess the tropopause height is a climatology as in Thomason et al. (2018)? I think it would be useful to remind here the period used to derive this climatology, as well as the reanalysis used (MERRA if I remember correctly). Additionally, the tropopause height is quite variable at high-latitude and is increasing in the tropics as a consequence of anthropogenic forcing. Given these two points, I am wondering why you are using a climatology instead of the reanalysis data directly? Differences would likely be small but it would be a bit more rigorous approach? As an example figure S1 in Aubry et al (2020, https://agupubs.onlinelibrary.wiley.com/doi/full/10.1029/2019JD031303) show GloSSAC SAOD at v1.0 and v1.1 using the MERRA climatology and the NCEP-NCAR reanalysis for tropopause height with some interesting differences. I haven't analyzed these differences further, but the SAOD I get are smaller for the 21st century which

could be in part due to tropopause height increase?

Section 5: please see my specific comment #1, but I really believe that this section would be more complete if you: -show differences between all 3 versions of GloSSAC -show differences for the full time period shared between the 3 version (1979-2016) -show global mean SAOD time series in addition to SAOD contour plot -extend text in section 5 to include discussion of the above

page 14 lines 22-23: it feels like you could add a few references to support this statement? There have been multiple modelling studies showing that post-2005 SAOD enhancement can be largely explained by SO2 emissions from explosive volcanic eruptions as well as wildfire for some of the recent years. See e.g. Schmidt et al. (2018, https://doi.org/10.1029/2018JD028776), Peterson et al. (2018, https://www.nature.com/articles/s41612-018-0039-3) or Aubry et al. (2020, https://agupubs.onlinelibrary.wiley.com/doi/full/10.1029/2019JD031303).

page 16 line 9: avoid repetition of "inferred 1020nm extinction" twice in the same sentence. Otherwise, I think this is a nice paragraph to close the conclusion section!

Section 6: the conclusions are ok but overall I feel like you could be a bit more succinct on some of the technical details, and that you should add a few sentence commenting on major differences in SAOD in GloSSAC 2.0 compared to 1.0/1.1. This really seems critical as the aim of the paper is to present the newest version of the GloSSAC dataset, so in general it really feels like you should do more to compare SAOD in the different versions. This would likely be the most expected results from this paper for users of the GloSSAC product.

Table 1 caption: replace "since 2002" by "over 2002-2018" as this table doesn't include 2019/2020 eruptions (e.g. Raikoke and Ulawun in 2019, Taal in 2020)

Figure 2: I initially got confused by how noisy these pdfs are. Even though the figure looks fine after consideration, it may be clearer if you use larger bins?

Figure 7: I like this figure a lot. Maybe you could have a SI table specifying the dates of "after"/"before" (or add these dates in the caption directly)

Figures 10/12/17: given the differences between datasets are relatively small, I'm wondering if you should not use lines instead of markers? The marker sometime overlaps a lot making it harder to distinguish any systematic difference.

Figures 13-16: on most of these figures, the density of contour labels is too high and prevent the reader to see clearly the data on the figure (this is made worse by the white rectangles in which each label is inserted). Consider removing the labels altogether or at least reducing their density/removing white rectangles.

Figure 16: it looks like this figure has been stretched horizontally?

---

## Referee Comment (RC2) · Anonymous Referee #2 · 6 Jul 2020

General Comments

The paper presents the new version of the GLoSSAC aerosol climatology (version 2), and details all changes brought with respect to the former version 1.1, for wich changes are also briefly described with respect to version 1.0. Prominent changes are the availability of a new version for OSIRIS (version 7.0) with an improved quality, and th release of a standard CALIOP extinction product. Beyonf an improvement of the overall quality, this brings new possibilities to refine the derivation of some GLoSSAC products (e.g. through the use of variable extinction-to-backscatter ratio).

Overall, the paper is clear, well written and well structured, in particular the introduction and conclusions.

In many places, citation of the first GloSSAC paper (Thomason et al., 2018) is required. I suggest to specify the section to which the citation refers in this paper, in order to ease the reading.

A recurrent assuymption is that the SAGE instrument SAGE II and SAGE III/ISS are golden standards, and the key benchmark by which all other data sets have to conform. A very good reason for this is that SAGE instruments are using solar occultation, a technique requiring few assumptions for the data retrieval. SAGE II also has an excellent reputation and was a very long-duration mission. However, doing so ignores the possibility that SAGE II ageing affects the quality of the measurements at the end of the SAGE II, although the use made of SAGE II to calibrate OSIRIS and CALIOP is of critical importance for GLoSSAC. It should be reminded that SAGE II is about 18-years old when OSIRIS and CALIPSO are launched. On the other hand, SAGE III/ISS is recent, and one could miss the broader view on the real quality of this data set. This point of view should also be discussed or at least mentioned, with reference to validation papers giving more insight into the quality of the dataset during the critical period overlapping with the OSIRIS and CALIPSO missions.

Overall, the paradigm is that everything is fine tuned to match the two SAGE datasets (and OSIRIS where these datasets are unavailable), but sometimes at all costs, without too much consideration for the consistency or physical significance of the methodology (e.g. different Angström exponents used for OSIRIS and CALIOP conversion purposes, "We do not assume that the derived Angstrom coefficient has any physical meaning (. . .) it is simply a mean to push OSIRIS (. . .) toward SAGE II", L. 22-24, p.9). The fact that instruments (SAGE, OSIRIS, CALIOP) are based on totally different measuring techniques that might have an impact in some altitude or latitude range is hardly considered or discussed, although this might provide an insight into main differences between the data sets. The way CALIOP backscatter coefficient is conformed to GLoSSAC extinction coefficient is also not fully convincing. Automatically considering a hierarchy of values ("SAGE II is the best instrument", "SAGE II/ISS is equal to SAGE

II", "OSIRIS in the best one after SAGE") without questioning the physics, the evolving atmospheric state, or any consideration related to aging instruments, has the consequence that SAGE II's spectre is still hovering on the quantification of the extinction coefficient in the stratosphere as it is about 14 years after SAGE II's death. This should be questioned or, at the least, discussed.

Specific comments

L. 18-20, p.3: This sentense sound odd. What do the authors mean by "data (...) are made to match or conform with SAGE II"? Do they refer to the transformation of the other source data sets in extinction coefficient profiles at 525 nm/1020 nm at the SAGE II vertical grid? This should be clarified.

L. 20-22, p.3: A reference to the first GLoSSAC paper, Thomason et al. (2018) is necessary here to make clear what the authors mean.

L. 23-27, p.3: Same remark for this discussion: a reference to Thomason et al. (2018) is needed.

L. 30-32, p.3: Aren't these differences due to fundamental differences in measurement principles and in such a case, wouldn't it be a useful way to explore differences and possibly reconcile both techniques?

L. 33-34, p. 3: What are these changes included in interim version 1.1? Please refer to Section 2.1 where it is described or possibly provide some reference.

L. 34-35, p.3: If this data set is key, it should at least be cited!

L. 12-13, p.4: This has definitely to be developed and described carefully. Which ground-based lidar product was used, at which location, and which assumptions were used to match them with remote sensing data? Combination of lidar measurements and extinction measurements are not straightforward. Which lidar ratio was used, and how were the data combined?

L. 14-16, p.4: Again, how was this combination (here: CLAES-HALOE with SAGE II) implemented? If only a few points are considered, the possible impact of biases may be high? These aspcts should be carefully discussed.

L. 18-19, p.4: "A few defects missed in v1.0": which kind of defects and what were the consequences of these defects? Is there any publication or technical report where these modifications could be found?

L. 19-20, p.4: If these changes are qualified as "important", they should definitely deserve an appropriate discussion.

L. 21, p.4: How did this outlier removal occur? Smooth curves may be esthetically more satisfactory, but at risk of leaving out minor events of interest, and possibly of importance for the climate modelling applications the authors want to serve. Also, outlier removal may imply the use of poorly controlled data manipulation and of changes in values very dificult to trace. How did the authors deal with this difficulty? See also comment on L.8, p.7.

L. 24-26, p.4: This sentence is useful for readers not familiar with the SAGE II dataset. Please provide a citation where this issue is discussed.

L. 26-29, p.4: In Thomason et al. (2018), (at least) two kinds of interpolation mechanisms are used for gap filling. One is a linear intepolation in time (but not in latitude and altitude), and another one is the use of an empirical relationship between the 1020 nm and 525 nm extinction coefficient values defined from a statistical analysis of pairs of (1020 nm, 525 nm) extinction coefficient values retrieved from SAGE II observations (Fig. 8 of this paper). Which one is meant here by the authors?

L. 30, p.4: The concept of equivalent latitude is unclear for a possible "new reader". Please provide a reference.

L. 31, p.4: "The new filling mechanism" is unclear. Do the authors mean: "the filling mechanism by use of equivalent latitude"? (or "new more elaborate mechanism" that

might be distinguished from "simple mechanism"). Also, "the simple intepolation": do the authors mean "a linear interpolation" (with respect to time?)?

L. 33, p.4: Please be specific to ease the reading: "the simple (linear?) interpolation process"?

L. 1, p.5: It might be useful to specify that these quantities reflect the natural variability and the instrumental error, respectively. In Thomason et al. (2018), an increased value of zonal standard deviation is described when averaging by latitude is used, especially at the boundary of the polar vortex. Is it observed accordingly here that the more extensive use of the equivalent latitude results in a decreased zonal standard deviation?

L. 1-2, p.5: Again, this sentence requires a citation.

L. 3-5, p.5: This sentence is particularly unclear. Please rephrase, and specify sections or figures in Thomason et al. (2018) that may ease the understanding of the method.

Title §2.1 and l. 6-16, p. 5: I suggest to keep the structure and similar titles as in Thomason et al. (2018) by splitting this section is a §2.1 "The SAGE II period" and "The pre-SAGE II period". This should ease the reading, and a possible combined reading of both paper in parallel (and e.g. the comparison of methods used, such as interpolatin methods).

L. 10, p.5: "the results": Do the authors mean "the values extended along isentropic surfaces between Nov. 1981 and Oct. 1984"?

L. 5-7, p.6: I guess the two "potential sources of bias" are basically a single one. Please rephrase. This source of bias is not "potential", but real and potentially quite significant. In Thomason et al. (2018) the lidar ration was equal to 50. Why this change, and what are the effects of this change?

L. 22-23 p.6: Please provide some explanation or a reference for the PSC identification.

L. 19, p.6: "found in the lower stratosphere": at all latitudes?

L. 5-7, p.7: See comment on L. 21 p.4.

L. 8, p.7: Being resigned to accept this fact is harmful because it is known that the accumulation of medium eruptions plays an important role in the correct assessment of the aerosol radiative forcing [Vernier et al., Geophys. Res. Letter, doi:10.1029/2011GL047563, 2011; Bingen et al., Remote Sensing Env., doi: 10.1016/j.rse.2017.06.002, 2017], a key issue CMIP6 is intended to address.

L. 13-15, p.7: "the extreme outlier was effective at identifying outliers in the aerosol distribution": The formulation is confusing, please revise. "outliers in the aerosol distribution": do the authors mean "outlying data possibly related to medium volcanic/pyrocumulonimbus events"?

L. 7, p.8: Using a constant Angstrom exponent implies the assumption that the particle size distribution is constant. This is potentially a rough assumption impacting the accuracy of the values of the extinction coefficient at 525 nm used in GLoSSAC.

L. 11, p.8: What is a "strong aerosol measurement wavelength"?

L. 18, p.8: What do the authors mean by "a rather benign October 2004".

L. 19 and 32, p.8: These estimates are particularly optimistic. Following the color bars, the differences often exceed 50% in both cases.

L. 1-3, p.9: The assumptions made for the conversion of OSIRIS extinction coefficient from 750 nm to 525 nm seems an obvious cause of deficiency, which is confirmed by the result of the revision of the conversion factor as illustrated in Figure 5c (and the end of Section 2.4). See comment on L. 7, p.8.

L. 17-20, p.9: Did the authors compare the results obtained only with SAGE II, and only with SAGE III? This seems important to assess possible differences, either between the two SAGE instruments, or between both periods.

L. 22-24, p.9: This statement is particularly strange! The Angstrom exponent does have a physical meaning, since it reflects the size properties of the aerosol population. Pursuing as sole purpose the replication of one data set at all costs (even one supposed to be good, although its comparison with the real truth is impossible – this should always be kept in mind!) and getting rid of any concern about the correct quantification of known underlying effects at this aim, looks problematic to me.

L. 24-25, p.9: "Angstrom exponent values".

Caption Figure 4, 5, and 8: The quantity provide should be precisely mentioned, e.g.: "OSIRIS and SAGE II extinction coefficient at 525 nm". In caption of Figure 5, "for at" is not correct and "for" should be removed. In caption of Figure 8, "Altitude versus Latitude of percent difference." is meaningless. Difference in what? The authors should also clearly mention the period covered by this plot.

L. 27, p.9: I suggest to stick to the naming "Angstrom exponent". Please check the whole document.

L. 32, p.9: After using an Angstrom exponent of 2.33 to convert OSIRIS extinction coefficient from 750 nm to 525 nm (cf. L. 7, p. 8), another value of the same Angstrom exponent, 1.50, is used to convert the CALIOP extinction coefficient from 532 nm to 525 nm. Why such a difference? This incoherence should be discussed or justified.

L. 7-8, p.10: Smaller eruptions also occurred during the SAGE II mission (1984-2005). Is there any similar observations by SAGE II that might support such tendency? This might help depicting if such effect is real, or is the reflect of some limitation either of the OSIRIS instrument, or of the OSIRIS retrieval.

L. 25, p.10: "roughly consistent with values for sulfuric aerosol in the stratosphere": The extinction-to-backscatter ratio shows much variability in the stratosphere (See for example Vernier et al., Geophys. Res. Lett., 38, L12807, doi:10.1029/2011GL047563, 2011), and the size characteristics also play a role in the variability of this parameter.

Hence, I think that this statement is not very relevant.

L. 29, p.10: "As a result"? This sentence is the transition between considerations about version 1.0, and work around version 2. This should be made clear by an adequate introduction. Furthermore, at this stage, it would ease the reading to remind that the CALIOP extinction coefficient product by Kar et al. (2019) is the one used in GLoSSAC, as mentioned in L. 3-4, p.3.

L. 31-32, p.10: Why are the authors using now another value of the Angstrom exponent (1.50) for the conversion CALIOP, while a value of 2.33 was used before for OSIRIS extinction conversion? This is quite confusing and increase the level of incoherence between the data sets.

L. 7-10, p.11: I don't understand what the authors intend here. In §3, p.10, it is explained that the CALIOP extinction used in GLoSSAC is the CALIOP extinction product (Kar et al., 2019) at 532 nm, converted to 525 nm based on an Ångström exponent of 1.50. Why do they use now the CALIOP 532 nm backscatter converted using an empirical scaling factor, with some kind of warning that this scaling factor will also reflect "any kind of biases"? This is extremely confusing.

L. 13-27, p.11: I don't really understand what the authors are doing here. The CALIOP backscatter is the primary quantity measured by CALIOP. What is the interest of red-eriving the primary measured quantity from the CALIOP extinction (derived with a simplified assumption of a constant lidar ratio equal to 50), using an empirical scaling factor taking into account all possible problems ("aerosol-related effects and bias between the two data sets"), based on modified ("bias-corrected") OSIRIS extinctions at another wavelength with some rough approximation about the atmospheric transmission (mentioned as "clearly not correct" by the authors themselves) , and a simplified formula to account for the scattering ratio and molecular backscatter. And from the conclusion that "it does not matter a great deal whether we use the standard CALIOP stratospheric backscatter product or the alternative alternative", the authors choose

using this hazardous construction of alternative backscatter product! This is extremely strange and confusing, and if the aim is – again – to "match" at all costs CALIOP with OSIRIS, the methodology used is, at the least, questionable.

L. 29-30, p.11: The SF values varying between 25 and 65 might reflect the objective to get rid of the fixed 50-value of the lidar ration used by Kar et al. (2019) to better match local aerosol features. If this indeed is the case, the authors should completely revise this discussion to make it clear, and they should justify why they expect improvement with respect to Kar et al. (2019), see previous comment.

Figure 5: It is very strange to mix both SAGE II and SAGE II/ISS overlap periods as if these two SAGE sensors were one single data set or mission. SAGE II and SAGE III/ISS are two different instruments measuring different situations in very different conditions. Assimilating the SAGE II and SAGE III/ISS to one single perfect data set looks excessive, and at least, results for both data sets should also be shown (or quantified in some way) to justify that just mixing both is appropriate.

L. 1-4, p.12: The methodology used here is expected to provide more variations of the extinction-to-backscatter ratio than the fixed one assumed by Kar et al. (2019). However, the question is to know it the whole construction with a succession of more or less coarse assumptions used here provide a better estimate of this parameter. See also comment on L. 29-30, p.11

L. 20, p.13: Is the linear interpolation implemented only in the time dimension? What about the possible use of equivalent latitudes? This should be specified.

Figure 15: The choice of dynamic range for the color scale of pannels 15(a), (b), (d), and (e) is particularly poor. Same for Figure 16. Differences mentioned in L. 2-3, p.14 are hardly visible, and the "substantially smaller enhancement" in 2005 in version 2.0 with respect to version 1.1, is just invisible in both cases to me.

L. 4-11, p.14: Could several latitudinal dependence and hemispheric dependences

possibly be explained by differences in data coverage and/or in intrumental techniques? This possibility has not been discussed.

L. 22-27, p.14: I think, indeed, that in view of all efforts made to force some data sets to fit in as much as possible some other one, any discussion about trends Is absolutely premature.

Technical corrections

L. 24, p.2: incorrect sentence: "whose accuracy" should be removed.

L. 25, p.2: "Which this change"? (Or another change?)

L. 3, p.5: missing period (".").

L. 16-17, p.4: odd sentence.

P. 25, p.6: New sentence starting with "However, " ?

L. 3, p.5: "its".

L. 20, p.7: incorrect sentence: "can transition".

L. 24, p.7: incorrect reference: should be "Thomason and Vernier (2013)".

L. 16, p.9: "Extinction".

Caption Figure 9: The authors should be more explicit: "(b) Relative standard deviation of the extinction-to-backscatter ratio shown in (a)". "deviation of (a) in percent" is unclear.

L. 30, p. 12: "We use" with capital letter.

L. 31-32, p.12, L. 18, p.15, and caption Figure 11: "SAGE II and SAGE III/ISS".

L. 19-21, p.19: Rieger et al. (2019) is published, and the reference should be adapted.

---

## Referee Comment (RC3) · Anonymous Referee #3 · 7 Jul 2020

General comments: The paper describe the development of the stratospheric aerosol data set GloSSAC V2.0 and changes made since the last release of V1.1. The methodology of the data set construction and rational is clearly described. The paper is well written, and the results are presented and discussed with suffcent details. I recommend for publication subject to the following changes.

Major comment:

I find the authors choice of conforming rather than excluding data that exhibit very large biases (>50%) somewhat concerning. Mainly because some of these data may be affected by clouds. Kar et al. (2019) reported large differences with SAGE III/ISS below 20 km, which can be explained in part by subvisible cirrus cloud scattering artifacts that may appear to within several kilometers above the tropopause. In addition, the authors

failed to explain the advantage of using the newly released CALIOP standard products instead of version 4.0 Level 1 data used in GloSSAC V1.0. Do they believe that their approach produces better product than what was used in V1.0 and V1.1?

General comments:

Table 1: Some of the volcanic eruptions listed in table 1 did not reach the stratosphere and thus are not relevant to this dataset. The table should be modified to include only volcanic eruption that are evident in GloSSAC dataset.

Figure 1: the figure is identical to figure 1 published earlier by (Thomason et al., 2018), which shows SAGE III/ISS as a future instrument. I suggest either updating the figure or simply just cite the figure in (Thomason et al., 2018).

Page 3, first paragraph: The authors need to add a brief statement justifying the change and summarizing the differences between the new CALIOP products and the one used in V1.1.

Page 8 L6: "OSIRIS extinction is also routinely produced at 525 nm, . . ." should be replaced by "OSIRIS extinction can be produced at 525 nm, . . ." or something like that. The original text implies that it is part of the official V7.0 release.

Page 8 L10: "Since the SAGE III/ISS instruments operates in a manner virtually identical to SAGE II" "virtually identical" should be replaced by "similar" since the two instruments have different designs and age. Toward the end of its life, SAGE II was an aging instrument that operated on reduced duty cycle as compared to the newly refurbished SAGE III/ISS instrument.

Page 8, L15: "SAGE II and SAGE III/ISS are relatively unbiased with each other" this not accurate since both (Thomason et al., 2010) and (Damadeo et al., 2013) reported 10% bias between SAGE II and SAGE III Meteor, which is supposed to be identical to SAGE III/ISS. The differences between SAGE II and III should be acknowledged and discussed in this section.

Page 8, footnote 1: "While the OSIRIS instrument performance has remained un-changed over time," This not exactly accurate. According to Bourassa et al. (2018) and Rieger et al. (2019), OSIRIS had a small drift that resulted in a pointing error and a correction was applied to V7.0. Please modify the text accordingly.

Page 9, L23: I suggest changing "Angstrom exponent" (where appropriate) to some-thing like "pseudo Angstrom exponent" to eliminate any confusion regarding its physical meaning.

Figure 7: Can you add the year to the volcanic eruption label?

Section 3.1: The paragraph describing the choice between using the standard CALIOP stratospheric backscatter or the alternative product is confusing and difficult to follow, especially when the authors conclude that "it ultimately does not matter a great deal whether we use the standard CALIOP stratospheric backscatter product or the alter-native product described above". If that is the case, why not use the standard product and eliminate the confusion? Also, the CALIPSO section in supplementary materials implies that the standard products were used.

Page 11, L29-30: "except at higher altitude at polar latitudes where it is possible that the impact of the polar vortex plays a role in producing SFs less than 10 sr" I find it difficult to believe that the effect of PSC can cause this small ratio at altitudes between 25 – 30 km, and the statement is pure speculation. Unless the authors can show that this low ratio takes place during the winter season and low temperature, which are ideal for PSCs formation, I suggest deleting it.

Page 12, L11: The authors claim that the difference between CALIOP and SAGE III/ISS is now below 20%, when in fact figure 8d clearly shows the differences are mostly 30 and 40%. Please revise the text accordingly.

Figure 10 and 12: Can you change the y-axis to linear scale instead of log. The log scale makes it difficult to see the differences between different measurements.

Figure 6 and 11: Can you modify the color scale to -1 to 5.

Page 13 second paragraph: The authors use angstrom exponents to infer 1020 nm, similar to the method used to derive 520 nm. Can they comment on any potential use the two wavelengths in climate models, which will most likely use it to infer particle size information?

Figure 13, 14 and 15: Can you modify the color scale to properly show the volcanic enhancements. Also, can you remove the labels as they are distracting and interfere with the figure.

Section 5: It's difficult to follow the arguments regarding figure 15 because of the color scale, which doesn't show the author's argument. Please modify the color scale accordingly.

Figure 16: Again, can you adjust the color scale to show the smaller volcanic eruptions? In addition, remove the labels and add a symbol or a label denoting the location and time of each volcano.

Figure 17: Can you comment on the lack of seasonality in CALIOP data in the southern and northern hemisphere compared to other data set?

Section 6 Conclusions and future work: There is no mention of any addition of new data sets when figure 1 implies that SCIAMACHY and OMPS will be added in the future.

Supplementary materials, Figure 1: The figure needs further explanation, what year, SAGE II or III/ISS? The text implies both datasets without explaining the methodology to combine it. What wavelength? 750 nm converted to 525? How?

---

## Author Comment (AC5) · 18 Aug 2020

**Response to Reviewer #1**

We thank the reviewer for helpful comments. Our responses to the reviewer's specific comments are listed below. The reviewer's concerns are in bold italicized font and our responses are in regular font. The page numbers and line numbers given in our responses below are in reference to the revised version of the manuscript.

*Radiative forcing from stratospheric aerosols is a major driver of climate variability. Building robust and consensual observational records of stratospheric aerosol observations is thus critical to understand past climate changes and to predicting future climate. For the satellite era, this task is made particularly challenging by the fact that different periods are covered by different instruments with different measurement technique, so that producing an homogeneous and consistent continuous record is difficult. To address this challenges, the Global Space-based Stratospheric Aerosol Climatology (GloSSAC) dataset v1.0 was produced (Thomason et al., ESSD 2018) and has since become the reference dataset of stratospheric aerosol observations. In particular, it has been used as input to historical experiments of Phase 6 of the Climate Model Intercomparison Project. This manuscript presents the latest version v2.0 of the GloSSAC dataset which follows the original version v1.0 and its update v1.1. Many improvements have been done since version v1.0 and in particular: i) the dataset is now extended to year 2018; ii) an error in the processing of data from the CLAES instrument has been corrected; iii) the processing of data from the OSIRIS and CALIPSO instrument has been improved, in particular thanks to data from SAGE III having become recently available and overlapping with the OSIRIS/CALIPSO observations. Given these major improvements and the importance of the GloSSAC dataset, this manuscript will be an extremely valuable contribution and I recommend its publication following minor revisions. The manuscript describes and justifies in great details the updates to GloSSAC, and generally reads well. There are a few places where I think the figures or text could be improved (cf minor comments below). My main comment is that even though technical changes since v1.0 are well described, the paper lacks figure(s) and discussion extensively comparing v2.0 to v1.1 and v1.0 (cf specific comment below), which seems important given this paper provides an update to an existing dataset. I think this would represent an important improvement to the paper, and that it wouldn?t require much work from the authors which is why I recommend minor revisions.*

Specific comments

*1 ) If I?m not mistaken Figure 15 is the only figure showing the final differences between two versions of the GloSSAC (v2.0 and v1.1) and only do so for the 2002-2016 period. I don?t think GloSSAC updates have been documented in a peer-reviewed literature since GloSSAC 1.0 release (Thomason et al. 2018)? It thus seems very*

*important to show the differences between all three versions (1.0, 1.1 and 2.0) and for the full period in common (1979-2016). From my own analyzes of GloSSAC version it seems that each version is different in the post-Pinatubo period, which is one of the period with the most research on stratospheric aerosol forcing. I thus really think that figure 15 should be extended to include all versions/the full common period. In addition to showing contour plot of SAOD as a function of latitude and time, I also think it would be very valuable to compare global mean SAOD time series between the three versions as this is the canonical metric for stratospheric aerosol impact on climate. Section 5 of the manuscript should then be extended to discuss these differences in greater details. Getting an idea of SAOD differences among GloSSAC versions will likely be a major expectation of GloSSAC users from this paper, so I strongly encourage the authors to address this comment.*

Figure 15 is now extended to the entire record of GloSSAC and now shows AOD plots of v 1.1 and v 2.0. We also included another figure (Figure 16) that show v 1.0 and v 2.0 and their differences. Additional plots include a global SAOD plot (Figure 18) with labels of volcanic eruptions on Figures, and we discuss these changes in section 5.0.

*2) In line with comment 1 above, I think that the abstract should end with a few sentences summarizing the main changes between versions in terms of SAOD. The abstract is very focused on the technical changes in GloSSAC 2.0 which is of course appropriate for an ESSD paper, but it is currently hard for a scientist with little expertise in remote sensing to get a sense of the impacts of these changes on the GloSSAC product from the current abstract.*

Thanks for this suggestion. While we appreciate the reviewer for this suggestion, due to the word limit constraints for the abstract, we are not able to include this into the abstract. We now address this in the conclusion section of the paper (lines 3:8, page 17).

*3) This comment is very much a suggestion. Figure 1 is an excellent introduction figure to the manuscript. I was wondering if it could be complemented (or if you could add a new figure) showing a timeline of some of the main features/limitations/challeng in the GloSSAC record, such as what type of instrument is used (e.g. solar occultation or other), the resolution/frequency of measurements (e.g. global daily coverage with OSIRIS/CALIOP vs global monthly coverage with SAGE instruments), assumptions required (e.g. periods in which an assumption on size distribution is required), etc... Such a figure would enable people with limited expertise in remote sensing to understand in one glance some of the main features of the GloSSAC dataset before using it, which I believe would be very valuable.*

We have added a paragraph (line 25:30, page 2 and line 1:5, page 3) about the main features, limitations and challenges. The new paragraph now reads as:

"Figure 1 depicts the measurements that are currently used for constructing GloSSAC data. While Thomason et al. (2018) discusses about the measurements that have been used in GloSSAC v 1.0 dataset in detail, some of the main features of entire GloSSAC v2.0 dataset including various space based measurements, their limitations and some challenges are worth mentioning here. We divide the entire dataset into three periods based on the measurements used. The first period being the pre-SAGE II period (January 1979- September 1984), followed by SAGE II period (October 1984 - August 2005) , post-SAGE II period (September 2005- May 2017), and SAGE III/ISS period (June 2017-present). Pre-SAGE II period data mostly consists of data from solar occultation measurements such as SAM II , SAGE and some surface based Lidar measurements (Thomason et al., 2018). For SAGE II period, the measurements are dominated by solar occultation measurements that provide multi-wavelength measurements for size information. For the post-SAGE II era, we are limited to single wavelength measurements from OSIRIS and/or CALIPSO. While OSIRIS and CALIPSO continue to make daily global measurements with a less direct measurement of aerosol extinction coefficient that requires further assumption of particle size, additional direct measurements of aerosol extinction coefficient from SAGE III/ISS are now available that provides a roughly monthly coverage of multi-wavelength measurements since June 2017. "

*page 1 line 1: I stumble a bit on the first sentence of the abstract. In addition to being a bit cumbersome, it introduces what a stratospheric aerosol dataset should do, but the second sentence does not follow on the dataset so it?s confusing.*

We rewrote the sentence to have the continuity.

*page 1 line 7: I don?t think Zanchettin et al. (2016) is the adequate reference for CMIP6 unless you are talking specifically about VolMIP.*

Replaced with Eyring et al. (2016).

*Page 2 line 4-5: "can impact climate on scales from the subtle [...]  to the more profound [...]": I find this wording vague and confusing: are you talking about the timescales of the impacts? Their magnitude?*

We changed it to "can impact climate on magnitudes from ....

*I would be more specific and clear about the difference in these modelling approaches, e.g. "Some of these modelling studies directly use observations of stratospheric aerosol optical properties as input, whereas other use observations of SO2 as input and interactively simulate stratospheric aerosol life cycle"*

We specifically state that in the following paragraph (lines 10-15, page 2).

*page 2, line 6-14: You focus on GCMs study as a motivation but I feel like you could*

*include other examples that have used the GloSSAC dataset to make important contributions. In a purely observational study, Stocker et al (2019, https://agupubs.onlinelibrary quantify the temperature footprint of 21st century eruptions using GloSSAC which in turns enable to better quantify temperature trends related to anthropogenic forcing. Aubry et al. (2020, https://agupubs.onlinelibrary.wiley.com/doi/full/10.1029/2019JD03130 used GloSSAC to calibrate a box model of volcanic forcing. Such box model is the typical tool used to derive SAOD/forcing time series from ice-core records so that?s an important application of dataset like GloSSAC. These are just suggestions and application to GCM study is of course a major motivation, but I think it?s nice to highlight that applications of the GloSSAC dataset go beyond that.*

Thanks for the suggestion. We now added a sentence on Stocker et al., 2019 (line 15, page 2).

*page 2, line 14: I am not too sure where to put this comment, but it feels like the paper under review by Rieger and co-authors (https://gmd.copernicus.org/preprints/gmd-2019-381/) is very relevant to your paper and should be mentioned if its publication status allows it. Maybe instead of in the introduction it would fit better in your discussion of an extended figure 15 (cf specific comments) showing the differences in SAOD between v1.0, v1.1 and v2.0 for 1979-2016.*

We have added this reference in SAOD discussion section.

*page 2, line 15: it?s a bit confusing as you say "mostly unchanged" followed by "significant improvements" and "major version change". These statements sound contradicting and you may want to reformulate.*

The data sources are unchanged. However the usage of the data (with some updated versions) and the analysis approach resulted in significant improvements. We have revised the sentence now and it reads as:

"While data sources are mostly unchanged from earlier versions, there are significant improvements in the use of OSIRIS and CALIPSO data with inclusion of SAGE III/ISS data for the first time."

*page 3, line 13-33: this feels like a very detailed and technical discussion of the changes you made for an introduction. I feel like this content should be in section 2 instead?*

We have now moved those lines to section 2.

*page 4, line 7: there and hereafter, I suggest you provide date in parenthesis when you refer to XXX instrument period. It will be very helpful for readers not perfectly familiar with the period spanned by different instrument.*

Done.

*page 4, lines 10-18: if I?m not mistaken no figure illustrate these results? The Pinatubo period is of course of utmost interest to climate modelers so it feels like there should be a figure accompanying this paragraph? (although if you extend Figure 15 according to my suggestions that would illustrate this paragraph well)*

We now discuss this in section 5.0.

*page 5, line 14-16: this is an important comment. Again it would be nice to specify dates in parenthesis for user who are not familiar with SAGE missions dates. I know that the reader could just look at Figure 1 but it would facilitate the reading if you also provide such dates directly.*

Done.

*General comment on section 2: I enjoyed this section and although I don?t have the expertise to understand all details, you clearly highlight the differences in methods/ limitation/challenges of different periods of the GloSSAC record. A figure with a timeline showing this features would be a very neat addition (see specific comment 3 for a more detailed suggestion)*

We have now added a paragraph in the beginning of section 2 and discuss Figure 1 and the instruments used in detail.

*page 6 line 7: maybe you could give an idea of the uncertainty on the 50 sr value?*

This number comes from OPC and lidar measurements study Jäger and Deshler (2002, 2003) that is based on specific size distributions. It could also be tested theoretically using Mie theory with an assumption of size distributions. Again, please note that lidar ratio strongly depends on size distribution. So, specifically using a constant number for extinction-to-backscatter ratio has limitations. For this study, we however used a pseudo-extinction to backscatter ratios (defined as "Scale Factor" in the manuscript) and its related uncertainties in Figure 9. Using a value of 53 sr seems reasonable between 30S and 30N from 18 km and above based on Figure 9a and the relative standard deviation based on our method is mostly within $\pm$ 20% (Figure 9b).

Also, Vernier et al. (2011) reported the variability in lidar ratios across various latitude bands, showing lidar ratios vary with latitude and altitude.

*page 6, lines 25-34: Is this cloud-clearing method more challenging to apply when there is a very large volcanic eruption (e.g. Pinatubo like or larger)? I?m just wondering whether the IQR would be larger following a large eruption.*

There are limitations on this method as well, especially when we use this in the vicinity of tropopause with large eruptions. It is particularly challenging to differentiate between clouds and aerosols near

tropopause during and following volcanic eruptions. We are currently working on developing a cloud screening algorithm for SAGE III/ISS in particular which could be incorporated in a future version of GloSSAC.

**Page 8 line 4-5: does the Raikoke 2019 eruption provide a good test for this hypothesis?**

The technique we use in here has limitations when it comes to periods of volcanic activity particularly due to change in size distributions and for the period 2005-2017, we are limited to using single wavelength measurement that lacks information about aerosol sizes. Yes, multiwavelengths measurements have been available from SAGE III/ISS since June 2017, which help us understand better as to how aerosol size changes during and following a volcanic eruption. For the Raikoke eruption, a detailed study using SAGE III/ISS measurements is in progress.

**Page 9 line 2-3: why do you say "though probably not at 756nm"? Doesn?t figure 5b show a strong high bias in the lower stratosphere and low bias in the tropical midstratosphere?**

We are aware of a low bias at 521 nm channel of SAGE III/ISS. We, however do not observe any such changes in other aerosol measurement wavelengths such as 756 nm. In addition, for Figure 5a, there is an additional complexity that the OSIRIS 525 nm extinction coefficient is computed from a constant Angstrom exponent of 2.33 while 750 nm extinction coefficient is the primary reported wavelength for OSIRIS and Figure 5b is a straightforward comparison with SAGE III/ISS. Therefore, comparing the differences between Figure 5a and 5b is not a direct one.

**Page 9 line 14: I can?t find the definition of lambda**

Lambda represents wavelength. The sentence now reads as : "$\left(\frac{\lambda_{525}}{\lambda_{750}}\right)$ represents ratio of wavelengths at 525 and 750 nm."

**section 3-4: these sections were generally clear and provide a good overview of differences between instruments and data processing/conforming procedures employed by the authors.**

Thanks.

**page 13 line 33: So I guess the tropopause height is a climatology as in Thomason et al. (2018)? I think it would be useful to remind here the period used to derive this climatology, as well as the reanalysis used (MERRA if I remember correctly). Additionally, the tropopause height is quite variable at highlatitude and is increasing in the tropics as a consequence of anthropogenic forcing. Given these two points, I am wondering why you are using a climatology instead**

*of the reanalysis data directly? Differences would likely be small but it would be a bit more rigorous approach? As an example figure S1 in Aubry et al (2020, https://agupubs.onlinelibrary.wiley.com/doi/full/10.1029/2019JD031303) show GloS-SAC SAOD at v1.0 and v1.1 using the MERRA climatology and the NCEP-NCAR reanalysis for tropopause height with some interesting differences. I haven?t analyzed these differences further, but the SAOD I get are smaller for the 21st century which could be in part due to tropopause height increase?*

The tropopause climatology is derived from MERRA for the SAGE II lifetime as stated in Thomason et al. (2018) paper. We continue to use that in here as well. Yes, we do agree that the tropopause is variable with latitude and time of the year and using a climatology may not be an accurate representation. We will definitely keep this suggestion in mind and make use of a variable tropopause in a future version of GloSSAC.

*Section 5: please see my specific comment 1, but I really believe that this section would be more complete if you: -show differences between all 3 versions of GloSSAC -show differences for the full time period shared between the 3 version (1979-2016) -show global mean SAOD time series in addition to SAOD contour plot -extend text in section 5 to include discussion of the above*

We have now revised this section to include the entire record of SAOD.

*page 14 lines 22-23: it feels like you could add a few references to support this statement? There have been multiple modelling studies showing that post- 2005 SAOD enhancement can be largely explained by SO2 emissions from explosive volcanic eruptions as well as wildfire for some of the recent years. See e.g. Schmidt et al. (2018, https://doi.org/10.1029/2018JD028776), Peterson et al. (2018, https://www.nature.com/a: 018-0039-3) or Aubry et al. (2020, https://agupubs.onlinelibrary.wiley.com/doi/full/10.102:*

Done. The sentence now reads as:

"As with v1.0, we cannot exclude the possibility that on-going volcanic activity plays a dominate role in the apparent enhancement after 2005. This possibility is bolstered by noting that optical depths shown in Figures 15b and 15e approach those observed in 2004, 2013, early 2014 during a lull in a decade of repeated minor volcanic stratospheric enhancements. We also note that several recent modeling studies (e.g. Schmidt et al., 2018; Aubry et al., 2020) using sulfur dioxide emissions in aerosol-climate models, have reported an enhancement in SAOD for the post-2005 time period."

*page 16 line 9: avoid repetition of "inferred 1020nm extinction" twice in the same sentence. Otherwise, I think this is a nice paragraph to close the conclusion section!*

Done.

*Section 6: the conclusions are ok but overall I feel like you could be a bit more succinct on some of the technical details, and that you should add a few sentence commenting on major differences in SAOD in GloSSAC 2.0 compared to 1.0/1.1. This really seems critical as the aim of the paper is to present the newest version of the GloSSAC dataset, so in general it really feels like you should do more to compare SAOD in the different versions. This would likely be the most expected results from this paper for users of the GloSSAC product.*

We now briefly discuss changes in SAOD occurred in version 2.0 compared to previous versions.

*Table 1 caption: replace "since 2002" by "over 2002-2018" as this table doesn?t include 2019/2020 eruptions (e.g. Raikoke and Ulawun in 2019, Taal in 2020)*

We have now revised the table to include all events since 1979.

*Figure 7: I like this figure a lot. Maybe you could have a SI table specifying the dates of "after"/"before" (or add these dates in the caption directly)*

We have added dates to each event in the figure. There is another paper in review in ACP ( https://doi.org/10.5194/acp-2020-480) that discusses this method in detail.

*Figures 10/12/17: given the differences between datasets are relatively small, I?m wondering if you should not use lines instead of markers? The marker sometime overlaps a lot making it harder to distinguish any systematic difference.*

Done. Replaced makers with lines.

*Figures 13-16: on most of these figures, the density of contour labels is too high and prevent the reader to see clearly the data on the figure (this is made worse by the white rectangles in which each label is inserted). Consider removing the labels altogether or at least reducing their density/removing white rectangles.*

Done. Removed labels and contours from all these plots and a new color scale has been used.

*Figure 16: it looks like this figure has been stretched horizontally?*

This has been fixed. It was in fact related to the adjustment of the figure size that was made in the latex version.

**References**

Aubry, T. J., Toohey, M., Marshall, L., Schmidt, A., and Jellinek, A. M.: A New Volcanic Stratospheric Sulfate Aerosol Forcing Emulator (EVA_H): Comparison With Interactive Stratospheric Aerosol Models, Journal of Geophysical Research: Atmospheres, 125, e2019JD031 303, https://doi.org/10.1029/2019JD031303, e2019JD031303 10.1029/2019JD031303, 2020.

Eyring, V., Bony, S., Meehl, G. A., Senior, C. A., Stevens, B., Stouffer, R. J., and Taylor, K. E.: Overview of the Coupled Model Intercomparison Project Phase 6 (CMIP6) experimental design and organization, Geoscientific Model Development, 9, 1937–1958, https://doi.org/10.5194/gmd-9-1937-2016, URL `https://gmd.copernicus.org/articles/9/1937/2016/`, 2016.

Jäger, H. and Deshler, T.: Lidar backscatter to extinction, mass and area conversions for stratospheric aerosols based on midlatitude balloonborne size distribution measurements, Geophys. Res. Lett., 29, 35–1–35–4, https://doi.org/10.1029/2002GL015609, 1929, 2002.

Jäger, H. and Deshler, T.: Correction to "Lidar backscatter to extinction, mass and area conversions for stratospheric aerosols based on midlatitude balloonborne size distribution measurements", Geophys. Res. Lett., 30, n/a–n/a, https://doi.org/10.1029/2003GL017189, 1382, 2003.

Schmidt, A., Mills, M. J., Ghan, S., Gregory, J. M., Allan, R. P., Andrews, T., Bardeen, C. G., Conley, A., Forster, P. M., Gettelman, A., Portmann, R. W., Solomon, S., and Toon, O. B.: Volcanic Radiative Forcing From 1979 to 2015, Journal of Geophysical Research: Atmospheres, 123, 12 491–12 508, https://doi.org/10.1029/2018JD028776, 2018.

Thomason, L. W., Ernest, N., Millán, L., Rieger, L., Bourassa, A., Vernier, J.-P., Manney, G., Luo, B., Arfeuille, F., and Peter, T.: A global space-based stratospheric aerosol climatology: 1979–2016, Earth System Science Data, 10, 469–492, https://doi.org/10.5194/essd-10-469-2018, 2018.

Vernier, J.-P., Thomason, L., Pommereau, J.-P., Bourassa, A., Pelon, J., Garnier, A., Hauchecorne, A., Blanot, L., Trepte, C., Degenstein, D., et al.: Major influence of tropical volcanic eruptions on the stratospheric aerosol layer during the last decade, Geophys. Res. Lett., 38, 2011.

---

## Author Comment (AC6) · 18 Aug 2020

Response to Reviewer #2

We thank the reviewer for helpful comments. Our responses to the reviewer's specific comments are listed below. The reviewer's concerns are in bold italicized font and our responses are in regular font. The page numbers and line numbers given in our responses below are in reference to the revised version of the manuscript.

*General Comments The paper presents the new version of the GLoSSAC aerosol climatology (version 2), and details all changes brought with respect to the former version 1.1, for wich changes are also briefly described with respect to version 1.0. Prominent changes are the availability of a new version for OSIRIS (version 7.0) with an improved quality, and th release of a standard CALIOP extinction product. Beyonf an improvement of the overall quality, this brings new possibilities to refine the derivation of some GLoSSAC products (e.g. through the use of variable extinction-to-backscatter ratio). Overall, the paper is clear, well written and well structured, in particular the introduction and conclusions. In many places, citation of the first GLoSSAC paper (Thomason et al., 2018) is required. I suggest to specify the section to which the citation refers in this paper, in order to ease the reading. A recurrent assuymption is that the SAGE instrument SAGE II and SAGE III/ISS are golden standards, and the key benchmark by which all other data sets have to conform. A very good reason for this is that SAGE instruments are using solar occultation, a technique requiring few assumptions for the data retrieval. SAGE II also has an excellent reputation and was a very long-duration mission. However, doing so ignores the possibility that SAGE II ageing affects the quality of the measurements at the end of the SAGE II, although the use made of SAGE II to calibrate OSIRIS and CALIOP is of critical importance for GLoSSAC. It should be reminded that SAGE II is about 18-years old when OSIRIS and CALIPSO are launched. On the other hand, SAGE III/ISS is recent, and one could miss the broader view on the real quality of this data set. This point of view should also be discussed or at least mentioned, with reference to validation papers giving more insight into the quality of the dataset during the critical period overlapping with the OSIRIS and CALIPSO missions. Overall, the paradigm is that everything is fine tuned to match the two SAGE datasets (and OSIRIS where these datasets are unavailable), but sometimes at all costs, without too much consideration for the consistency or physical significance of the methodology (e.g. different Angstrom exponents used for OSIRIS and CALIOP conversion purposes, ?We do not assume that the derived Angstrom coefficient has any physical meaning it is simply a mean to push OSIRIS toward SAGE II?, L. 22-24, p.9). The fact that instruments (SAGE, OSIRIS, CALIOP) are based on totally different measuring techniques that might have an impact in some altitude or latitude range is hardly considered or discussed, although this might provide an insight into main differences between the data sets. The way CALIOP backscatter coefficient is conformed to GLoSSAC extinction coefficient is also not fully convincing.*

*Automatically considering a hierarchy of values (?SAGE II is the best instrument?, ?SAGE II/ISS is equal to SAGE II?, ?OSIRIS in the best one after SAGE?) without questioning the physics, the evolving atmospheric state, or any consideration related to aging instruments, has the consequence that SAGE II?s spectre is still hovering on the quantification of the extinction coefficient in the stratosphere as it is about 14 years after SAGE II?s death. This should be questioned or, at the least, discussed.*

**Specific comments**

*L. 18-20, p.3: This sentense sound odd. What do the authors mean by ?data (: : :) are made to match or conform with SAGE II?? Do they refer to the transformation of the other source data sets in extinction coefficient profiles at 525 nm/1020 nm at the SAGE II vertical grid? This should be clarified.*

Here, for example, we use data from OSIRIS and CALIOP other than SAGE. And, their primary quantity of measurement is different (OSIRIS reports extinction coefficient (after retrieval) at its primary wavelength of 750 nm whereas CALIOP's primary measurement is backscatter coefficient). To match or conform to SAGE II means converting these into SAGE II primary measurements of aerosol extinction at 525 and 1020 nm, either using an Angstrom exponent (for OSIRIS) or using a lidar ratio (for CALIPSO). We included information about primary measurements of OSIRIS and CALIOP as examples in the text (Line 2, page 4).

*L. 20-22, p.3: A reference to the first GLoSSAC paper, Thomason et al. (2018) is necessary here to make clear what the authors mean.*

Thomason et al. (2018) reference is now included in the text whenever it is needed.

*L. 23-27, p.3: Same remark for this discussion: a reference to Thomason et al. (2018) is needed.*

Done. (line 18, page 4)

*L. 30-32, p.3: Aren?t these differences due to fundamental differences in measurement principles and in such a case, wouldn?t it be a useful way to explore differences and possibly reconcile both techniques?*

There are fundamental differences in measurement principles. However, the difference that we see, for example in the conversion of backscatter coefficient to extinction clearly shows some anomalous extinction to backscatter ratios when CALIPSO backscatter coefficient is compared against OSIRIS and/or SAGE extinctions, indicating these differences are due to measurement/conformance deficiencies. These anomalous extinction to backscatter ratios are consistently seen in the entire record of comparison. In addition, we note the differences between OSIRIS and SAGE II/III extinctions and their possible causes were reported earlier (Bourassa et al., 2012; Rieger et al., 2015; Kremser

et al., 2016).

**L. 33-34, p. 3: What are these changes included in interim version 1.1? Please refer to Section 2.1 where it is described or possibly provide some reference.**

The changes made to v 1.1 was mostly to correct how CLAES data is used in the lower stratosphere for the period between July 1991 and April 1993.

We have added a sentence that reads as "The changes made to interim version 1.1 is described below in Section 2.1." (Line 26, page 4)

**L. 34-35, p.3: If this data set is key, it should at least be cited!**

We added a reference in here. The sentence now reads as :

"Within v2.0, a key data set (McCormick et al., 1979) used in the SAGE/SAGE II gap period (1982-1984) has been updated." (line 27, page 4)

**L. 12-13, p.4: This has definitely to be developed and described carefully. Which ground-based lidar product was used, at which location, and which assumptions were used to match them with remote sensing data? Combination of lidar measurements and extinction measurements are not straightforward. Which lidar ratio was used, and how were the data combined?**

A detailed description about this is given in section 2.2 (p 8-10) of Thomason et al. (2018) and in section 4.2 of SPARC (2006). The lidar ratios were used from Jäger and Deshler (2002, 2003). We added the reference now and the sentence now reads as:

"CLAES data becomes available in October 1991 but is used in combination with a ground-based lidar product to estimate the aerosol levels from July to September 1991 and is used standalone for a decreasing span of altitude and latitude until the end of its mission in April 1993 (Thomason et al., 2018; SPARC, 2006)."

**L. 14-16, p.4: Again, how was this combination (here: CLAES-HALOE with SAGE II) implemented? If only a few points are considered, the possible impact of biases may be high? These aspcts should be carefully discussed.**

Please note that this is described in detail in section 2.2 of Thomason et al. (2018). Actual coincidences, by usual standards is pretty low, so we are using some binned comparisons but it still ends up averaging hundreds of data points. The uncertainty in the fits is a part of the error budget for these parameters. In addition, figures 6,7, and 8 of Thomason et al. (2018) show how the combinations were used.

**L. 18-19, p.4: ?A few defects missed in v1.0?: which kind of defects and what were the consequences of these defects? Is there any publication or technical report where**

*these modifications could be found?*

While it did not make much difference between the two approaches, we decided that the more conservative use of CLAES and HALOE data preserved the sampling pattern of SAGE II that appears in the entire SAGE II part of the record (1984-2005). Unfortunately, we do not have any publication on this. There is a product quality summary document available from ASDC for v 1.1 and 2.0.

*L. 21, p.4: How did this outlier removal occur? Smooth curves may be esthetically more satisfactory, but at risk of leaving out minor events of interest, and possibly of importance for the climate modelling applications the authors want to serve. Also, outlier removal may imply the use of poorly controlled data manipulation and of changes in values very dificult to trace. How did the authors deal with this difficulty? See also comment on L.8, p.7.*

We do not think it adversely affects the data by applying an outlier filter as we note that the filtering has only very minimal impact on 452, 525 and 1020 nm extinction coefficients but it impacts 386 nm extinction by reducing noise- a channel with increased noise (Thomason et al., 2008) and should be used cautiously due to the noise in the data which we note in the manuscript and in the previous version (Thomason et al., 2018).

*L. 24-26, p.4: This sentence is useful for readers not familiar with the SAGE II dataset. Please provide a citation where this issue is discussed.*

We added a couple of references here and the sentence now reads as:

"Users should continue to use caution using the SAGE II 386 nm aerosol extinction coefficient data as a low bias is evident in this data in the lower and upper stratosphere and at all altitudes as aerosol extinction coefficient magnitudes approach background levels (Thomason et al., 2008, 2018). "

*L. 26-29, p.4: In Thomason et al. (2018), (at least) two kinds of interpolation mechanisms are used for gap filling. One is a linear intepolation in time (but not in latitude and altitude), and another one is the use of an empirical relationship between the 1020 nm and 525 nm extinction coefficient values defined from a statistical analysis of pairs of (1020 nm, 525 nm) extinction coefficient values retrieved from SAGE II observations (Fig. 8 of this paper). Which one is meant here by the authors?*

We used the empirical relationship between 1020 and 525 nm extinction coefficient values.

*L. 30, p.4: The concept of equivalent latitude is unclear for a possible ?new reader?. Please provide a reference.*

We have now added reference and the sentence now reads as:

"In addition, with the apparent success of filling the high latitudes using the equivalent latitude/latitude mechanism developed based on Manney et al. (2007) for v1.0 (Thomason et al., 2008), we have reduced the role of simple linear interpolation at high latitudes and allow the new equivalent latitude/latitude mechanism fill more of the missing data at high latitudes."

*L. 31, p.4: ?The new filling mechanism? is unclear. Do the authors mean: ?the filling mechanism by use of equivalent latitude?? (or ?new more elaborate mechanism? that might be distinguished from ?simple mechanism?). Also, ?the simple intepolation?: do the authors mean ?a linear interpolation? (with respect to time?)?*

We have revised the sentence now and it reads as:

"In addition, with the apparent success of filling the high latitudes using the equivalent latitude/latitude mechanism developed based on Manney et al. (2007) for v1.0 (Thomason et al., 2008), we have reduced the role of simple linear interpolation at high latitudes and allow the new equivalent latitude/latitude mechanism fill more of the missing data at high latitudes."

A detailed description as to how the interpolation is done is given in Thomason et al. (2018).

*L. 33, p.4: Please be specific to ease the reading: ?the simple (linear?) interpolation process??*

We are not sure what the reviewer is referring to ? If it is line 31, we now changed to "linear" interpolation.

*L. 1, p.5: It might be useful to specify that these quantities reflect the natural variability and the instrumental error, respectively. In Thomason et al. (2018), an increased value of zonal standard deviation is described when averaging by latitude is used, especially at the boundary of the polar vortex. Is it observed accordingly here that the more extensive use of the equivalent latitude results in a decreased zonal standard deviation?*

The use of equivalent latitude reduces the zonal standard deviation in areas where strong zonal gradients occur. The standard deviations we report are always a combination of measurement noise and natural zonal variability.

*L. 1-2, p.5: Again, this sentence requires a citation.*

We are not sure about the citation of Line 1-2. If it is about CLAES and HALOE data sets, we have now included references to those data sets. The sentence now reads as : "The CLAES (Massie et al., 1996) and HALOE (Thomason, 2012) data sets now include zonal standard deviation and median reported measurement uncertainty following the approach used for SAGE II data. The conversion of CLAES and HALOE data follow the methodology described in Thomason et al. (2018)."

*L. 3-5, p.5: This sentence is particularly unclear. Please rephrase, and specify*

***sections or figures in Thomason et al. (2018) that may ease the understanding of the method.***

The conversion of CLAES and HALOE data is described in detail in Thomason et al. (2018). We, think that including the same method here will be a repetition.

***Title 2.1 and l. 6-16, p. 5: I suggest to keep the structure and similar titles as in Thomason et al. (2018) by splitting this section is a 2.1 ?The SAGE II period? and ?The pre-SAGE II period?. This should ease the reading, and a possible combined reading of both paper in parallel (and e.g. the comparison of methods used, such as interpolatin methods).***

Done. We now use two subsections under Section 2.1.

***L. 10, p.5: ?the results?: Do the authors mean ?the values extended along isentropic surfaces between Nov. 1981 and Oct. 1984??***

Yes. The sentence now reads as:

"While the revised file was nominally created in same way as the existing data file, the values extended along the isentropic surfaces can be significantly smaller at times than those used in v1.0 particularly in the Southern Hemisphere during the Spring."

***L. 5-7, p.6: I guess the two ?potential sources of bias? are basically a single one. Please rephrase. This source of bias is not ?potential?, but real and potentially quite significant. In Thomason et al. (2018) the lidar ration was equal to 50. Why this change, and what are the effects of this change?***

The sentence now reads as: "Finally, the conversion from backscatter coefficient to extinction coefficient presents a source of bias; as this process depends on details of an unknown aerosol composition and size distribution (Kar et al., 2019) that is a another potential source of bias."

While the method described in Thomason et al. (2018) is dependent on a median value that is obtained from a relationship between CALIOP backscatter coefficient and OSIRIS extinction, the lidar ratio of 53 is from Kar et al. (2019). We do not think that it makes much of a difference between using 50 and 53 in the lidar ratios. Also, the method used in Thomason et al. (2018) for the CALIOP data is different compared to Kar et al. (2019) method. We also see significant difference in the backscatter coefficient data between these two versions in the lower stratosphere and also at higher latitudes. Therefore, comparing these two data set based on lidar ratio difference may not be a direct comparison.

***L. 22-23 p.6: Please provide some explanation or a reference for the PSC identification.***

Different methods have been used for detecting PSCs in different data sets. For SAGE measurements, we use a threshold temperature of 200 K, meaning if the temperature is below 200 K between tropopause and 25 km, then those measurements are eliminated as PSCs. For OSIRIS data, additional constraints are used in addition to temperature based PSC detection (Rieger et al., 2019). For CALIPSO, a method employed by Pitts et al. (2009) is used which is also a temperature based approach.

**L. 19, p.6: ?found in the lower stratosphere?: at all latitudes?**

Occurrences of clouds have often reported in the Upper Troposphere Lower Stratosphere (UTLS) region in the tropics ($\pm 20^0$) and midlatitudes ($\pm 40\text{-}60^0$) in addition to the PSC's in the polar latitudes.

**L. 5-7, p.7: See comment on L. 21 p.4.**

Again, we do not think it adversely affects the data by applying an outlier filter. And, as mentioned in the paper, we are not using the conservative IQR method which appears to remove some of the enhanced aerosol extinction data. We also ensured that we are not removing any peak data points that occurs due to any volcanic/fire events in the data as it is evident from the time series plots (Figure 10, 12 and 17) that show "before" and "after" conformance of OSIRIS data. These plots clearly show peaks associated with any volcanic/fire events. Again, we are using only a very minimal outlier removal as it can be seen from Figure 1.

**L. 8, p.7: Being resigned to accept this fact is harmful because it is known that the accumulation of medium eruptions plays an important role in the correct assessment of the aerosol radiative forcing [Vernier et al., Geophys. Res. Letter, doi:10.1029/2011GL047563, 2011; Bingen et al., Remote Sensing Env., doi: 10.1016/j.rse.2 2017], a key issue CMIP6 is intended to address.**

We are not sure what the reviewer is referring to. If the reviewer is concerned about the depiction of volcanic events in the GloSSAC data, it is clear from the time series plot (Figures 10, 12, and 17) that almost all volcanic events can be identified with the peak in the data and the cloud filtering method has a minimal impact on them.

**L. 13-15, p.7: ?the extreme outlier was effective at identifying outliers in the aerosol distribution?: The formulation is confusing, please revise. ?outliers in the aerosol distribution?: do the authors mean ?outlying data possibly related to medium volcanic/ pyrocumulonimbus events??)**

The sentence now reads as:

"We found that the conservative outlier appeared to remove many enhanced aerosol measurements particularly when stratosphere is perturbed due to volcanic/pyrocumulus events, whereas the extreme outlier was effective at identifying outliers in the density distribution. Therefore we use the

extreme outlier to clear cloud-affected observations from the data set."

***L. 7, p.8: Using a constant Angstrom exponent implies the assumption that the particle size distribution is constant. This is potentially a rough assumption impacting the accuracy of the values of the extinction coefficient at 525 nm used in GLoSSAC.***

This method has been employed in previous studies (e.g. Bourassa et al., 2012; Rieger et al., 2015). And, the conversion factor and the difference between OSIRIS and SAGE II measurements were noted in (Rieger et al., 2015) which led to a scaling of extinction based on OSIRIS to SAGEII extinction ratio (Rieger et al., 2015). As described in the following sections of the paper, we are using a climatological Angstrom exponent and currently we do not have any other way to address this issue. May be a possible transient Angstrom exponent can be implemented in a future version.

***L. 11, p.8: What is a ?strong aerosol measurement wavelength??***

What we mean by strongest aerosol wavelengths is with least uncertainty and increased accuracy in measurements. For SAGE II and SAGE III/ISS, there are two strongest wavelengths channel in common which are 525 and 1020 nm. We revise the sentence now as:

" Since the SAGE III/ISS instruments operates in a manner similar to SAGE II, the expectation is that there would be minimal bias between these instruments at least at the strongest aerosol measurement wavelengths of 525 and 1020 nm."

***L. 18, p.8: What do the authors mean by ?a rather benign October 2004?.***

The sentence is now revised as:

"In figures 4a and 4b, it is apparent that for much of the stratosphere the difference between SAGE II and OSIRIS is less than 10% particularly in Figure 4a for a rather benign (less affected by volcanic/fire events) October 2004."

***L. 19 and 32, p.8: These estimates are particularly optimistic. Following the color bars, the differences often exceed 50% in both cases.***

The sentence now reads as:

"However, it is also clear that OSIRIS extinction is consistently higher than SAGE II in the lower stratosphere with percentage difference exceeding 50% near the tropopause. Another departure is shown in Figure 4b for March 2005 that shows similar features as October 2004. However, in the tropical low and middle stratosphere there is a difference of about 50% in an enhanced aerosol layer associated with the eruption of Manan in January 2005."

***L. 1-3, p.9: The assumptions made for the conversion of OSIRIS extinction coefficient from 750 nm to 525 nm seems an obvious cause of deficiency, which is confirmed by the result of the revision of the conversion factor as illustrated in***

***Figure 5c (and the end of Section 2.4). See comment on L. 7, p.8..***

We agree that using a constant Angstrom exponent may not work well during volcanic events and we state that in the manuscript as one of the caveats. Again, this method has been employed in previous studies (e.g. Bourassa et al., 2012; Rieger et al., 2015). And, the conversion factor and the difference between OSIRIS and SAGE II measurements were noted in Rieger et al. (2015) which led to a scaling of extinction based on OSIRIS to SAGEII extinction ratio (Rieger et al., 2015). As described in the following sections of the manuscript, we are using a climatological Angstrom exponent and at this point we do not have any other way to address this issue. May be, a possible transient Angstrom exponent could be implemented in a future version.

***L. 17-20, p.9: Did the authors compare the results obtained only with SAGE II, and only with SAGE III? This seems important to assess possible differences, either between the two SAGE instruments, or between both periods.***

Since the SAGE III/ISS instruments operates in a manner similar to SAGE II and SAGE III meteor, the expectation is that there would be minimal bias between these instruments at least at 525 and 1020 nm. Some previous studies have compared/validated SAGE II with SAGE III meteor (Thomason et al., 2010; Damadeo et al., 2013). While the differences between SAGE II and SAGE III meteor aerosol extinction coefficient are relatively smaller, previous studies (Thomason et al., 2010; Damadeo et al., 2013) reported a small bias between SAGE II (v 7.0) and SAGE III (v 4.0) meteor that are within $\pm$ 10% for measurement wavelengths at 525 and 1020 nm for the altitudes between 7 and 25 km.

***L. 22-24, p.9: This statement is particularly strange! The Angstrom exponent does have a physical meaning, since it reflects the size properties of the aerosol population. Pursuing as sole purpose the replication of one data set at all costs (even one supposed to be good, although its comparison with the real truth is impossible ? this should always be kept in mind!) and getting rid of any concern about the correct quantification of known underlying effects at this aim, looks problematic to me.***

We have revised text and replaced Angstrom exponent with "Pseudo Angstrom exponent". Please note that we are conforming the data based on OSIRIS and SAGE II/III extinction comparisons as is described in the manuscript. The pseudo angstrom exponent we use here is merely a scaling factor that conforms OSIRIS data to SAGE II/SAGEIII. We do agree that size changes during volcanic events do matter and we currently do not have a way to address this issue. However, we are working on developing a method as to how various volcanic eruptions affect the particle sizes (Thomason et al., 2020), which might help us understand better about the process. We plan to implement a possible correction on size dependence particularly following a volcanic/fire event in a future version of GloSSAC.

***L. 24-25, p.9: ?Angstrom exponent values?.***

Yes. It is now corrected.

*Caption Figure 4, 5, and 8: The quantity provide should be precisely mentioned, e.g.: ?OSIRIS and SAGE II extinction coefficient at 525 nm?. In caption of Figure 5, ?for at? is not correct and ?for? should be removed. In caption of Figure 8, ?Altitude versus Latitude of percent difference.? is meaningless. Difference in what? The authors should also clearly mention the period covered by this plot.*

Done.

*L. 27, p.9: I suggest to stick to the naming ?Angstrom exponent?. Please check the whole document.*

Done. We now checked for consistency and only "Angstrom exponent" is used throughout the document.

*L. 32, p.9: After using an Angstrom exponent of 2.33 to convert OSIRIS extinction coefficient from 750 nm to 525 nm (cf. L. 7, p. 8), another value of the same Angstrom exponent, 1.50, is used to convert the CALIOP extinction coefficient from 532 nm to 525 nm. Why such a difference? This incoherence should be discussed or justified.*

The Angstrom exponent of 1.5 is typically used for CALIOP conversion of aerosol extinction (Vernier et al., 2011). We, however, changed that to 2.33 to be consistent with the values used in the manuscript. We have updated Figure 8a, b that use Angstrom exponent to convert CALIOP extinction.This method is just for the comparison purpose (between this version and the conformed version of CALIOP extinction). And, we note that this conversion does not matter as far as the GloSSAC data is concerned as we are not using extinction computed by this method in GloSSAC. We instead use the conformed aerosol extinction that have been computed using scale factor (Figure 9) as described in the manuscript.

*L. 7-8, p.10: Smaller eruptions also occurred during the SAGE II mission (1984-2005). Is there any similar observations by SAGE II that might support such tendency? This might help depicting if such effect is real, or is the reflect of some limitation either of the OSIRIS instrument, or of the OSIRIS retrieval.*

Yes. There is another paper in ACP which is in review ( https://doi.org/10.5194/acp-2020-480) that discusses how various volcanic eruptions impact aerosol sizes that occurred during SAGE II mission and also in the current SAGE III/ISS mission.

*L. 25, p.10: ?roughly consistent with values for sulfuric aerosol in the stratosphere?: The extinction-to-backscatter ratio shows much variability in the stratosphere (See*

*for example Vernier et al., Geophys. Res. Lett., 38, L12807, doi:10.1029/2011GL047563, 2011), and the size characteristics also play a role in the variability of this parameter. Hence, I think that this statement is not very relevant.*

We revised the sentence and it reads as:

"This value, 53 sr, is roughly consistent with the extinction-to-backscatter ratio used within CALIOP data processing (50 sr)."

*L. 29, p.10: ?As a result?? This sentence is the transition between considerations about version 1.0, and work around version 2. This should be made clear by an adequate introduction. Furthermore, at this stage, it would ease the reading to remind that the CALIOP extinction coefficient product by Kar et al. (2019) is the one used in GLoSSAC, as mentioned in L. 3-4, p.3.*

Not sure what the reviewer's comment is. We have already cited Kar et al. (2019) in here where we mention standard stratospheric aerosol product.

*L. 31-32, p.10: Why are the authors using now another value of the Angstrom exponent (1.50) for the conversion CALIOP, while a value of 2.33 was used before for OSIRIS extinction conversion? This is quite confusing and increase the level of incoherence between the data sets.*

The Angstrom exponent of 1.5 is typically used for CALIOP conversion of aerosol extinction (Vernier et al., 2011). We, however, changed that to 2.33 to be consistent with the values used in the manuscript. We have updated Figure 8a, b that use Angstrom exponent to convert CALIOP extinction.This method is just for the comparison purpose (between this version and the conformed version of CALIOP extinction). And, we note that this conversion does not matter as far as the GloSSAC data is concerned as we are not using extinction computed by this method in GloSSAC. We instead use the conformed aerosol extinction that have been computed using scale factor (Figure 9) as described in the paper.

*L. 7-10, p.11: I don?t understand what the authors intend here. In 3, p.10, it is explained that the CALIOP extinction used in GLoSSAC is the CALIOP extinction product (Kar et al., 2019) at 532 nm, converted to 525 nm based on an Angstrom exponent of 1.50. Why do they use now the CALIOP 532 nm backscatter converted using an empirical scaling factor, with some kind of warning that this scaling factor will also reflect ?any kind of biases?? This is extremely confusing.*

We compared the standard CALIOP extinction product (after using angstrom exponent of 2.33 to convert from 532 nm to 525 nm) with conformed OSIRIS and SAGE III/ISS data. We however, consistently see an enhanced aerosol extinction in the lower stratosphere and also at higher latitudes (poleward of 40 N/S), indicating that the CALIOP data is biased high. We, therefore thought it is appropriate not to use standard CALIOP extinction, but use CALIOP backscatter and the scale

factor (based on OSIRIS 525 nm extinction and CALIOP 532 backscatter) to convert backscatter coefficient to extinction. This scale factor is like a pseudo lidar ratio which has altitude-latitude dependence. We note that Kar et al. (2019) has also pointed out increased extinction levels in the lower stratosphere and higher latitudes in the standard CALIOP data and also computed a lidar ratio (Figure 13 of Kar et al. (2019)) that are retrieved using SAGE III/ISS extinction and CALIOP backscatter measurements which also shows that lidar ratios are variable.

*L. 13-27, p.11: I don?t really understand what the authors are doing here. The CALIOP backscatter is the primary quantity measured by CALIOP. What is the interest of rederiving the primary measured quantity from the CALIOP extinction (derived with a simplified assumption of a constant lidar ratio equal to 50), using an empirical scaling factor taking into account all possible problems (?aerosol-related effects and bias between the two data sets?), based on modified (?bias-corrected?) OSIRIS extinctions at another wavelength with some rough approximation about the atmospheric transmission (mentioned as ?clearly not correct? by the authors themselves) , and a simplified formula to account for the scattering ratio and molecular backscatter. And from the conclusion that ?it does not matter a great deal whether we use the standard CALIOP stratospheric backscatter product or the alternative alternative?, the authors choose using this hazardous construction of alternative backscatter product! This is extremely strange and confusing, and if the aim is ? again ? to ?match? at all costs CALIOP with OSIRIS, the methodology used is, at the least, questionable.*

We initially thought of using the standard particulate backscatter product. We later realized that the particulate backscatter in the Level 3 data file is retrieved using a lidar ratio 50 Sr. So, if we use the retrieved particulate backscatter for computing scale factor (SF) which is based OSIRIS extinction to CALIPSO backscatter ratio, we are in fact using a SF (which is similar to a lidar ratio) on a product that was already retrieved using a constant lidar ratio of 50. We, therefore used an alternate method that does not use any fixed value for lidar ratio. Please note that a similar method has been used earlier for retrieving backscatter measurements (Vernier et al., 2009). Below is a formulation that we used to derive particulate backscatter as described in the current version of the manuscript. We start with the scattering ratio which is defined as the ratio of total backscatter coefficient to molecular backscatter coefficient. We then assume the transmission of atmosphere $(T_{[\lambda,p]}(z)^2)$ is close to 1 as shown in the second step of the formulation. The particulate backscatter $(\beta_{[\lambda,p]}(z))$ is then derived using scattering ratio and molecular backscatter.

$$\text{Scattering Ratio} \equiv SR \equiv \frac{\text{Total Attenuated Backscatter}}{\text{Molecular Attenuated Backscatter}} \tag{1}$$

$$SR = \frac{(\beta_{\lambda,m}(z) + \beta_{\lambda,p}(z))\, T_{\lambda,m}^2(z) T_{\lambda,oz}^2(z) \overset{1}{\cancel{T_{\lambda,p}^2(z)}}}{\beta_{\lambda,m}(z) T_{\lambda,m}^2(z) T_{\lambda,oz}^2(z)} \tag{2}$$

$$SR = \frac{\beta_{\lambda,m}(z) + \beta_{\lambda,p}(z)}{\beta_{\lambda,m}(z)} \tag{3}$$

$$SR = 1 + \frac{\beta_{\lambda,p}(z)}{\beta_{\lambda,m}(z)} \tag{4}$$

$$SR \cdot \beta_{\lambda,m}(z) = \beta_{\lambda,p}(z) + \beta_{\lambda,m}(z) \tag{5}$$

$$\beta_{\lambda,p}(z) = (SR \cdot \beta_{\lambda,m}(z)) - \beta_{\lambda,m}(z) \tag{6}$$

, where $\beta_{[\lambda,m]}(z)$, $\beta_{[\lambda,p]}(z)$, $T_{[\lambda,oz]}(z)^2$, and $T_{[\lambda,p]}(z)^2$ are molecular backscatter, particulate backscatter, ozone and particulate transmittance respectively.

We have computed a percent difference between the standard retrieved backscatter coefficient and the backscatter coefficient computed using the alternate method (inferred backscatter). As shown below in Figure 1a, the percent difference computed between retrieved and inferred backscatter for March 2007. At altitudes above 18 km, the percent difference is below ±10%, while the percent difference increases to about ±30% near below 15 km. While there is increased difference below 18 km, it does not really matter much as we scale those differences away in the conformance process by using OSIRIS extinction to CALIOP backscatter, defined as scale factor (SF) in the manuscript. We then computed the ratio of 525 nm OSIRIS extinction to 532 nm CALIOP backscatter coefficient using both the retrieved and the inferred CALIOP backscatter. Figure 1b,c show the ratio of OSIRIS extinction to retrieved and inferred CALIOP backscatter coefficient respectively for March 2007. There are differences between the two methods particularly below 18 km, where they match with the increased percent difference shown in Figure 1a. While the SF computed using retrieved CALIOP backscatter shows values below 30 sr below 18 km (Figure 1b), the SF using inferred CALIOP backscatter shows a higher SF which is around 40 sr (Figure 1c). Generally, below 18 km the retrieved backscatter coefficient is larger than inferred backscatter coefficient. However, these differences are scaled away in the conformance process where we use OSIRIS extinction to CALIOP backscatter ratios (SF) as they are evident from Figure 1. We, therefore believe that our alternate method to infer backscatter coefficient is not a hazardous construction of backscatter.

[Figure]

Figure 1: Percent difference and extinction to backscatter ratios for 200703. (a) percent difference between standard retrieved and inferred CALIOP backscatter coefficient computed as (Inferred-Standard/Standard)*100, (b) 525 nm OSIRIS extinction to retrieved 532 nm CALIOP backscatter and (c) 525 nm OSIRIS extinction to inferred 532 nm CALIOP backscatter.

Additionally, as pointed out by reviewer 3, Kar et al. (2019) reported large differences ($> \pm 50\%$) with SAGE III/ISS below 20 km, which could be due to the presence of clouds. These differences are consistent with OSIRIS data as well. This was the reason we decided to use the conformance process based on the extinction to backscatter ratio as shown in Figure 9 of the manuscript, also shown in Figure 1 above.

**L. 29-30, p.11: The SF values varying between 25 and 65 might reflect the objective to get rid of the fixed 50-value of the lidar ration used by Kar et al. (2019) to better match local aerosol features. If this indeed is the case, the authors should completely revise this discussion to make it clear, and they should justify why they expect improvement with respect to Kar et al. (2019), see previous comment.**

Our response to the previous comment in detail, answers this comments as well.

*Figure 5: It is very strange to mix both SAGE II and SAGE II/ISS overlap periods as if these two SAGE sensors were one single data set or mission. SAGE II and SAGE III/ISS are two different instruments measuring different situations in very different conditions. Assimilating the SAGE II and SAGE III/ISS to one single perfect data set looks excessive, and at least, results for both data sets should also be shown (or quantified in some way) to justify that just mixing both is appropriate.*

Since the SAGE III/ISS instruments operates in a manner similar to SAGE II and SAGE III meteor, the expectation is that there would be minimal bias between these instruments at least at 525 and 1020 nm. Some previous studies have compared/validated SAGE II with SAGE III meteor (Thomason et al., 2010; Damadeo et al., 2013). While the differences between SAGE II and SAGE III meteor aerosol extinction coefficient are relatively smaller, previous studies (Thomason et al., 2010; Damadeo et al., 2013) reported a small bias between SAGE II (v 7.0) and SAGE III (v 4.0) meteor that are within ± 10% for measurement wavelengths at 525 and 1020 nm for the altitudes between 7 and 25 km. We are not sure if any quantification can be done with SAGE III/ISS and SAGE II as they differ in measurement time period. The best we can do is to compare these measurements with OSIRIS that have overlap measurements with these two instruments, which we have done in the manuscript. We do not think assimilating SAGE III/ISS and SAGE II will create any bias in the monthly climatology of pseudo angstrom exponent, followed by the conformance of the data.

*L. 1-4, p.12: The methodology used here is expected to provide more variations of the extinction-to-backscatter ratio than the fixed one assumed by Kar et al. (2019). However, the question is to know it the whole construction with a succession of more or less coarse assumptions used here provide a better estimate of this parameter. See also comment on L. 29-30, p.11*

Please see a detailed response to an earlier comment L 13-27, p. 11.

Additionally, as pointed out by reviewer 3, Kar et al. (2019) reported large differences ($> \pm 50\%$) with SAGE III/ISS below 20 km, which could be due to the presence of clouds in CALIOP data. These differences are consistent with OSIRIS data as well. We also note the pattern of the lidar ratio based on SAGE III/ISS extinction and CALIOP backscatter (Figure 13 of Kar et al. (2019)) is more or less consistent with our SF in Figure 9a.

*L. 20, p.13: Is the linear interpolation implemented only in the time dimension? What about the possible use of equivalent latitudes? This should be specified.*

When no data is available, grids are filled using linear interpolation in time.

The sentence is now revised and it reads as:

"It should also be noted that, in some cases (particularly for February 2016 and October 2018) when no CALIOP data is available, we linearly interpolate CALIOP data in time between January (September) and March (November) of 2016 (2018) to fill in the missing monthly data following methods used in interpolating SAGE II data."

*Figure 15: The choice of dynamic range for the color scale of pannels 15(a), (b), (d), and (e) is particularly poor. Same for Figure 16. Differences mentioned in L. 2-3, p.14 are hardly visible, and the ?substantially smaller enhancement? in 2005 in version 2.0 with respect to version 1.1, is just invisible in both cases to me.*

Done.

***L. 4-11, p.14: Could several latitudinal dependence and hemispheric dependences possibly be explained by differences in data coverage and/or in intrumental techniques? This possibility has not been discussed.***

Some differences may be the result of the different performances of the instruments and that those differences may well be of latitude dependence (e.g. like the scattering angle effect for limb scatter instruments).

***L. 22-27, p.14: I think, indeed, that in view of all efforts made to force some data sets to fit in as much as possible some other one, any discussion about trends Is absolutely premature.***

Please note that the conformance was done based on studying each data set carefully. Many previous studies have reported the difference between SAGE and OSIRIS measurements (e.g. Bourassa et al., 2012; Rieger et al., 2015) in the first place and for CALIOP, with the standard stratospheric aerosol product we clearly see an enhancement in the lower stratospheric/higher latitude data which is also pointed out by Kar et al. (2019). So, the conformance process was based on studying each data set carefully. While the conformance process applied here merely force the data toward SAGE II/ III-ISS based on intercomparison of individual data sets, we do not believe any signatures of volcanic/fire aerosol is compromised as they are evident in the entire record. Additionally, these signatures are evident from figures 10, 12, 15, 16, 17 and 18.

***L. 24, p.2: incorrect sentence: ?whose accuracy? should be removed.***

Done.

***L. 25, p.2: ?Which this change?? (Or another change?)***

The sentence is revised and now reads as:

"With these changes, the retrieved extinction coefficient at 750 nm is in better agreement with observations by SAGE II and SAGE III/ISS than the version used in GloSSAC v1.0 (v5.07)."

***L. 3, p.5: missing period (?.?).***

Done.

***L. 16-17, p.4: odd sentence.***

We are not sure what the author is referring to. The sentence seem to be correct.

***P. 25, p.6: New sentence starting with ?However, ? ?***

Revised the sentence and now reads as:

"While the OSIRIS version 7.0 aerosol data product is similarly cloud screened (Rieger et al., 2019),we, however found some additional clearing was beneficial to the analysis."

**L. 3, p.5: ?its?.**

We do not see any "its" in this line though.

**L. 20, p.7: incorrect sentence: ?can transition?.**

The sentence is now revised and reads as: "Cloud identification is complicated by mixed fields of view where observations transition between mostly cloudy extinctions and extinction ratios and those more typical of purely aerosol."

**L. 24, p.7: incorrect reference: should be ?Thomason and Vernier (2013)?.**

Done. The sentence now reads as: "Various techniques to parse these mixed measurements have been developed and GloSSAC makes use of the technique developed by Thomason and Vernier (2013). "

**L. 16, p.9: ?Extinction?.**

The sentence is now revised and reads as:

"where, $k_{525[t,m,i,j]}$, and $k_{750[t,m,i,j]}$ are extinctions at 525 nm and 750 nm respectively, $\eta_{[m,i,j]}$ is the pseudo Angstrom exponent while the indices $[t,m,i,j]$ represent year, month, latitude, and altitude respectively. $(\frac{\lambda_{525}}{\lambda_{750}})$ represents ratio of wavelengths at 525 and 750 nm. All data are gridded to 5 degree latitude and 0.5 km altitude resolution. "

**Caption Figure 9: The authors should be more explicit: ?(b) Relative standard deviation of the extinction-to-backscatter ratio shown in (a)?. ?deviation of (a) in percent? is unclear.**

Done. It now reads as

"(b) Relative standard deviation of (a) is computed at each grid point with respect to the median value in percent"

**L. 30, p. 12: ?We use? with capital letter.**

Done.

**L. 31-32, p.12, L. 18, p.15, and caption Figure 11: ?SAGE II and SAGE III/ISS?.**

Revised as "between OSIRIS and SAGE II/SAGE III-ISS".

***L. 19-21, p.19: Rieger et al. (2019) is published, and the reference should be adapted.***
Done.

---

## Author Comment (AC7) · 18 Aug 2020

**Response to Reviewer #3**

We thank the reviewer for helpful comments. Our responses to the reviewer's specific comments are listed below. The reviewer's concerns are in bold italicized font and our responses are in regular font. The page numbers and line numbers given in our responses below are in reference to the revised version of the manuscript.

General comments: The paper describe the development of the stratospheric aerosol data set GloSSAC V2.0 and changes made since the last release of V1.1. The methodology of the data set construction and rational is clearly described. The paper is well written, and the results are presented and discussed with sufficient details. I recommend for publication subject to the following changes.

Major comment: I find the authors choice of conforming rather than excluding data that exhibit very large biases (> 50 %) somewhat concerning. Mainly because some of these data may be affected by clouds. Kar et al. (2019) reported large differences with SAGE III/ISS below 20 km, which can be explained in part by subvisible cirrus cloud scattering artifacts that may appear to within several kilometers above the tropopause. In addition, the authors failed to explain the advantage of using the newly released CALIOP standard products instead of version 4.0 Level 1 data used in GloSSAC V1.0. Do they believe that their approach produces better product than what was used in V1.0 and V1.1?

When we have started our efforts to incorporate CALIOP data in to GloSSAC v 2.0 by using version 4.0, level 1 data as it was used in GloSSAC 1.0 and 1.1, the official CALIPSO stratospheric aerosol product was released. The release of CALIPSO standard stratospheric aerosol product was followed by a paper that was published (Kar et al., 2019). After the release of the official CALIOP stratospheric aerosol product, it was appropriate for us to use the standard stratospheric product from CALIPSO team instead of using version 4.0 level 1.0 data. To some extent we had no choice but to use the CALIPSO data as it is a priority for the data set to be gap free. The old technique used a fixed extinction to backscatter ratio that created what we believe have biases in the lower stratosphere. The new approach seems to minimize this issue. We do agree with the reviewer that there is large bias in CALIOP data, sometime exceeding 50% when compared to other data products. This was the reason we decided to use the conformance process based on the extinction to backscatter ratio as shown in Figure 9. As you can see from Figure 9a that the extinction to backscatter ratio decreases to even less than 30 sr as noted in the manuscript. By using these values, we believe the bias in the lower stratosphere has been reduced to a greater extent although not completely bias free. We also note that CALIOP data is used only when other extinction measurements (OSIRIS or SAGE III/ISS) are not available which occurs mostly at higher latitudes. Attached is a figure that shows the percent difference between the standard CALIOP product and the bias corrected CALIOP extinction (after conformance process) for 201711. It is evident from this figure that the conformance process help reduce the bias in the data particularly in the lower stratosphere where the percent difference (STANDARD-CORRECTED/CORRECTED)\*100) exceeds 80% indicating the conformed CALIOP extinctions values have been reduced significantly. We do agree that this will not completely remove the bias in the lower stratosphere but it clearly helped reduce the bias in the lower stratosphere in particular.

Figure 1: CALIOP extinction coefficient and percent difference for 201711. (a) Standard CALIOP extinction coefficient (CS) at 525 from the standard CALIOP data product, (b) CALIOP data after conformance (CC), following method described in manuscript, and (c) Percent difference computed as (CS-CC/CC)\*100. The standard CALIOP extinction is available at 532 nm and is converted to 525 nm using a constant Angstrom exponent of 2.33 in (a).

**General comments:**

**Table 1: Some of the volcanic eruptions listed in table 1 did not reach the stratosphere and thus are not relevant to this dataset. The table should be modified to include only volcanic eruption that are evident in GloSSAC dataset.**

We have now revised Table 1 to extend the record back to 1979 as one of the reviewers suggested to include the entire record of GloSSAC. We also label these in Figure 15, 16, and 18.

**Figure 1: the figure is identical to figure 1 published earlier by (Thomason et al., 2018), which shows SAGE III/ISS as a future instrument. I suggest either updating the figure or simply just cite the figure in (Thomason et al., 2018).**

We have revised this figure and made necessary changes so that the figure now shows SAGE III/ISS as current instrument.

Page 3, first paragraph: The authors need to add a brief statement justifying the change and summarizing the differences between the new CALIOP products and the

**one used in V1.1.**

We have added a brief statement toward the end of the paragraph and it now reads: "While we use the standard CALIPSO stratospheric aerosol product, enhanced levels of aerosol extinction in the lower stratosphere are consistently noted in the entire dataset after comparing against OSIRIS and SAGE III/ISS. We, therefore decided to use a conformance process which is described below that helps reduce the bias in the lower stratosphere and also at higher latitudes."

**Page 8 L6: ?OSIRIS extinction is also routinely produced at 525 nm, : : :? should be replaced by ?OSIRIS extinction can be produced at 525 nm, : : :? or something like that. The original text implies that it is part of the official V7.0 release.**

The sentence is revised and now reads as " In addition to 750 nm, OSIRIS extinction can be calculated at 525 nm...."

**Page 8 L10: ?Since the SAGE III/ISS instruments operates in a manner virtually identical to SAGE II? ?virtually identical? should be replaced by ?similar? since the two instruments have different designs and age. Toward the end of its life, SAGE II was an aging instrument that operated on reduced duty cycle as compared to the newly refurbished SAGE III/ISS instrument.**

The sentence now reads as "Since the SAGE III/ISS instruments operates in a manner similar to SAGE II, the expectation is that there would be minimal bias between these instruments at least at the strongest aerosol measurement wavelengths at 525 and 1020 nm."

**Page 8, L15: ?SAGE II and SAGE III/ISS are relatively unbiased with each other? this not accurate since both (Thomason et al., 2010) and (Damadeo et al., 2013) reported 10 % bias between SAGE II and SAGE III Meteor, which is supposed to be identical to SAGE III/ISS. The differences between SAGE II and III should be acknowledged and discussed in this section.**

We now add another footnote about the bias between SAGE II and SAGE III meteor which now reads as "While the differences between SAGE II and SAGE III meteor aerosol extinction coefficient are relatively smaller, some previous studies (Thomason et al., 2010; Damadeo et al., 2013) reported a small bias between SAGE II (v 7.0) and SAGE III (v 4.0) meteor that are within  $\pm 10\%$  for measurement wavelengths of 525 and 1020 nm for the altitudes between 7 and 25 km."

**Page 8, footnote 1: ?While the OSIRIS instrument performance has remained unchanged over time,? This not exactly accurate. According to Bourassa et al. (2018) and Rieger et al. (2019), OSIRIS had a small drift that resulted in a pointing error and a correction was applied to V7.0. Please modify the text accordingly.**

Yes. We changed the footnote to "While the OSIRIS instrument performance has relatively remained unchanged over time, the scattering angle has slowly drifted, and the fraction of ascending/descending node measurements has changed. These factors may affect overall data quality."

**Page 9, L23: I suggest changing ?Angstrom exponent? (where appropriate) to something like ?pseudo Angstrom exponent? to eliminate any confusion regarding its physical meaning.**

Done.

Figure 7: Can you add the year to the volcanic eruption label?

Done.

Section 3.1: The paragraph describing the choice between using the standard CALIOP stratospheric backscatter or the alternative product is confusing and difficult to follow, especially when the authors conclude that ?it ultimately does not matter a great deal whether we use the standard CALIOP stratospheric backscatter product or the alternative product described above?. If that is the case, why not use the standard product and eliminate the confusion? Also, the CALIPSO section in supplementary materials implies that the standard products were used.

Sorry about the confusion. We initially thought of using the standard particulate backscatter product. We later realized that the particulate backscatter in the Level 3 data file is retrieved using a lidar ratio 50 Sr. So, if we use the retrieved particulate backscatter for computing scale factor (SF) which is based OSIRIS extinction to CALIPSO backscatter ratio, we are in fact using a SF (which is similar to a lidar ratio) on a product that was already retrieved using a constant lidar ratio of 50. We, therefore used an alternate method that does not use any fixed value for lidar ratio.

We have computed a percent difference between the standard retrieved backscatter coefficient and the backscatter coefficient computed using the alternate method (inferred backscatter). The Figure 2a below, shows the percent difference computed between retrieved and inferred backscatter for March 2007. At altitudes above 18 km, the percent difference is below  $\pm 10\%$ , while the percent difference increases to about  $\pm 30\%$  near below 15 km. While there is increased difference below 18 km, it does not really matter much as we scale those differences away in the conformance process by using OSIRIS extinction to CALIOP backscatter, defined as scale factor (SF) in the manuscript. We then computed the ratio of 525 nm OSIRIS extinction to 532 nm CALIOP backscatter coefficient using both the retrieved and the inferred CALIOP backscatter. Figure 2b,c show the ratio of OSIRIS extinction to retrieved and inferred CALIOP backscatter coefficient respectively for March 2007. There are differences between the two methods particularly below 18 km, where they match with the increased percent difference shown in Figure 2a. While the SF computed using retrieved CALIOP backscatter shows values below 30 sr below 18 km (Figure 2b), the SF using inferred CALIOP backscatter shows a higher SF which is around 40 sr (Figure 2c). Generally, below 18 km the retrieved backscatter coefficient is larger than inferred backscatter coefficient. However, these differences are scaled away in the conformance process where we use OSIRIS extinction to CALIOP backscatter ratios (SF) as they are evident from Figure 2b,c.